# Synthesizing Global Carbon-Nitrogen Coupling Effects – the MAGICC Coupled Carbon-Nitrogen Cycle Model v1.0

Gang Tang[1], Zebedee Nicholls[2], Alexander Norton[3], Sönke Zaehle[4], Malte Meinshausen[1]

[1]School of Geography, Earth and Atmospheric Sciences, The University of Melbourne, Melbourne, Australia
[2]Energy, Climate and Environment (ECE) Program, International Institute for Applied Systems Analysis (IIASA), Laxenburg, Austria
[3]Research School of Biology, Australian National University, Canberra, Australia
[4]Department of Biogeochemical Signals, Max Planck Institute for Biogeochemistry, Jena, Germany

*Correspondence to*: Gang Tang (gang.tang.au@gmail.com, gang.tang@student.unimelb.edu.au)

**Abstract.** The integration of a nitrogen cycle represents a recent advancement in Earth System Models (ESMs). However, diverse formulations introduce uncertainty in the nitrogen effect on the carbon cycle, leaving the global carbon-nitrogen coupling effect unclear. In this study, we present a newly developed carbon-nitrogen cycle model. By way of demonstration, this model is coupled into the reduced complexity model (RCM) MAGICC. We have calibrated this coupled carbon-nitrogen cycle model to two land surface models (CABLE and OCN) and (the land component of) a set of CMIP6 ESMs. The new
coupled carbon-nitrogen model is able to capture the dynamics of the more complex models' carbon-nitrogen cycle at the global-mean, annual scale. The emulation results suggest a consistent nitrogen limitation on net primary production (NPP) in CMIP6 ESMs, persisting throughout the simulations (i.e. over the period 1850-2100) in most models. The emulation may provide a way to disentangle diverse nitrogen effects on carbon pool turnovers in CMIP6 ESMs, with our results suggesting that nitrogen deficiency generally inhibits litter production and decomposition while enhancing soil respiration (from a
multi-model mean perspective). However, this disentanglement is limited due to a lack of simulations from CMIP6 ESMs which would allow us to fully separate the nitrogen and carbon responses. The results imply a potential reduction in land carbon sequestration in the future due to nitrogen deficiency. Future studies will use the newly developed model to further investigate the carbon-nitrogen coupling effect, including uncertainty, in future climate projections.

## 1 Introduction

Atmosphere-Ocean General Circulation Models (AOGCMs) and Earth System Models (ESMs) are currently the most powerful tools we have for integrating our understanding of climate physics and providing comprehensive projections of the global climate and its variability (Meehl, 1990). However, these complex models require large computational power for their simulations while the difference in assumptions, parameterizations and structures across models often hinders a systematic quantification of uncertainties (Ohgaito et al., 2013). To combine the latest insights from various AOGCMs and ESMs,
Simple Climate Models (SCMs) - also called Reduced-Complexity Climate Models (RCMs) - are developed and routinely

updated to represent and integrate the full uncertainty spectrum across the cause-effect chain of climate change (Nicholls et al., 2020; Nicholls et al., 2021). The highly parameterized formulations used in RCMs can, in some cases, parameterize the structural uncertainties from complex models. As a result, this flexibility of RCM structures allows for factor separation analysis to disentangle the key processes affecting the climate. With these features, RCMs are widely used for ensemble projections of scenarios and regularly feed into climate policy.

The Model for the Assessment of Greenhouse Gas Induced Climate Change (MAGICC), originally introduced by Wigley and Raper (Wigley and Raper, 1987, 1992, 2001) and further developed since (Meinshausen et al., 2011a; Nauels et al., 2017; Meinshausen et al., 2020), is a key RCM that has been used for scenario classification in multiple IPCC reports [e.g., Climate Change 2014: Synthesis Report (IPCC, 2015); Global Warming of 1.5°C (IPCC, 2019); Climate Change 2023: Synthesis Report (IPCC, 2023)]. MAGICC's main design principle is this: be as simple as possible while as mechanistic as necessary in the sense of being based on physical principles and/or long-term ESM calibrations (Meinshausen et al., 2011a).

The nitrogen cycle is a critical part in the Earth system's biogeochemistry which has a significant impact on climate alongside other element cycles like carbon, phosphorus, etc. (Fowler et al., 2013; Elser et al., 2007). As an essential nutrient for numerous fundamental biological processes, nitrogen is one of the major factors controlling the terrestrial carbon cycle and thus influences the carbon-concentration and carbon-climate feedbacks (Zaehle et al., 2010; Zaehle and Dalmonech, 2011; Fowler et al., 2013; Zaehle, 2013), the two main carbon cycle feedbacks (Arora et al., 2020). The integration of the nitrogen cycle and its effects within carbon cycle models is a recent advancement in ESMs. Only three CMIP5 ESMs (CCSM, CESM, NorESM), all of which had the same land component (CLM4), included the nitrogen cycle (Flato et al., 2014). However, at least 17 out of 39 CMIP6 ESMs included a nitrogen cycle [see Climate Change 2021: The Physical Science Basis. Annex II: Models (IPCC, 2021)]. Various assumptions and formulations have been incorporated into the nitrogen cycle and the carbon-nitrogen coupling (Meyerholt and Zaehle, 2015; Meyerholt et al., 2020), resulting in divergent responses of the carbon cycle (Zaehle et al., 2015; Davies-Barnard et al., 2020; Arora et al., 2020; Kou-Giesbrecht and Arora, 2022).

Generally, the inclusion of the nitrogen cycle reduces land carbon sequestration under increasing atmospheric CO2 and warming conditions by limiting photosynthesis (thus limiting NPP) and amplifying both plant respiration and soil organic matter decomposition (Thornton et al., 2007; Sokolov et al., 2008). On average, the carbon-nitrogen coupled ESMs have smaller carbon-concentration feedbacks and smaller carbon-climate feedbacks (weaker absolute strength of the feedback parameters) compared to their carbon-only counterparts (Arora et al. 2020). Plant nitrogen uptake, carbon:nitrogen ratio, nitrogen regulation of photosynthesis, and biological nitrogen fixation contribute to the NPP difference (Du et al., 2018). Carbon-nitrogen interaction simulations from "Jena Scheme for Biosphere-Atmosphere Coupling in Hamburg" (JSBACH) have suggested a moderate reduction of the carbon-concentration feedback, while showing a negligible effect on the carbon-climate feedback (Goll et al., 2017). However, enhanced soil organic matter decomposition under warming increases mineral

nitrogen availability, thereby leading to increased land carbon sequestration on vegetation. The relative strength of these compensating effects remains unclear. Therefore, there is a need for better understanding and comparing ESMs, with the integration and parameterization of a nitrogen cycle in RCMs providing one way to develop this understanding and comparison.

The significance of the nitrogen cycle also highlights the need to capture its effects within a key tool in climate science, namely reduced complexity climate models. To the best of our knowledge, there is currently no RCM featuring the nitrogen effect, let alone a fully coupled carbon-nitrogen cycle. As a result, this study introduces a coupled carbon-nitrogen model (referred to as CNit), which has evolved from the previous MAGICC carbon cycle model. Section 2 presents a detailed description of the CNit model. Section 3 provides the results of offline calibration of CNit, firstly to two land surface models and then to a series of CMIP6 ESMs across multiple scenarios. Section 4 offers related discussions and analysis of the carbon-nitrogen coupling effect, primarily focusing on the CMIP6 ESMs. Section 5 discusses the limitations and implications of CNit's emulation. The results demonstrate that CNit captures the global aggregate effects of coupling the nitrogen cycle with the carbon cycle, in line with the latest generation of specialized domain models and ESMs. Future work will use MAGICC, updated to include CNit, to explore one of the key uncertainties in future climate projections: the uncertain evolution of future $CO_2$ concentrations given the intertwined carbon cycle feedback, $CO_2$ fertilization, and nitrogen cycle effects.

## 2 Model description

### 2.1 Overview of MAGICC and CNit

MAGICC is one of the most widely used RCMs. MAGICC features variable climate sensitivities and a carbon cycle that has successfully emulated a series of CMIP3 AOGCMs and C4MIP carbon cycle models (Meinshausen et al., 2011a). The most recent updates of MAGICC include the introduction of variable climate sensitivities and the updated carbon cycle (Meinshausen et al., 2011a), the incorporation of a sea level model (Nauels et al., 2017) and various improvements over time with regard to radiative forcing schemes etc. (Meinshausen et al., 2020). In its latest calibration, MAGICC was shown to reproduce the IPCC AR6 WG1 assessment well over a range of metrics (see Chapter 7 in Climate Change 2021: The Physical Science Basis).

The continuously expanding understanding of climate physics, chemistry and biology, coupled with the rapid development of complex models, necessitates corresponding advancement of RCMs. Here we focus on the development of the terrestrial carbon-nitrogen cycle model in MAGICC, referred to as CNit, which builds upon the previous MAGICC carbon cycle (Meinshausen et al., 2011a). As a brief background, the initial design of CNit considered carbon and nitrogen processes at a similar level of detail as complex models do (albeit at a global scale rather than grid-box scale, e.g., the box model design starts from the major state variables and fluxes that are required by C4MIP) (Jones et al., 2016). However, during model

parameterization and refinement, some processes were simplified or integrated with others to improve efficiency. For

instance, biological nitrogen fixation is directly allocated to organic nitrogen pools, bypassing the intermediate step of mineral nitrogen enrichment and subsequent plant uptake (Fig. 1). Additionally, certain representations, such as land-use emissions, were updated to achieve a balance between model simplicity and mechanistic insights (Section 2.7). These refinements align with MAGICC's need for computational efficiency and overall design philosophy of being as simple as possible but as mechanistic as necessary.

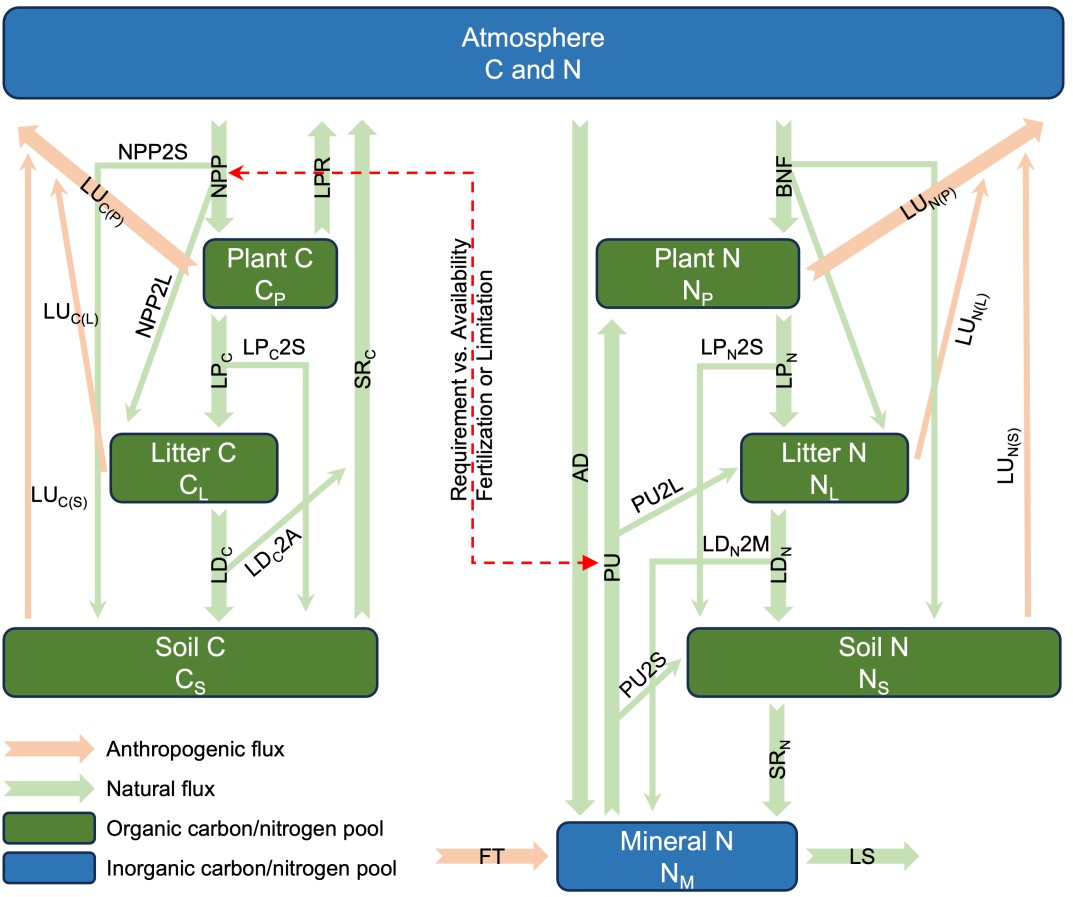

**Figure 1. The 'coupled carbon-nitrogen cycle' (CNit) model in MAGICC (NPP: net primary production, LPR: litter production respiration, PU: nitrogen plant uptake, BNF: biological nitrogen fixation, LP: litter production, LD: litter decomposition, SR: soil respiration, LS: mineral nitrogen loss, LU: land use emission, AD: atmosphere nitrogen deposition, FT: nitrogen fertilizer application, 2P, 2L, 2S, and 2M: the partition of fluxes into plant, litter, soil, and mineral pools). The carbon and nitrogen mass**

**conservation is described in Section 2.2. The NPP simulation is described in Section 2.3. The carbon-nitrogen coupling is described in Section 2.4. The LPR flux is described in Section 2.5. Each pool's turnover time and its response to climate and carbon-nitrogen coupling are described in Section 2.6. The land use emission parameterisation is described in Section 2.7.**

CNit is a globally integrated, annually averaged box model (Fig. 1) designed to simulate terrestrial carbon and nitrogen dynamics. It includes carbon and nitrogen pools for 'plant' (P), 'litter' (L), and 'soil' (S), along with an inorganic 'mineral'

(M) nitrogen pool. The 'atmosphere' (A) exchanges carbon with the land carbon pools via 'net primary production' (NPP), 'heterotrophic respiration' (RH), and 'land-use or other anthropogenic fluxes' (LU$_C$). Similarly, the atmosphere exchanges nitrogen with the land nitrogen pools via 'nitrogen atmospheric deposition' (AD), 'biological nitrogen fixation' (BNF), 'gaseous nitrogen loss' (LS2A), and land-use or anthropogenic fluxes LU$_N$. CNit takes the land use emissions of carbon and nitrogen, 'nitrogen fertilizer application' (FT), AD, and BNF, as the inputs. Then, it models key fluxes and solves a system of mass conservation equations to determine the fluxes and states for carbon and nitrogen. The resulting net land-to-atmosphere carbon and nitrogen fluxes are then used to estimate atmospheric concentrations, which subsequently inform radiative forcing and climate responses. These climate responses, in turn, interact with the carbon-nitrogen cycle, creating a feedback loop (see details in Meinshausen et al., 2011a).

The following sections outline the key components of CNit: Section 2.2 introduces the mass conservation framework and key fluxes; Section 2.3 details the NPP simulation; Section 2.4 explains carbon-nitrogen coupling, where we link 'nitrogen plant uptake' (PU) and NPP; Section 2.5 describes the litter production respiration flux; Section 2.6 focuses on carbon and nitrogen turnover calculations; and Section 2.7 addresses the implementation of land-use emissions.

**2.2 Carbon and nitrogen mass conservation in CNit**

The pools are interlinked by a system of first-order differential equations (Eqs. 1-9). In the equations, the subscripts 'c' and 'n' for the turnover fluxes ['litter production' (LP), 'litter decomposition' (LD), and 'soil respiration' (SR)] and land use fluxes (LU) denote the carbon and nitrogen fluxes. The subscripts X2Y for the partitioning factor '$f$' refer to the fraction of flux X that enters pool Y, whereas for land use flux, the subscripts LU2Y specifically represent the partitioning of the flux leaving pool Y due to land-use changes. The flux partition accounts for the intended time domain of applicability - specifically, the long timestep that integrates carbon and nitrogen cycle dynamics over periods of one year or more. Within this framework, fluxes such as NPP, BNF, PU, LU, and turnover fluxes contribute not only to changes in their immediate target pools but also propagate to subsequent pools. For instance, the NPP flux is simultaneously partitioned to plant, litter, and soil carbon pools (Eqs. 1-3). The partitioning factors always sum to unity, ensuring that no mass is lost as a result of this partitioning.

The carbon mass balance in plant ($C_P$), litter ($C_L$), and soil ($C_S$) pools:

$$\frac{dC_P}{dt} = f_{NPP2P}NPP - LPR - LP_c - f_{LU2P_c}LU_c \tag{1}$$

$$\frac{dC_L}{dt} = f_{NPP2L}NPP + f_{LP2L_c}LP_c - LD_c - f_{LU2L_c}LU_c \tag{2}$$

$$\frac{dC_S}{dt} = f_{NPP2S}NPP + f_{LP2S_c}LP_c + f_{LD2S_c}LD_c - SR_c - f_{LU2S_c}LU_c \tag{3}$$

Note that the 'litter production respiration' (LPR) is the litter respiration produced from plant litter production that is released back to the atmosphere within a single timestep (typically one year for MAGICC and CNit). Further details on the LPR flux are provided in Section 2.5.

For the total land carbon (sum of plant, litter, and soil carbon, i.e., combining Eqs. 1-3):

$$\frac{dC_{LAND}}{dt} = NPP - RH - LU_c \tag{4}$$

Note that the RH includes LPR, SR, and the 'litter decomposition that directly goes into atmosphere' (LD2A, i.e. litter respiration).

The nitrogen mass balance in plant ($N_P$), litter ($N_L$), soil ($N_S$), and mineral ($N_M$) pools:

$$\frac{dN_P}{dt} = f_{BNF2P}BNF + f_{PU2P}PU - LP_n - f_{LU2P_n}LU_n \tag{5}$$

$$\frac{dN_L}{dt} = f_{BNF2L}BNF + f_{PU2L}PU + f_{LP2L_n}LP_n - LD_n - f_{LU2L_n}LU_n \tag{6}$$

$$\frac{dN_S}{dt} = f_{BNF2S}BNF + f_{PU2S}PU + f_{LP2S_n}LP_n + f_{LD2S_n}LD_n - SR_n - f_{LU2L_n}LU_n \tag{7}$$

$$\frac{dN_M}{dt} = AD + FT + f_{LD2M_n}LD_n + SR_n - PU - LS \tag{8}$$

Note that a) The 'mineral nitrogen loss' (LS) is the mineral nitrogen turnover; b) The 'nitrogen plant uptake' (PU) is the mineral nitrogen taken up by the organic nitrogen pools; c) The mineral nitrogen pool receives additional nitrogen from 'nitrogen fertilizer application' (FT); and d) The sum of the fraction of litter decomposition nitrogen entering the mineral pool (LD2M$_n$, i.e., litter mineralization) and the nitrogen released during soil respiration (SR$_n$, i.e., soil organic matter mineralization) constitutes the ecosystem's 'net mineralization' (netMIN).

For the total land nitrogen (sum of plant, litter, soil, and mineral nitrogen, i.e., combing Eqs. 5-8):

$$\frac{dN_{LAND}}{dt} = BNF + AD + FT - LS - LU_n \tag{9}$$

**2.3 NPP simulation: Effect of CO₂ and temperature forcings**

The NPP flux is modeled by scaling an initial NPP ($NPP_0$) with the effect from changes in atmospheric CO₂ ($\epsilon_{CO_2}$), temperature change ($\epsilon_{dT(NPP)}$), carbon-nitrogen coupling ($\epsilon_{CN(NPP)}$), and land use change ($\epsilon_{LU}$):

$$NPP = NPP_0 \times \epsilon_{CO_2} \times \epsilon_{dT(NPP)} \times \epsilon_{CN(NPP)} \times \epsilon_{LU} \tag{10}$$

### 2.3.1 CO₂ fertilization

The $CO_2$ fertilization formulations can take multiple forms. The first, the logarithmic formulation, is adapted from (Bacastow and Keeling, 1973):

$$\epsilon_{CO_2}^{log} = 1 + s_{CO_2}^{log} \times ln\left(CO_2/CO_{2ref}\right) \tag{11}$$

where $s_{CO_2}^{log}$ represents the sensitivity of NPP to the logarithm of the ratio of current atmospheric CO₂ concentration ($CO_2$) to a reference CO₂ level ($CO_{2ref}$, e.g., the pre-industrial CO₂ concentration) (i.e., the relative change of $CO_2$ to $CO_{2ref}$).

The second, the rectangular hyperbolic formulation, is adapted from (Hunt et al., 1991; Gifford, 1993):

$$\epsilon_{CO_2}^{rect} = \frac{1/\left(CO_{2ref} - CO_{2b}\right) + s_{CO_2}^{rect}}{1/\left(CO_2 - CO_{2b}\right) + s_{CO_2}^{rect}} \tag{12}$$

where $CO_{2b}$ is the CO₂ concentration when NPP = 0, which has a default value of 31 ppm (Gifford, 1993) and $s_{CO_2}^{rect}$

determines the CO₂ sensitivity of NPP in the rectangular hyperbolic formulation.

When the CO₂ concentration increases from 340 to 680 ppm, the ratio of the feedback factor at 680 ppm to that at 340 ppm ($r$) is designed to be the same for both formulations to ensure better compatibility:

$$r = \epsilon_{CO_2}^{log}(680)/\epsilon_{CO_2}^{log}(340) = \epsilon_{CO_2}^{rect}(680)/\epsilon_{CO_2}^{rect}(340) \tag{13}$$

The sensitivities of NPP in the two formulations are therefore related by:

$$r = \frac{1 + s_{CO_2}^{log} \times ln\left(680/CO_{2ref}\right)}{1 + s_{CO_2}^{log} \times ln\left(340/CO_{2ref}\right)} \tag{14}$$

$$s_{CO_2}^{rect} = \frac{(680 - CO_{2b}) - r(340 - CO_{2b})}{(r-1)(680 - CO_{2b})(340 - CO_{2b})} \tag{15}$$

The previous MAGICC carbon cycle model uses a linear combination of the logarithmic and rectangular hyperbolic formulations to calculate the final CO₂ fertilization effect (Meinshausen et al., 2011a). However, because the logarithmic formulation is an unbounded function, the linear combination becomes unbounded as well unless the logarithmic formulation

is removed, resulting in an overreliance on the rectangular hyperbolic formulation. The rectangular hyperbolic formulation itself can increase steeply if the CO₂ sensitivity of NPP in the rectangular formulation ($s_{CO_2}^{rect}$) is small. Considering $s_{CO_2}^{rect}$ is dependent on $r$ (or $s_{CO_2}^{log}$, Eqs. 14 and 15), the small $s_{CO_2}^{log}$ value is easily attainable, thereby leading to a high CO₂ fertilization factor. To fix this problem, a sigmoidal CO₂ fertilization formulation is introduced and included in the updated carbon-nitrogen model presented here.

$$\epsilon_{CO_2}^{sig} = \frac{\epsilon_{CO_2max}^{sig}}{1 + \left(\epsilon_{CO_2max}^{sig} - 1\right) \times e^{-s_{CO_2}^{sig}(CO_2 - CO_2ref)}} \tag{16}$$

where $\epsilon_{CO_2max}^{sig}$ denotes the maximum of the sigmoidal $CO_2$ fertilization (always >=1, which occurs when $CO_2$ reaches infinity) and $s_{CO_2}^{sig}$ is the $CO_2$ sensitivity of NPP in the sigmoidal formulation.

We allow for a linear combination of $CO_2$ fertilization formulations. A method factor ($m_{CO_2}$) ranging from 0 to 2 is used to combine the formulations and calculate the effective $CO_2$ feedback (illustrated in Fig. A1).

$$\epsilon_{CO_2} = \begin{cases} \left(1 - m_{CO_2}\right) \times \epsilon_{CO_2}^{log} + m_{CO_2} \times \epsilon_{CO_2}^{rect} & 0 \leq m_{CO_2} \leq 1 \\ \left(2 - m_{CO_2}\right) \times \epsilon_{CO_2}^{rect} + \left(m_{CO_2} - 1\right) \times \epsilon_{CO_2}^{sig} & 1 < m_{CO_2} \leq 2 \end{cases} \tag{17}$$

### 2.3.2 Feedback from temperature change

Global-mean 'temperature change' (dT) is taken as a proxy for climate-related impacts on the carbon cycle fluxes, i.e. for representing the carbon-climate feedback. The response of NPP to dT is assumed to follow an exponential or sigmoidal scaling of $NPP_0$, determined by a given temperature sensitivity ($s_{dT(NPP)}^{exp}$ or $s_{dT(NPP)}^{sig}$). The latter is introduced to better capture the trend of NPP in low emission scenarios.

$$\epsilon_{dT(NPP)}^{exp} = e^{s_{dT(NPP)}^{exp} \times dT} \tag{18}$$

$$\epsilon_{dT(NPP)}^{sig} = \frac{2}{1 + e^{-s_{dT(NPP)}^{sig} \times dT}} \tag{19}$$

Similarly, a method factor ($m_{dT}$, between 0 and 1) is used to control the effective temperature change feedback on NPP (illustrated in Fig. A1).

$$\epsilon_{dT(NPP)} = (1 - m_{dT}) \times \epsilon_{dT(NPP)}^{exp} + m_{dT} \times \epsilon_{dT(NPP)}^{sig} \tag{20}$$

### 2.4 The carbon-nitrogen coupling in CNit

The current carbon-nitrogen coupling is based on the mineral nitrogen requirement and availability, from which the nitrogen deficiency (or surplus) can be calculated and, subsequently, the influence on NPP. A direct link between NPP (or photosynthesis) and plant nitrogen status is a common treatment in complex carbon-nitrogen coupled models (Zaehle and Dalmonech, 2011; Zaehle et al., 2014).

The overall formulation design of the carbon-nitrogen coupling effect on NPP is as follows: First, we establish a relationship between NPP (carbon fixation) and 'nitrogen plant uptake' (PU, nitrogen fixation). We calculate the 'potential NPP'

(NPP$_{pot}$), i.e., the carbon-only NPP) by setting $\epsilon_{CN(NPP)} = 1$ in Eq. 10. The corresponding 'PU requirement' (PU$_{req}$) is determined by this NPP$_{pot}$ (as well as a temperature effect, see details in Eqs. 21-22). Then, the nitrogen deficiency or surplus is calculated based on PU$_{req}$ and 'nitrogen atmospheric deposition' (AD), with which the $\epsilon_{CN(NPP)}$ is updated and the 'actual NPP' (NPP$_{act}$) is determined. The following describes the process in detail.

### 2.4.1 The 'nitrogen plant uptake' (PU) and NPP

First, we assume that the PU is a function of the NPP (whether potential or actual) and scaled by the temperature response of PU ($\epsilon_{dT(PU)}$):

$$PU = PU_{max} \times e^{-\frac{NPP_{ref}}{NPP}} \times \epsilon_{dT(PU)} \tag{21}$$

$$\epsilon_{dT(PU)} = e^{s_{dT(PU)} \times dT} \tag{22}$$

where $NPP_{ref}$ is a reference NPP used to normalize the real time NPP (further explanation in the next paragraph), $PU_{max}$ sets the upper bound of the PU without the temperature feedback, and $s_{dT(PU)}$ is the temperature sensitivity of PU.

When NPP = $NPP_{ref}$, with $dT = 0$, the PU is fixed to $PU_{max}/e$. The $PU_{max}$ and $NPP_{ref}$ parameter pair defines a unique plant carbon-nitrogen assimilation system. Specifically, $PU_{max}/(e \times NPP_{ref})$ reflects a default setting of the plant nitrogen:carbon ratio. It is also worth reiterating that $NPP_{ref}$ is not necessarily the same as the initial NPP ($NPP_0$ in Eq. 10).

The formulation presented in Eq. 21 suggests an increasing PU with increasing NPP. However, as NPP increases, the PU needed per additional unit NPP gradually decreases. This provides a way for the model to represent a declining carbon:nitrogen ratio in plant biomass with increasing NPP due to, for example, $CO_2$ fertilization.

### 2.4.2 The 'nitrogen plant uptake requirement' (PU$_{req}$)

When the nitrogen effect is not considered, NPP can reach NPP$_{pot}$, which can be calculated using Eq. 10 (fixing $\epsilon_{CN(NPP)} = 1$). The PU$_{req}$ can be calculated based on Eq. 21:

$$PU_{req} = PU_{max} \times e^{-\frac{NPP_{ref}}{NPP_{pot}}} \times \epsilon_{dT(PU)} \tag{23}$$

The integration of the PU$_{req}$ over a certain time period (e.g., the model time step) indicates the amount of mineral nitrogen needed to realize the potential NPP.

### 2.4.3 The nitrogen effect on NPP

The mineral nitrogen availability depends on the current mineral nitrogen pool size and fluxes (Eq. 8). The 'net mineralization' (netMIN), which is the largest natural source of mineral nitrogen, comes from litter and soil nitrogen

turnovers. The turnovers are dependent on the pool sizes based on a first-order decay formulation (see Section 2.6). Considering that PU is the predominant influx for the organic nitrogen pools (5 to >10 times greater than biological nitrogen fixation in the complex models we have examined), the accumulation of nitrogen in plant, litter, and soil pools is effectively controlled by PU. In other words, PU and netMIN, the two major fluxes channeling through the organic and inorganic nitrogen pools by either consuming mineral nitrogen to enrich organic nitrogen or vice versa, are closely intertwined and mutually influence each other. This is supported by the results from complex models (CABLE, OCN, and multiple CMIP6 models) that show a similar value and trend for PU and netMIN at the global-mean, annual-mean level (Fig. A2).

When approximating netMIN as being linearly correlated with PU, the unmet $PU_{req}$ from netMIN alone can be calculated as a linear function of the $PU_{req}$ itself [i.e., netMIN - $PU_{req}$ = f($PU_{req}$)]. Considering that 'nitrogen atmospheric deposition' (AD) also supports the $PU_{req}$, its effect of alleviating the nitrogen deficiency is added by another linear function. The carbon-nitrogen coupling effect on NPP ($\epsilon_{CN(NPP)}$) is then determined by:

$$\epsilon_{CN(NPP)} = \epsilon_{CN(NPP)0} \times e^{f_1 \times AD - f_2 \times PU_{req}} \tag{24}$$

Where $\epsilon_{CN(NPP)0}$ is a base nitrogen effect on NPP when there is neither deficiency nor surplus. $f_1$ and $f_2$ are fitted parameters whose values are always positive. The ($f_1 \times AD - f_2 \times PU_{req}$) term determines the relative strength of the current AD and unmet $PU_{req}$ (i.e., nitrogen deficiency/surplus).

This formulation is transformed from complex models with the key idea of comparing mineral nitrogen availability and plant nitrogen requirement. In complex carbon-nitrogen models, the nitrogen availability is typically based on the current mineral nitrogen pool size (with mass unit) and the nitrogen requirement is computed from the integrated fluxes in a given time step (with mass unit) (Thornton et al., 2007; Wiltshire et al., 2021; Zaehle et al., 2014). The competition from microbial immobilization is also considered in some complex models. However, in a model with a much longer time step (e.g., annually) like ours, such a system would be inherently unstable since the mineral nitrogen pool size would be orders of magnitude smaller than the annual nitrogen demand (i.e., the system would be unstable because the turnover of the mineral nitrogen pool would be substantially smaller than the timestep).

Fixing $f_1$ = 1 indicates that on the timescale of interest (e.g., annually), all mineral nitrogen from AD is 100% available for the plant. However, this is not necessarily correct considering that 1) the process-level nitrogen limitation/fertilization does not remain constant over the timescale of interest (e.g., annually); 2) the mineral nitrogen accumulation from previous time steps can be used for the current time step; and 3) the direct mineral nitrogen loss, whose magnitude is determined by the mineral pool turnover time, may counterbalance the effect of nitrogen deposition on the fertilization. Therefore, giving freedom to both the $f_1$ and $f_2$ parameters implicitly allows our formulation to consider the above effects.

Note that both $f_1$ and $f_2$ parameters are calibrated, the $\epsilon_{CN(NPP)}$ does not necessarily have to be $\epsilon_{CN(NPP)0}$ at the start of the experiments, which gives flexibility to the model to determine the carbon-nitrogen coupling effect in the pre-industrial condition.

### 2.4.3 The actual NPP and PU

After the nitrogen effect is calculated, the 'actual NPP' (NPP$_{act}$, i.e., NPP with the carbon-nitrogen coupling effect) is determined by:

$$NPP_{act} = NPP_{pot} \times \epsilon_{CN(NPP)} \tag{25}$$

And based on Eq. 21, the corresponding 'actual PU' (PU$_{act}$) becomes:

$$PU_{act} = PU_{max} \times e^{-\frac{NPP_{ref}}{NPP_{act}}} \times \epsilon_{dT(PU)} \tag{26}$$

The modelled NPP$_{act}$ and PU$_{act}$ are then used for the differential equations (Eq. 1-9) to solve the carbon and nitrogen states.

### 2.5 The litter production respiration flux

MAGICC separately simulates a 'litter production respiration' flux (LPR, the fast litter respiration produced from the plant litter production that does not carry over into the subsequent year i.e. that returns to the atmosphere on sub-annual timescales) by scaling an initial plant litter production respiration flux ($LPR_0$) with the $CO_2$ fertilization effect ($\epsilon_{CO_2}$), the carbon-nitrogen coupling effect on NPP ($\epsilon_{CN(NPP)}$), the climate effect ($\epsilon_{dT(LPR)}$), and the land use change effect ($\epsilon_{LU}$).

$$LPR = LPR_0 \times \epsilon_{CO_2} \times \epsilon_{dT(LPR)} \times \epsilon_{CN(NPP)} \times \epsilon_{LU} \tag{27}$$

$$\epsilon_{dT(LPR)} = e^{s_{dT(LPR)} \times dT} \tag{28}$$

The plant litter production respiration flux is assumed to be a fast over-turning (e.g., within one-year) outflux from the plant carbon pool that circulates through the NPP to plant to litter to the atmosphere within a single timestep. Considering the close relationship between plant litter production respiration flux and NPP, it is scaled by the same CO2 and carbon-nitrogen coupling feedback as NPP, as well as the same land use effect. Given that LPR results from three cascading processes - NPP, litter production, and decomposition - it is modeled with an exponential temperature response based on its own temperature sensitivity ($s_{dT(LPR)}$).

### 2.6 The turnover of carbon and nitrogen pools

The 'litter production' (LP), 'litter decomposition' (LD), and 'soil respiration' (SR) for plant, litter, and soil carbon/nitrogen pools are assumed to be proportional to the corresponding pool sizes, linked by the turnover time ($\tau$) and scaled by the effect from temperature change ($\epsilon_{dT}$) and carbon-nitrogen coupling ($\epsilon_{CN}$).

$$turnover = \frac{poolsize_i}{\tau_i} \times \epsilon_{dT} \times \epsilon_{CN} \tag{29}$$

where *i* refers to the plant, litter, and soil carbon/nitrogen pools.

For the temperature feedback, it is assumed that each process has its own temperature sensitivity. The feedback is then modelled by an exponential relationship.

$$\epsilon_{dT(i)} = e^{s_{dT(i)} \times dT} \tag{30}$$

where *i* refers to the carbon or nitrogen turnover process LP, LD, or SR.

The carbon-nitrogen coupling feedback takes current 'nitrogen plant uptake' (PU) and 'nitrogen atmospheric deposition' (AD) as proxies to represent the plant nitrogen status and the nitrogen forcing, respectively. And similarly, each turnover process has its own response to the current PU ($s_{PU(i)}$) and AD ($s_{AD(i)}$). An exponential relationship is used to simulate their effects on the processes.

$$\epsilon_{CN(i)} = e^{s_{PU(i)} \times PU} \times e^{s_{AD(i)} \times AD} \tag{31}$$

where *i* refers to the carbon or nitrogen turnover processes LP, LD, or SR.

The 'mineral nitrogen loss' (LS) is the turnover flux for the mineral nitrogen pool, which follows a first-order decay (Eq. 29). However, only a temperature effect is applied to LS since the current AD and PU, the two proxies for the carbon-nitrogen feedback, are already directly the influx and outflux for the mineral pool (Fig. 1).

## 2.7 The updated implementation of land use emissions and their impact on NPP

Most scenarios run in MAGICC, for example those in AR6 [see Chapter 7 in Climate Change 2022: Mitigation of Climate Change (Riahi et al., 2022)], directly report net land use emissions (gross deforestation - regrowth) instead of the separation of those two parts. Considering a box model with fixed turnover times, when the input is constant (e.g., constant NPP), a one-off subtraction of carbon out of one of the pools (e.g., a one-off gross deforestation) would not lead to a permanent reduction in the amount of carbon in this pool. Instead, it would yield an asymptotic rebound to the pre-intervention carbon pool size over time (as the carbon that was moved out of the pool is taken up by the land again), which implies full regrowth in the long term. In MAGICC's previous carbon cycle, the partial regrowth was accounted for using a factor that determines the fraction of expected regrowth and adjustments to the turnover times (Meinshausen et al., 2011a). This implementation blends the regrowth with changes in the pool's turnover fluxes. As the turnovers are also impacted by feedbacks (Eq. 29), the previous MAGICC carbon cycle setup required a parallel calculation of the non-feedback case to retrieve the regrowth flux and correctly handle the land-use input.

### 2.7.1 The updated land use emission implementation

The updated land use emission implementation described here makes the regrowth and gross deforestation more straightforward. The LU input in Eqs. 1-4 refers to the net land use emissions, which can be written as:

$$LU_c = LU_{grsd} - LU_{rgr} \tag{32}$$

where $LU_{grsd}$ and $LU_{rgr}$ refer to the gross deforestation (instantaneous biomass extraction from land organic carbon pool) and regrowth (legacy biomass addition to land organic carbon pool), respectively.

The regrowth formulation assumes a constant regrowth flux during the growing years following each instance of gross deforestation. Therefore, the regrowth flux is calculated as the total regrowth (part of the gross deforestation) divided by the regrowth time, formulated as follows:

$$LU_{rgr} = \frac{\varphi \int_{max(0,t-\tau_{rgr})}^{t} LU_{grsd}}{\tau_{rgr}} \tag{33}$$

where $\varphi$ refers to the fraction of gross deforestation that can regrow; $\tau_{rgr}$ denotes the regrowth time required to reach the partial regrowth.

The formulation assumes a constant regrowth flux ($LU_{rgr}$) for $\tau_{rgr}$ years after every single gross deforestation - the regrowth flux at time $t$ is thus the potential total regrowth ($\varphi$ times the integration of $LU_{grsd}$) divided by the required regrowth time ($\tau_{rgr}$).

Note that, in this formulation for regrowth and gross deforestation, both fluxes exist only within a certain time period ($\tau_{rgr}$), hence they cannot change the long-term equilibrium of the system because the fixed turnover times always return the system to its pre-deforestation state (Eqs. 1-4). We discuss our solution for this in the next section.

### 2.7.2 The effect of land use emission on NPP

Instead of adjusting the turnover times (as in previous versions of MAGICC carbon cycle), CNit applies the effect of land use change to NPP to change the long-term equilibrium after the gross deforestation ($\epsilon_{LU}$ in Eq. 10), which is formulated as:

$$\epsilon_{LU} = \frac{C_{LAND_0} - (1 - \varphi) \int_{0}^{t} LU_{grsd}}{C_{LAND_0}} \tag{34}$$

where $C_{LAND_0}$ is the initial land carbon pool size.

The land use effect on NPP assumes that the NPP is reduced immediately after gross deforestation activities. Considering the simultaneous regrowth, the NPP reduction is proportional to the accumulated permanent gross deforestation [ $(1 -$

$\varphi) \int_0^t LU_{grsd}$]. With the land use effect on NPP, when there is a one-off gross deforestation, the NPP gradually decreases to a new value, with that value depending on the regrowth fraction $\varphi$ (e.g., $\varphi = 0$ refers to zero-regrowth and the gross deforestation causes permanent carbon loss from the land carbon pools). Because of the NPP change, the long-term equilibrium of the system changes too.

The new formulation enables the separate calculation of regrowth and gross deforestation without the non-feedback run. Arguably, it also makes more physical sense, because deforestation reduces the amount of forest (area) available to grow and hence the amount of NPP. The regrowth amount is a linear function of regrowth time, representing a simplified approach compared to sigmoidal growth models. This is a trade-off that could be re-considered in future work. The applied effect on NPP also means an additional constraint on MAGICC's simulation of regrowth and gross deforestation. However, it should be noted that the definition discrepancy of land use change emissions itself still exists among different models/approaches and leads to substantial differences in land-use emission estimates (Gasser and Ciais, 2013; Stocker and Joos, 2015; Grassi et al., 2023). Thus, the LU input must align with the outputs of complex models, the scenarios being applied, and specific definitions of land-use change emissions (e.g., ELUC in the Global Carbon Budget project) to ensure consistency in model applications.

### 2.7.3 The limitation of land use emission in CNit

The formulation presented here has certain limitations. For example, it does not account for variations in regrowth rates among different ecosystems or across ecosystem successional stages - an inherent constraint of the global box model approach. Additionally, it aggregates deforestation and harvest fluxes into a single LU input, although the regrowth fraction parameter may provide some indication of harvest activities that do not result in regrowth.

It should also be noted that MAGICC, whether using the previous carbon cycle model or the CNit version, does not simulate carbon or nitrogen storage in the product pool due to its relatively small size (Eqs. 1–9 and Fig. 1). Thus, the deforestation and harvest fluxes going into the product pool, expected to be small, are accounted for in the land use emission input ($LU_c$ or $LU_n$) and associated regrowth parameter, while the land carbon (or nitrogen) pool does not include the correspondingly stored carbon (or nitrogen) within the product.

### 3 Model calibration

### 3.1 Overview of the calibration process and results

This section presents the offline calibration results for CNit, i.e. calibrations using prescribed land surface temperature and atmospheric $CO_2$ concentration from the original model outputs. We first describe the data acquisition and post-processing of land surface model outputs and CMIP6 ESM outputs (Section 3.2). Next, we define the calibration targets (major fluxes and

pool sizes) and weight them to create a cost function. Finally, we apply optimization algorithms to identify the "best-estimate" parameter set (Section 3.3). For a single model, all experiments are calibrated simultaneously, resulting in one "best- estimate" parameter set that captures the model's behavior across experiments. Using these "best-estimate" parameter values, we evaluate CNit emulation against model outputs and calculate the 'root mean squared error' (RMSE) and normalized RMSE to assess emulation performance. The discussion of the calibration results for CABLE, OCN, CMIP6 ESMs are provided in Section 3.4 and Section 3.5.

## 3.2 Data acquisition and processing

### 3.2.1 Land surface models and outputs

The CABLE and OCN land surface model output datasets (global-, annual-mean values) were obtained directly from the modeling groups. CABLE is the land surface component for the Australian Community Climate and Earth System Simulator (ACCESS-ESM1) (Law et al., 2017) and ACCESS-CM2 (Bi et al., 2020). OCN is the updated land surface model built on ORCHIDEE (Zaehle and Friend, 2010), the land surface component of the IPSL-CM climate model (Boucher et al., 2020). Both CABLE and OCN provided the results from the carbon-only and carbon-nitrogen coupled setups for the representative concentration pathway 8.5 (RCP85) scenario (Fleischer et al., 2019), with OCN also providing the results for the RCP26 scenario (Meyerholt et al., 2020). The data from both the carbon-only and carbon-nitrogen coupled setups of the land surface models offers valuable information to constrain nitrogen interactions separately from the climate and $CO_2$ effects. The carbon-nitrogen coupled runs in CABLE consisted of two experiments with constant and dynamic atmospheric deposition inputs, respectively (Fleischer et al., 2019). These experiments are useful for diagnosing the standalone effect of atmospheric deposition. The climate data and atmosphere $CO_2$ concentration for the land surface model experiments were derived either from their corresponding ESM outputs (Meyerholt et al., 2020) or the RCP greenhouse gas concentrations (Meinshausen et al., 2011b). One OCN model structure with flexible carbon:nitrogen ratio, linear biological nitrogen fixation and actual evaporation relationship, and explicit mineral nitrogen loss representation - namely the FLX/FOR/NL1 structure - from the 30 ensemble model structures in the original paper (Meyerholt et al., 2020) was selected for the CNit calibration in this paper. The selected OCN model structure serves as a proof-of-concept for the proposed CNit model. Future work will explore alternative structures and address structural uncertainties.

### 3.2.1 CMIP6 ESMs and outputs

We select target ESMs based on the following criteria: 1) The model includes a terrestrial nitrogen cycle; 2) The outputs encompass the majority of carbon-nitrogen cycle variables needed for C4MIP; and 3) The model has completed all selected experiments, including one idealized experiment (1pctCO2), historical simulations (historical), and four Shared Socioeconomic Pathways (SSPs) scenarios (SSP126, SSP245, SSP370, and SSP585), all of which are CMIP6 tier 1

concentration-driven experiments. Based on the data availability and completeness (in Earth System Grid Federation), six ESMs were chosen for this calibration (Table 1).

**Table 1. The list of Earth System Models (ESMs) selected for the calibration**

| ESMs | Variant_label | Land component | Reference |
|---|---|---|---|
| CMCC-CM2-SR5 | r1i1p1f1 | CLM4.5 | (Cherchi et al., 2019) |
| CMCC-ESM2 | r1i1p1f1 | CLM4.5, BGC configuration | (Lovato et al., 2022) |
| MPI-ESM1-2-LR | r1i1p1f1 | JSBACH3.2 | (Mauritsen et al., 2019) |
| NorESM2-LM | r1i1p1f1 | CLM5 | (Seland et al., 2020) |
| UKESM1-0-LL | r1i1p1f2 | JULES-ES-1.0 | (Sellar et al., 2019) |
| MIROC-ES2L | r1i1p1f2 | VISIT-e | (Hajima et al., 2020) |

The monthly gridded CMIP6 ESM outputs were downloaded from Earth System Grid Federation (ESGF, https://esgf-node.llnl.gov/projects/cmip6/, last access, January 18, 2024) and processed into global-mean, annual-mean values. Our grid-to-global aggregation used the model-specific grid area (areacella) and land fraction (sftlf) to avoid issues with the resolution variation across different model outputs. Our monthly-to-annual aggregation first concatenated the original monthly outputs along the time dimension and then calculated the annual values, weighted by the number of days in each month as defined by 410 each ESM's output calendar. It should be noted that, even though the selected ESMs and experiments provided relatively complete outputs for the pools and fluxes, none of them reported all the required fluxes by C4MIP, especially those related to the land use and anthropogenic perturbations (Tang et al., 2024a). As part of the processing, it was not trivial to reproduce the models' mass balance based on the reported outputs (Tang et al., 2024a). For this paper, we have applied a workaround, which we describe in the next section.

**3.3 Calibration setup**

**3.3.1 Temperature profile and $CO_2$ concentration**

For CABLE and OCN, temperature and $CO_2$ concentration data were obtained directly from the original providers (Fig. A3). For CMIP6 ESMs, temperature data (tas) was sourced from model outputs. CNit uses 'temperature change' (dT) as a proxy for climate-related impacts, calculated as the difference between the current temperature and the initial year's temperature 420 (Fig. A3 and Fig. A4). $CO_2$ concentrations were taken from the CMIP6 forcing datasets (Meinshausen et al., 2017; Meinshausen et al., 2020) – as all calibrated experiments are concentration-driven.

### 3.3.2 Inputs for CNit

The inputs for CNit (Fig. 1) include nitrogen atmospheric deposition, biological nitrogen fixation, and fertilizer application fluxes, all directly available from model outputs. For land-use emissions of carbon and nitrogen, CABLE and OCN provide readily available data. However, CMIP6 ESMs do not provide such data (Tang et al., 2024a). To address this, we calculated land-use carbon emission ($LU_c$ in Fig. 1 and Eqs. 1-4) as "NPP - heterotrophic respiration - dcLand/dt", ensuring carbon mass conservation [as there is no straightforward land use emission from the reported outputs alone, see details in Tang et al., 2024a)].

Where available, land-use and anthropogenic disturbance-related nitrogen fluxes (fNAnthDisturb and fNProduct in the CMIP6 Data Request) were combined to create the land-use nitrogen input for CNit ($LU_n$ in Fig. 1 and Eqs. 5-9).

### 3.3.3 Calibration target and optimization

Calibration targets for both land surface models and CMIP6 ESMs included NPP, heterotrophic respiration, 'nitrogen plant uptake' (PU), and all carbon and nitrogen pool sizes. The cost function was calculated as the sum of normalized errors for each target flux or pool size timeseries [i.e., square (emulation - target) / ($target_{max}$ - $target_{min}$)]. This normalization accounted for the differing magnitudes among target variables and gives lower weight to variables with greater variability. All available experiments were calibrated simultaneously without additional weighting, meaning the final cost was calculated as the sum of the costs from all experiments.

For CMIP6 ESMs, historical period data and SSP scenario results were combined (referred to as hist_SSP) into a unified time axis spanning 1850-2100 or 1850-2300, depending on data availability.

The calibration process employed the differential evolution algorithm (Storn and Price, 1997) for global optimization and the Nelder-Mead algorithm (Gao and Han, 2012) for local minimization. First, differential evolution ran for 30,000 iterations with ten random initializations. Next, the resulting ten global-minimum parameter sets were used as initial guesses for the Nelder-Mead algorithm, yielding respective local minimums. Finally, the parameter set with the lowest cost was selected as the "best-estimate" parameter set. A full parameter list and the "best-estimate" values are provided in Table A1 and Table A2 in the Appendix.

### 3.4 Calibrating CNit to CABLE and OCN: Results and comparison

CNit has successfully emulated the fluxes and pool sizes from the CABLE and OCN experiments (Fig. 2 and Fig. A5), with the RMSE (or normalized RMSE) ranging from 0.9-2.6 GtC/yr (1.3-2.2%) for NPP, 4.0-32.9 GtC (0.2-1.0%) for land carbon pool size, 0.016-0.020 GtN/yr (1.1-1.9%) for PU, 0.05-0.11 GtN (<0.1%) for land organic nitrogen pool size, and 0.0022-0.026 GtN (2.3-6.2%) for the mineral nitrogen pool size, respectively. There is a strong and increasing nitrogen limitation on NPP in CABLE, inhibiting NPP by 9.4% to 48.2% (constant atmospheric deposition) or 46.2% (dynamic atmospheric

deposition) from 1901 to 2100. The total land carbon storage, i.e. the total amount of carbon taken up by the land carbon cycle, by 2100 under the RCP85 scenario in the constant and dynamic atmospheric deposition experiments is 353 GtC and 398 GtC, respectively, which is significantly lower than the 1634 GtC land carbon storage in the runs that use a carbon cycle without nitrogen limitation. The large reduction in NPP (influx of the system, Fig. 2) due to the carbon-nitrogen coupling, along with the relatively smaller reduction in heterotrophic respiration (major outflux of the system, Fig. A5), result in the decrease of net land carbon flux (Fig. 2) and thus, the decreased land carbon storage (Fig. 2).

The OCN model exhibits significantly less nitrogen limitation on NPP (<5.0% inhibition) for both the RCP85 and RCP26 scenarios. This disparity may be attributed to the fact that the NPP in OCN in the carbon-only simulations (~85 GtC/yr in 2100 in RCP85) is notably lower than that in CABLE (~140 GtC/yr in 2100 in RCP85). The carbon cycle in OCN has also experienced a higher temperature change (~8°C from 1850 to 2100 in RCP85, Fig. A3) compared to CABLE (~5.5°C from 1900 to 2100 in RCP85, Fig. A3). The emulation captures the varied NPP increases by applying a stronger $CO_2$ fertilization effect and a more negative climate feedback in CABLE than in OCN. These respective changes are captured in CNit by the logarithmic carbon-concentration feedback formulation with a high sensitivity ($s_{CO_2}^{log} = 2.582$) and negative $s_{dT(NPP)}^{exp}$ and $s_{dT(NPP)}^{sig}$ parameters (Table A2). Nevertheless, the relatively minor NPP limitation in OCN, along with the resulting decrease in net land carbon flux (Fig. 2), leads to a reduction of 45 GtC (or 10%, RCP85) and 26 GtC (or 8%, RCP26) in land carbon storage by the end of 2100 compared to their respective simulations that use a carbon cycle without a nitrogen limitation.

In the RCP85 experiment, CABLE and OCN show similar relative changes in PU over time; however, their starting PU fluxes differ (~0.6-1.0 GtN/yr in CABLE vs ~1.0-1.4 GtN/yr in OCN). The emulation is able to capture both dynamics by adjusting the 'maximum PU' ($PU_{max}$) and 'temperature sensitivity of PU' ($s_{dT(PU)}$) parameters. Specifically, the different starting PU values stem from the higher $PU_{max}$ in OCN (2.4 GtN/yr) than in CABLE (1.9 GtN/yr). The similar trends occur because of the higher $s_{dT(PU)}$ in CABLE compared to OCN (0.014 K$^{-1}$ vs 0.008 K$^{-1}$, Table A2), which compensates for the effect of its lower $PU_{max}$ (Eqs. 21-22). In the dynamic atmospheric deposition experiment, the PU in CABLE is slightly higher than in the constant atmospheric deposition experiment, primarily accumulating in the soil organic nitrogen pool (+3 GtN in 2100 compared to the constant atmospheric deposition scenario, Fig. A5). The accumulated organic nitrogen enriches the mineral nitrogen pool instead of facilitating the nitrogen loss, as the system remains nitrogen limited. In contrast, in the RCP85 experiment from OCN, the new organic nitrogen is mainly stored in the plant pool (Fig. A5). The trend and magnitude of the mineral nitrogen pool size in OCN are also largely different from those in CABLE. The emulation captures the diverse mineral nitrogen trends by adapting the temperature response of mineral nitrogen loss for the two models (weak and negative $s_{dT(LS)}$ of -0.007 K$^{-1}$ in CABLE vs strong and positive $s_{dT(LS)}$ of 0.088 K$^{-1}$ for OCN, Table A2). The order-of-magnitude difference in mineral nitrogen pool sizes demonstrates the huge uncertainty in estimated mineral nitrogen quantities.

Comparing the behavior of the nitrogen cycle in OCN's RCP85 and RCP26 experiments, it is found that PU follows the trend of NPP, supporting our assumption that PU can be modeled based on NPP (Eq. 21). However, unlike in the RCP85 scenario, the new nitrogen introduced by PU is mainly accumulated in the soil nitrogen pool in the RCP26 scenario (Fig. A5). Based on the formulation, the larger soil nitrogen pool size implies more nitrogen mineralization (the first-order turnover, Eq. 29), enabling the emulation to capture the increasing trend of mineral nitrogen pool size from 2050 to 2100 (Fig. A5). The organic nitrogen accumulation in the RCP85 and RCP26 experiments does not precisely follow their corresponding carbon storage trends, where the new carbon introduced by NPP is predominantly stored in the plant carbon pool in both scenarios. This difference indicates that the carbon cycle and nitrogen cycle, even though closely intertwined, may react in a divergent manner to climate change.

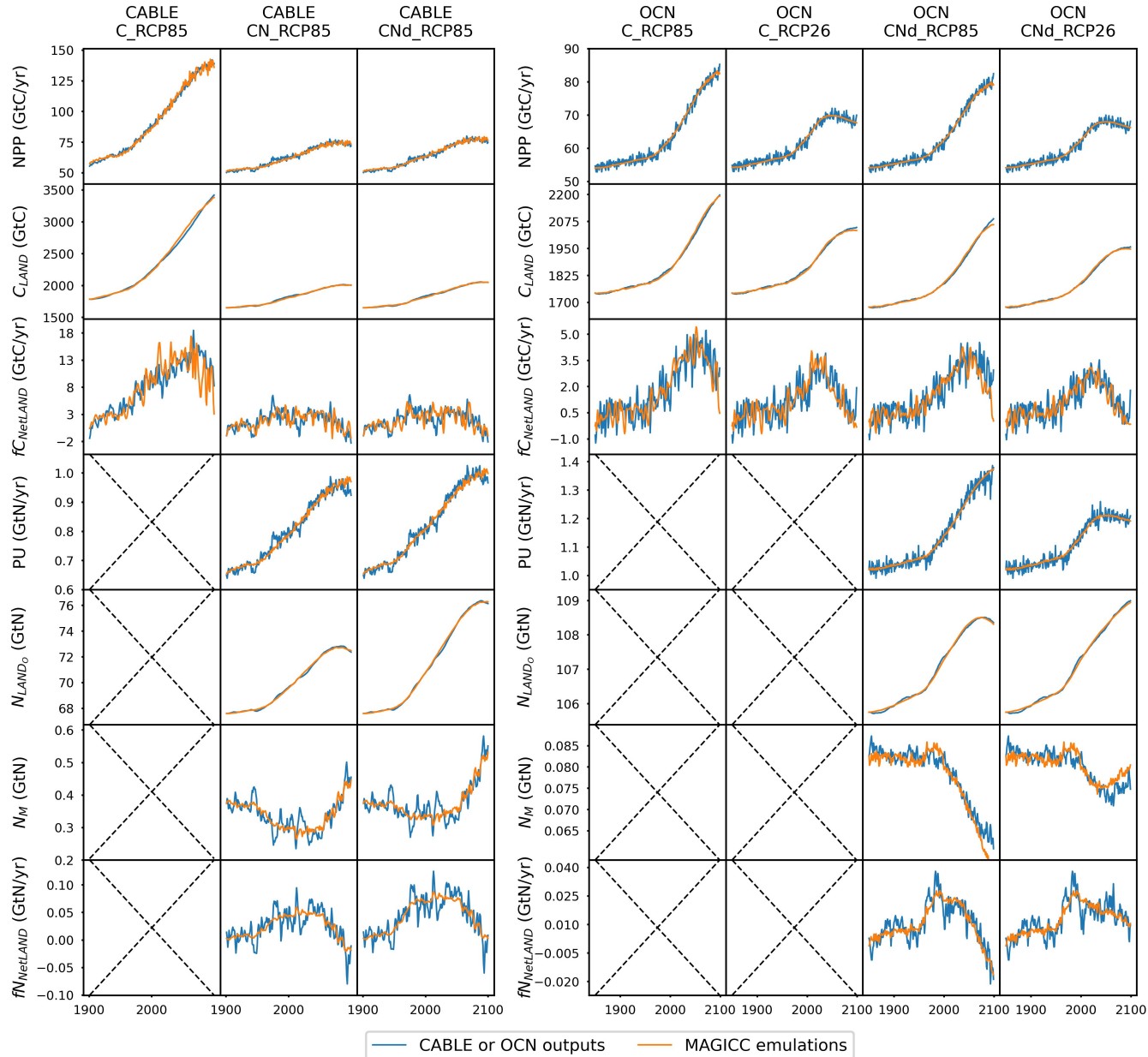

**Figure 2. Comparison of net primary production (NPP), land carbon pool size (C$_{LAND}$), net land carbon flux (fC$_{NetLAND}$), nitrogen plant uptake (PU), land organic nitrogen pool size (N$_{LANDo}$, sum of nitrogen in plant, litter, and soil pools), mineral nitrogen pool size (N$_M$), and net land nitrogen flux (fN$_{NetLAND}$) between CABLE or OCN outputs (blue lines) and CNit emulations (orange lines). The experiments labeled as C, CN, and CNd denote the carbon-only, carbon-nitrogen coupled with constant nitrogen atmospheric deposition, and carbon-nitrogen coupled with dynamic nitrogen atmospheric deposition configurations in the land surface models, respectively.**

## 3.5 Calibrating CNit to CMIP6 ESMs: Results and comparison

CNit emulation captures the dynamics of major fluxes and pool sizes for all the CMIP6 ESMs and experiments (Fig. 3 and Fig. A6), despite the diverse climate and $CO_2$ forcings (Fig. A4). More detailed comparisons of ESM outputs and CNit emulations are provided in Fig. A6. Overall, the model is able to emulate the wide range of ESM behaviour. In the rest of this section, we focus on cases where this is not the case.

In the 1pctCO2 experiment, the emulated NPP is lower than that from the UKESM1-0-LL and MIROC-ES2L outputs. The underestimation is mainly from the inconsistent behavior of these two ESMs in the idealized 1pctCO2 experiment and hist_SSP experiments. Both ESMs have simulated higher NPP at the end of their 1pctCO2 runs (~100 GtC/yr in 1999 for both ESMs) than that at the end of their SSP runs (e.g., SSP126, <80 GtC/yr in 2100 for both ESMs), which is contradictory to their PU results (lower in 1pctCO2 and higher in SSPs, Fig. A6). Such behavior is in direct contradiction with our assumption that higher NPP requires a higher PU (section 2.4, Eq. 21). Since NPP and PU are both set as calibration targets, CNit has tried to minimize the gap between the emulated fluxes and targets, resulting in the simultaneous underestimation of NPP and overestimation of PU for UKESM1-0-LL and MIROC-ES2L (Fig. 3 and Fig. A6). However, such different behavior is only observed in the 1pctCO2 experiments for these two ESMs, indicating there could be either some model response nonlinearities between their 1pctCO2 and SSP runs that our model is not capturing or some regionally distinct effects that we are not seeing in the global, annual averages.

The underestimated NPP in the 1pctCO2 experiment has led to an underestimation of ~190 GtC of the land carbon pool size for MIROC-ES2L (Fig. A6). But the underestimation of emulated land carbon in UKESM1-0-LL is much narrower (~78 GtC, Fig. A6). This is because the CNit emulation has overestimated the soil carbon storage at the later phase of the 1pctCO2 experiment, which compensates for some carbon loss by the underestimated NPP. The emulated land carbon pool sizes (specifically the plant carbon pool sizes) are slightly but systematically smaller than the outputs from CMCC-CM2-SR5 and CMCC-ESM2 in their 1pctCO2 runs (Fig. A6). Such results primarily stem from CNit's underestimation of NPP during the middle of the 1pctCO2 experiment (Fig. A6). The CNit emulation of pool turnovers also contributes to the inconsistency between the emulation results and ESM outputs. The first-order decay of carbon pool turnover (Eq. 29) is not particularly effective in modeling the minor changes in pool sizes relative to the substantial initial pool size. For example, both CMCC models exhibit a soil carbon loss of approximately 45 GtC in their 1pctCO2 runs, with an initial soil pool size of around 2870 GtC. Consequently, the calibration has prioritized achieving better fit for soil pool sizes at the expense of accurately representing plant and litter pool sizes. The different starting soil pool sizes for ESMs' 1pctCO2 and historical simulations further complicate the soil carbon turnover emulation (Fig. A6). These different initial pool sizes in CMIP6 data do not make much sense. However, without explicit reasons to discredit the data, we have not adjusted our model to accommodate such oddities; rather, they compromise the model's fit.

The net land carbon flux is a highly variable flux (<10 GtC/yr for all the ESMs, Fig. A6). CNit emulation of the ESMs has both positive and negative errors (Fig. 3). The CNit emulation has captured both their trends and magnitudes (Fig. A6).

The new nitrogen cycle in MAGICC has demonstrated the ability to capture the dynamics of nitrogen fluxes and pools, especially the PU and land organic nitrogen pool, in the ESM's SSP scenario runs (Fig. 3). In the 1pctCO2 experiment, however, the emulated PU is overestimated for UKESM1-0-LL and MIROC-ES2L, which is accompanied by the underestimation of their NPP, in an attempt to compensate for the conflicting high NPP and low PU in the ESMs' outputs (Fig. A6). The emulated organic nitrogen pool size is overestimated for NorESM2-LM (Fig. A6), which is primarily attributed to the overestimated PU and soil nitrogen pool sizes (Fig. A6).

The mineral nitrogen pool sizes exhibit the most diverse results between the CNit emulation and the ESM outputs (Fig. 3, details in Fig. A6). Firstly, the mineral nitrogen pool sizes in UKESM1-0-LL are relatively well-emulated. Secondly, the emulated mineral nitrogen pool sizes show significant differences compared to NorESM2-LM outputs, both in magnitudes and trends. Considering that both the PU and the organic nitrogen pool size in NorESM2-LM are relatively well-emulated (Fig. A6), such differences in mineral nitrogen pool size could be attributed to either the input nitrogen fertilizer application flux or the turnover flux of the mineral pool. The CNit emulation of mineral nitrogen turnover is a first order decay with an exponential temperature effect scaling (Eq. 29). Thus, the initial pool size and turnover time determine the magnitude of the nitrogen loss flux. Among all the studied CMIP6 ESMs, NorESM2-LM has the largest initial mineral nitrogen pool (5.64 GtN vs <1.50 GtN for all the other ESMs), which can lead to a large turnover flux. Conversely, the pool size change of mineral nitrogen is not significant in NorESM2-LM (0.78 GtN for 1pctCO2 and <0.35 GtN for all the hist_SSPs throughout the entire duration of the simulation). The large flux and small pool size change are naturally incompatible with each other in a first-order decay assumption, indicating uncaptured nonlinearities of the mineral pool simulation in NorESM2-LM. And lastly, the CNit emulation has effectively captured the trends in mineral nitrogen pool sizes simulated in CMCC-CM2-SR5, CMCC-ESM2, MPI-ESM1-2-LR, and MIROC-ES2L across all experiments (Fig. A6). Overall, the mineral pool is only a small part of the land nitrogen pool (the mineral to organic nitrogen ratio across the time series was 0.23-0.31%, 0.24-0.31%, 1.36-2.80%, 2.05-2.53%, 0.06-0.19%, and 0.14-0.43% for CMCC-CM2-SR5, CMCC-ESM2, MPI-ESM1-2-LR, NorESM2-LM, UKESM1-0-LL, and MIROC-ES2L, respectively). As a result, the net land nitrogen flux is mainly controlled by the organic nitrogen dynamics and shows consistency for the results from CNit emulation and ESM simulation (Fig. 3 and Fig. A6).

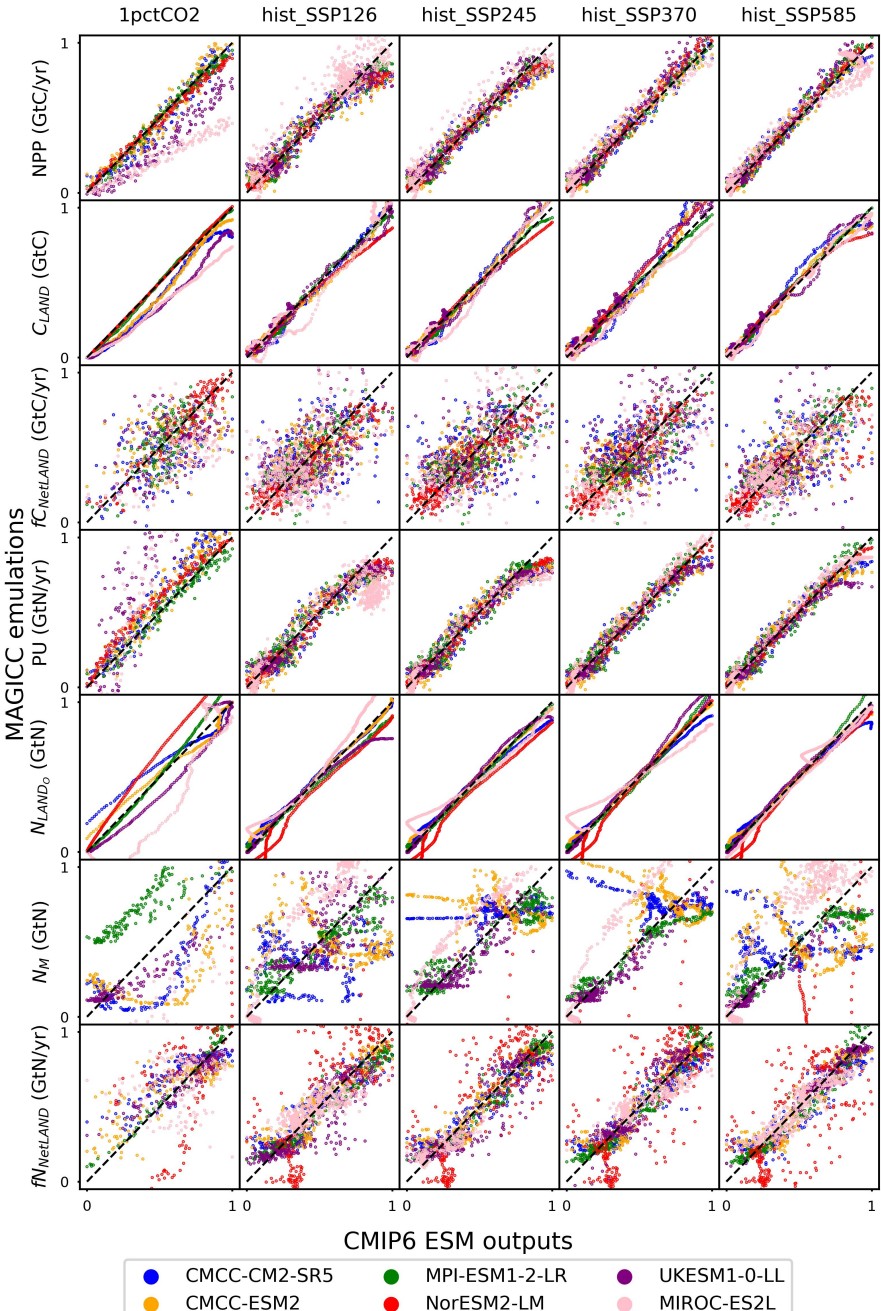

Figure 3. Comparison of 'net primary production' (NPP), 'land carbon pool size' ($C_{LAND}$), 'net land carbon flux' ($fC_{NetLAND}$), 'nitrogen plant uptake' (PU), 'land organic nitrogen pool size' ($N_{LANDo}$, sum of nitrogen in plant, litter, and soil pools), 'mineral nitrogen pool size' ($N_M$), and 'net land nitrogen flux' ($fN_{NetLAND}$) between CMIP6 ESM outputs and CNit emulations. Results are normalized to a range of 0-1 using the following transformation: x-axis = (target − $target_{min}$) / ($target_{max}$ − $target_{min}$), y-axis = (emulation − $target_{min}$) / ($target_{max}$ − $target_{min}$). The diagonal dashed line represents points where the emulation matches the target exactly, with positions below and above the line indicating underestimation and overestimation by the emulator, respectively.

## 4 Discussions

### 4.1 The climate and carbon-nitrogen cycle in CMIP6 ESMs

The climate and carbon-nitrogen cycle in CMIP6 ESMs remain considerably different from each other (Fig. 4, Fig. A7, and Text A1), which contributes to the imperfect emulation (especially the mineral nitrogen pool size, Fig. 3). The following

discussion of the ESM outputs provide an overview of the current model-based understanding of the climate and carbon-nitrogen cycle from a global-mean, annual-mean perspective, from which we suggest future research needs. For the sake of comparison, the subsequent figures/discussions focus on the common experimental periods for scenarios and ESMs (e.g., 1850-2100 for the hist_SSPs). If not specified, the value and spread in the discussion are expressed as mean ± one standard deviation across ESMs.

As a start, different ESMs have different temperature responses (Fig. A3), which has flow-on effects for the carbon cycle and nitrogen cycle (Fig. A6). Regardless of the absolute land carbon/nitrogen pool sizes (Fig. A8 and Text A2), the accumulation of land carbon/nitrogen during the same experimental period and their trends are considerably more similar (Fig. 4A and Fig. 4B). The accumulated land carbon storage is 472±201, 127±28, 164±50, 165±100, and 227±100 GtC for the 1pctCO2, hist_SSP126, hist_SSP245, hist_SSP370, and hist_SSP585 experiments, respectively. The corresponding land nitrogen

accumulation is 3.04±4.65, 3.28±1.03, 3.92±1.33, 3.76±1.32, and 3.54±1.64 GtN. Generally, the land carbon and nitrogen accumulation are proportional to maintain the stoichiometric relationship between carbon and nitrogen (Fig. 4A and Fig. 4B, the mean values). However, the opposite trend of land carbon and nitrogen change is found in the two CMCC ESMs (compared to the other ESMs) in their 1pctCO2 runs, which explains the much larger spread of land organic nitrogen accumulation at the end of 1pctCO2 compared to the four hist_SSPs (Fig. 4B). Both the land carbon and nitrogen storage

show larger spread in the 1pctCO2 experiment (without land use changes), which further highlights the model structure uncertainty (and potentially also points to some inconsistency in how models are running these experiments). The higher spread of carbon and nitrogen storage in higher warming scenarios indicates the feedback uncertainty (Melnikova et al., 2021).

The continuous and rapid depletion of mineral nitrogen is observed in NorESM2-LM under the 1pctCO2 scenario,

coinciding with the highest accumulation of organic nitrogen (Fig. 4C). There are contrasting trends in mineral nitrogen pool size changes among models across all scenarios (Fig. 4C), indicating a large discrepancy and very limited agreement about mineral nitrogen estimation among ESMs. Better understanding and proper representation of organic nitrogen decomposition (nitrogen mineralization) is key to narrow the gap (Thomas et al., 2015; Forsmark et al., 2020; Davies-Barnard et al., 2020).

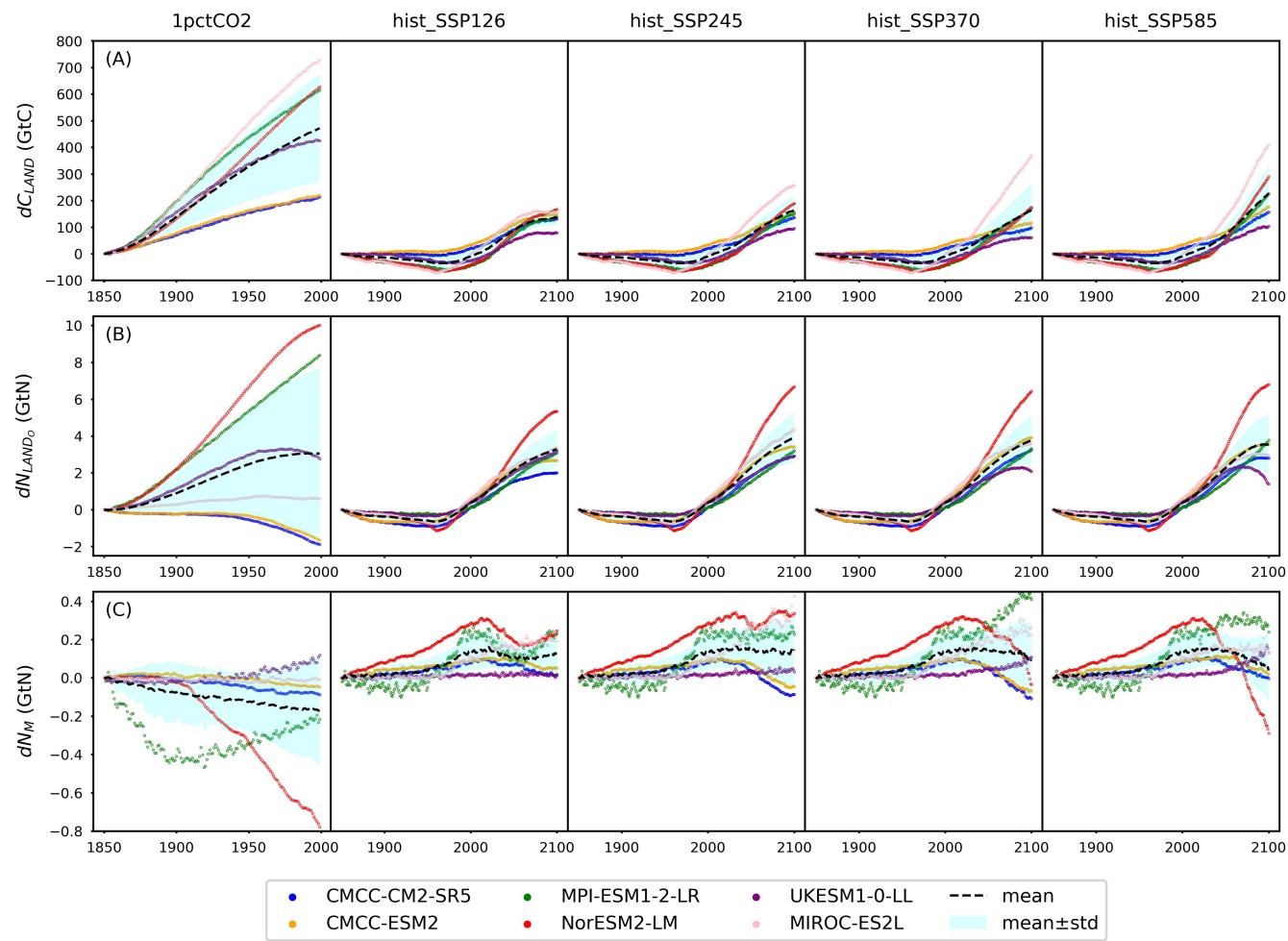

**Figure 4. Carbon and nitrogen pool size dynamics [$dC_{LAND}$, delta land carbon pool size; $dN_{LANDo}$, delta land organic nitrogen pool size (sum of nitrogen in plant, litter, and soil pools); $dN_M$, delta mineral nitrogen pool size] from CMIP6 ESMs across different scenarios.**

Because of the lack of observational constraints and process-level understanding, the terrestrial carbon cycle is a major source of uncertainty contributing to future climate projections and past climate simulations (Friedlingstein et al., 2014; Ciais et al., 2014). By analyzing outputs from 12 CMIP5 ESMs, a study has found that the projection uncertainty of global land carbon storage by 2100 is >160 GtC (more than 50% larger than that of the studied CMIP6 ESMs here), primarily driven by model structure differences (Lovenduski and Bonan, 2017). Though ESMs have been significantly improved over the past years (Eyring et al., 2021; Chen et al., 2021; Eyring et al., 2019; Washington et al., 2009), with the inclusion of a nitrogen cycle as a realistic constraint on the carbon cycle being the most recent (at the time of CMIP6) (Davies-Barnard et al., 2020; Wei et al., 2022), the spread of carbon-concentration feedback is not much narrowed from CMIP5 to CMIP6 (Arora et al., 2020). The variability in nitrogen pool size dynamics (Fig. 4B and Fig. 4C) indicates considerable nitrogen-related

uncertainties accompanied with the carbon-nitrogen coupling (Du et al., 2018; Thomas et al., 2015; Thomas et al., 2013). So far, there is no standardized nitrogen cycle validation for ESMs, largely because of limited observations, diverging representations, and the diverse upscaling approaches used in CMIP6 ESMs (Zaehle et al., 2014; Zaehle and Dalmonech, 2011; Spafford and Macdougall, 2021; Zhu et al., 2018). An improved quantification of nitrogen effects on the carbon cycle and climate, e.g., isolating the nitrogen cycle feedback from the carbon cycle feedback, though requiring meticulous experiment design and extra model simulation, might be necessary to improve climate projection uncertainty attribution (Spafford and Macdougall, 2021). The lack of nitrogen observations and limited mechanistic understanding remain a fundamental challenge in reducing uncertainty related to the nitrogen effect (Zaehle et al., 2014). The forthcoming online calibration of the updated MAGICC to CMIP6 ESMs and its application (e.g., sensitivity analyses, perturbed parameter analyses, and feedback analysis) should provide new insights into the projection uncertainty which would be too computationally expensive to obtain from ESMs (Bonan et al., 2019; Deser et al., 2020).

## 4.2 The nitrogen effect on NPP

The NPP simulated by CMIP6 ESMs has a consistent trend and is much more constrained than the pool sizes (Fig. 5A and Fig. A8). The NPP simulation in CMIP6 ESMs has been improved from the CMIP5 ESMs (Wei et al., 2022), attributed to advancements in nitrogen processes and the availability of more observational data (Collier et al., 2018; Randerson et al., 2009). Based on our calibrations, there is generally nitrogen limitation on global-mean, annual-mean NPP (Fig. 5B), which is consistent with experimental studies and other model simulations (Lebauer and Treseder, 2008; Wieder et al., 2015b; Thornton et al., 2009; Thornton et al., 2007). According to our calibrations, NorESM2-LM is the only model that shows nitrogen fertilization effects on NPP during the historical period, alongside a persistent intensification of nitrogen limitation during the scenario period. This historical fertilization coincides with an enrichment of the model's mineral nitrogen pool, while the increasing limitation correlates with the accumulation of organic nitrogen (see Fig. 4B, Fig. 4C, and Fig. A8D). These findings suggest that nitrogen mineralization is constrained by nitrogen availability during the scenario period in NorESM2-LM, resulting in reduced mineral nitrogen levels and thereby exacerbating NPP limitation.

Based on our calibrations, MPI-ESM1-2-LR and UKESM1-0-LL show the strongest nitrogen limitation on NPP during their hist_SSPs runs, with an $\epsilon_{CN(NPP)}$ range of 0.73-0.88 and 0.77-0.90, respectively. The strong nitrogen limitation from MPI-ESM1-2-LR in its 1pctCO2 simulation matches the continuous depletion of its mineral nitrogen pool (Fig. 4C). It is noted that JSBACH, the land component of MPI-ESM1-2-LR (Mauritsen et al., 2019), shows very limited nitrogen limitation on NPP in its CMIP5 1pctCO2 idealized simulation (Goll et al., 2017). The relatively more severe depletion of mineral nitrogen in its CMIP6 output (maximum >0.43 GtN, Fig. 4C) than its CMIP5 result (maximum <0.3 GtN, value from the reference publication), along with the much higher NPP simulated in CMIP6 (maximum ~110 GtC/yr, Fig. 5A) than CMIP5 (maximum ~40 GtC/yr, value from the reference publication), might be the reason for the strong nitrogen limitation. On the other hand, we suspect that the strong nitrogen limitation on NPP inferred for UKESM1-0-LL, is primarily the result of the

incongruent high NPP and low PU results from UKESM1-0-LL outputs (Fig. A6, i.e. this could be a calibration issue rather

than an easily explained feature of UKESM1-0-LL).

Based on our calibration, CMCC-CM2-SR5 and CMCC-ESM2 exhibit the least pronounced nitrogen limitation (or even fertilization) on NPP during their hist_SSPs runs, with an $\epsilon_{CN(NPP)}$ range of 0.93-1.05 and 0.93-1.03, respectively. The similarly weak limitation is also observed in our calibration to their 1pctCO2 runs. Their organic nitrogen pool sizes keep decreasing in the 1pctCO2 experiment (Fig. 4B and Fig. A8B), which should contribute a large flux of mineral nitrogen via

decomposition. However, their mineral nitrogen pools are not enriched (Fig. A6 and Fig. A8D), indicating a considerable mineral nitrogen loss from these two models. The slight nitrogen fertilization on NPP ($\epsilon_{CN(NPP)} > 1$) is found in our calibration to the CMCC models from 1975 to 2025, or 2100 depending on the scenario. The Community Land Model (CLM) serves as the land component for both CMCC models (Lovato et al., 2022). In the Duke and Oak Ridge National Laboratory (ORNL) Free-Air $CO_2$ Enrichment (FACE) experiments, CLM has exhibited a nearly negligible initial-year

nitrogen-based NPP response (defined as NPP / canopy nitrogen), alongside a 6-10% NPP response to elevated $CO_2$ (Zaehle et al., 2014). The near-zero nitrogen-based NPP response and significant NPP increase suggest that CLM perceives a relatively high canopy nitrogen content (as supported by the large land organic nitrogen pool sizes in CMCC models' output in Fig. A8B). This potentially contributes to the weak nitrogen limitation observed in the CMCC models at the global scale and underscores the potential utility of regional observation studies to constrain the global nitrogen effect. Most nitrogen

limitation factors fall within the range of 0.85-1.0, with limitations increasing in higher scenarios. This finding aligns with the OCN results from the RCP85 and RCP26 simulations [Fig. 2 and (Meyerholt et al., 2020)].

Although both land surface models and our calibrations to ESMs indicate a continuous nitrogen limitation on NPP at the global scale, there is room for debate regarding the realism of a long-term nitrogen limitation. Considering the substantial amounts of atmospheric nitrogen deposition and anthropogenic additions, alongside the ubiquitous presence of nitrogen-

fixing organisms, the ability of ecosystems to optimize nitrogen use efficiency in response to varying nitrogen availability remains unclear. Previous studies suggest that ecosystems should be capable of effectively balancing these factors to alleviate long-term limitations on overall NPP (Vitousek and Howarth, 1991). This holds particularly true for tropical forests where nitrogen is abundant and rapidly circulated (Hedin et al., 2005; Cusack et al., 2011). In such cases, other nutrients like phosphorus and potassium might play a critical role (Wright et al., 2011; Alvarez-Clare et al., 2013). Further validation is

required to assess the long-term effects of nitrogen limitation on NPP and to understand the differences between regional and global patterns.

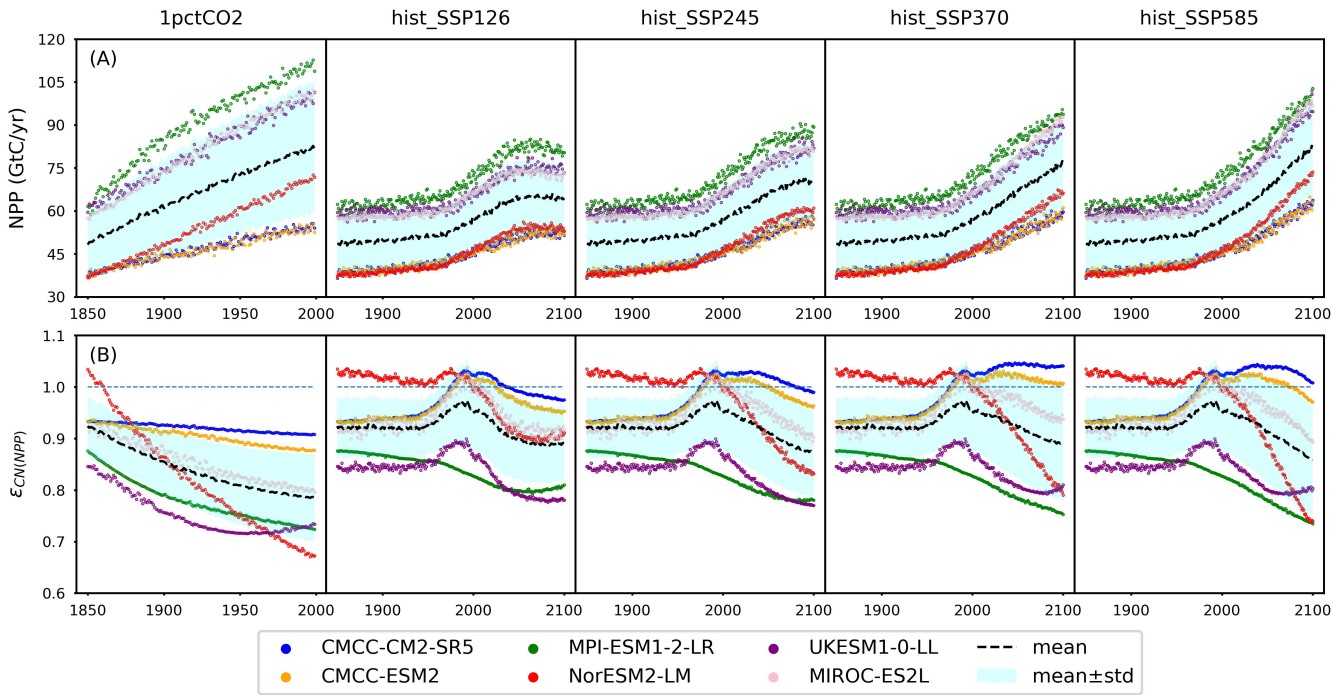

**Figure 5. Net primary production (NPP) from CMIP6 ESMs and the emulated nitrogen effect on NPP ($\epsilon_{CN(NPP)}$) across different scenarios (The blue dashed line serves as a reference for $\epsilon_{CN(NPP)} = 1$)**

## 4.3 The nitrogen effect on pool turnovers

According to our calibrations, the carbon-nitrogen coupling has largely enhanced plant carbon turnover in both CMCC models and MIROC-ES2L, as evidenced by $\epsilon_{CN(LP_C)}$ values ranging from 1.09 to 1.50, 1.07 to 1.50, and 1.12 to 1.40 respectively, during the hist_SSP period (Fig. 6A). The $\epsilon_{CN(LP_C)}$ values exhibit an increasing trend from the SSP126 scenario to the SSP585 scenario. The calibration to the other ESMs, however, suggest that the plant nitrogen status has inhibited litter production ($\epsilon_{CN(LP_C)}$ in the range of 0.6-0.8 during the hist_SSP period and no significant difference among different scenarios). The carbon-nitrogen coupling effect ($\epsilon_{CN}$) and temperature effect ($\epsilon_{dT}$) effectively change the turnover rate determined by the initial turnover times (Eq. 29). Considering that the temperature feedback is always 1 at the beginning of each experiment (dT = 0) and the dynamics of plant carbon are similar across different ESMs (Fig. 4A and Fig. A6), the strong nitrogen enhancement on litter carbon production in the CMCC models is mainly because of their high initial plant carbon turnover times (31 and 66 years, respectively, Table A2), compared to the other ESMs (15 to 25 years, Table A2). The combination allows for the total litter production rate to remain at a similar level for all the ESMs, a key requirement given that they have similar plant pool size changes (Fig. A6), relatively consistent NPP (the influx for plant carbon pool, Fig. 5), and similar plant-litter respiration (the outflux for plant carbon pool, the $LPR_0$ parameter in Table A2). The enhanced litter production in MIROC-ES2L, however, is needed to compensate for its much lower plant respiration flux ($LPR_0 = 0.96$

GtC/yr in MIROC-ES2L vs. 4.16-9.90 GtC/yr in other ESMs). The weak limitation (or even fertilization) of NPP found for the CMCC models (Fig. 5B) and the continuous loss of soil organic nitrogen (Fig. 4B and Fig. A6, 1pctCO2 and historical period) suggest that the system is relatively less nitrogen limited. The resulting plant nitrogen availability partially contributes to the fast plant carbon turnovers in our calibration to the CMCC models.

    Our calibrations to all the ESMs except for MPI-ESM1-2-LR show consistently inhibited litter decomposition after the

nitrogen effect is applied ($\epsilon_{CN(LD_c)}$ in the range of 0.55-0.96 for the hist_SSP runs) and such inhibition slightly increases from SSP126 to SSP585. The significantly enhanced litter decomposition in our calibration to MPI-ESM1-2-LR is attributed to the high carbon input into its litter pool and the small temperature sensitivity, which are supported by 1) the highest NPP and its partition into the litter pool among the ESMs (Fig. 5A and Table A2, $f_{NPP2L}$ = 0.59); 2) the highest litter production partition into litter pool ($f_{LP2L_c}$ = 0.96, Table A2); and 3) the lowest temperature sensitivity of litter decomposition ($s_{dT(LD_c)}$

= 0.024, Table A2). The strong nitrogen limitation on NPP (Fig. 5B), the continuous depletion of the mineral nitrogen pool (Fig. 4C and Fig. A6, 1pctCO2 and historical period), and the highest biological nitrogen fixation (Fig. 7A) in our calibration to MPI-ESM1-2-LR indicates that the plant nitrogen is insufficient. The strong $\epsilon_{CN(LD_c)}$ along with the low $f_{LD2S_c}$ (0.07, Table A2) suggests that the system is trying to mineralize more litter carbon to mediate the plant nitrogen deficiency and to maintain an ecologically reasonable carbon:nitrogen stoichiometry.

The soil respiration is found to be restricted in our calibration to UKESM1-0-LL ($\epsilon_{CN(SR_c)}$ = 0.61-0.69 during the hist_SSP period), while it is significantly enhanced in other ESMs. The strongest enhancement is found in the two CMCC models ($\epsilon_{CN(SR_c)}$ = 1.24-1.50 during the hist_SSP period), followed by NorESM2-LM ($\epsilon_{CN(SR_c)}$ = 1.20-1.36 during the hist_SSP period). The calibrations to these three models show longer soil carbon turnover times (283-476 years, Table A2) than the other ESMs. The order of magnitude larger mineral nitrogen pool size in NorESM2-LM than other ESMs (Fig. A8D) and the

continuously growing mineral nitrogen and organic nitrogen pool sizes (Fig. 4C, hist_SSPs) support the enhanced soil organic matter decomposition. It is observed that the mineral nitrogen pool size exhibits a continuous decrease during the NorESM2-LM's 1pctCO2 run, while it first increases and then decreases in its SSP585 run (Fig. 4D and Fig. A6). Considering that plant uptake and net mineralization are the two major fluxes controlling the mineral nitrogen dynamics, this result suggests a potential threshold associated with climate or CO2 concentration, limiting the net mineralization rate from

matching the ongoing increase in plant uptake (Fig. A6). The high temperature change (Fig. A7) and its subsequent high temperature effect on respiration in UKESM1-0-LL could be responsible for the nitrogen-inhibited soil respiration. The substantial land carbon accumulation in MIROC-ES2L (Fig. 4A) requires less respiration, thus explaining the neglectable nitrogen effect on its soil respiration. The diverse impacts of nitrogen on soil carbon turnover align with existing experimental findings, which have demonstrated contrasting trends in nitrogen additions across various substrate

decompositions (Averill and Waring, 2018; Hobbie, 2008). As a result, studies have suggested that the classic stoichiometric decomposition theory should be revised (Craine et al., 2007).

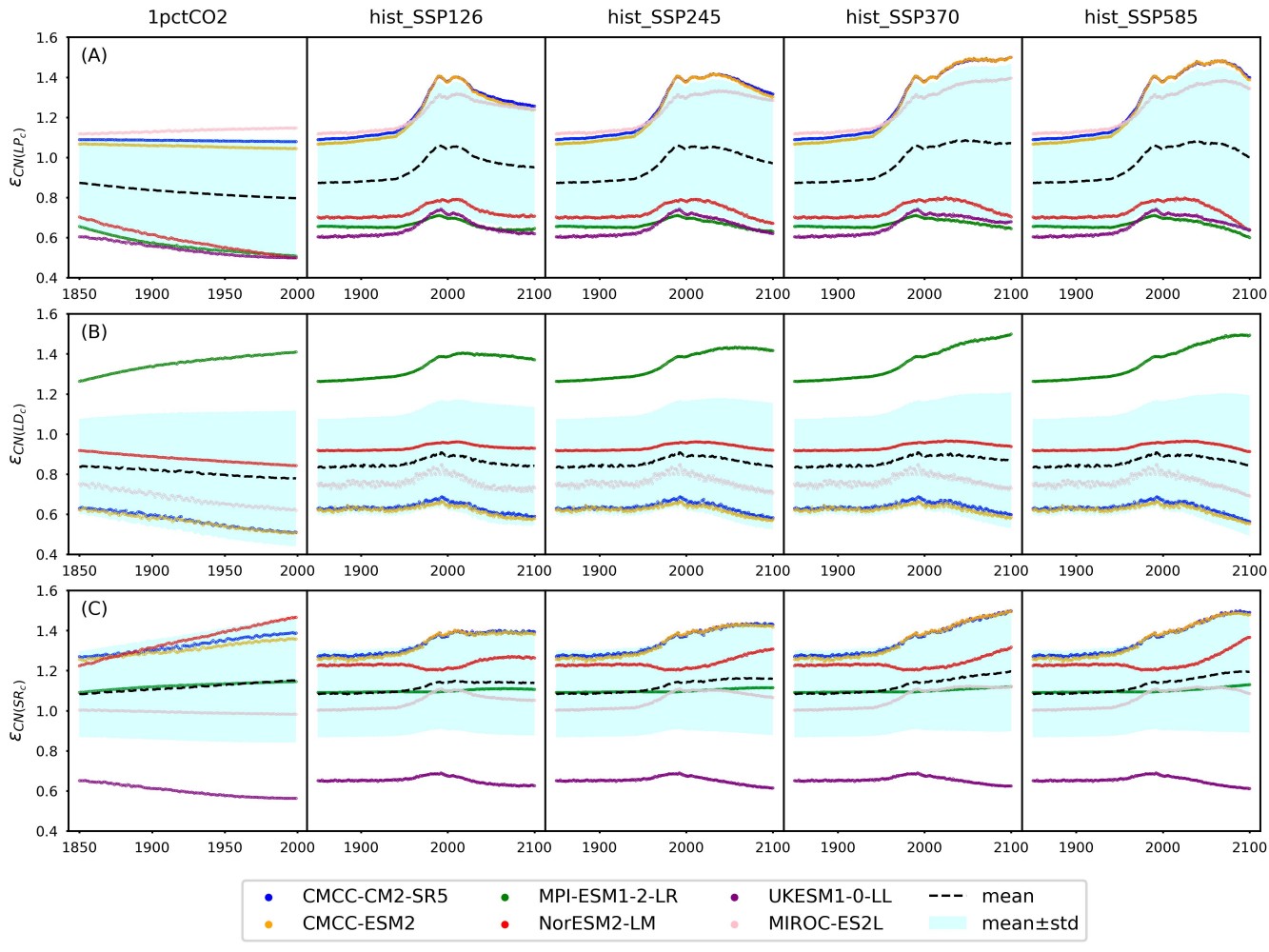

**Figure 6. The emulated nitrogen effect on carbon pool turnovers, including litter production ($\epsilon_{CN(LP_c)}$), litter decomposition ($\epsilon_{CN(LD_c)}$) and soil respiration ($\epsilon_{CN(SR_c)}$) from CMIP6 ESMs across different scenarios.**

## 5 Limitations and implications

A key limitation of RCMs is their resolution, which arises from the trade-off between spatial heterogeneity and computational efficiency. As such, it is important to note that the feedbacks, pool size dynamics, and fluxes discussed in this paper are aggregated from extensive spatiotemporal datasets into an annual and global framework. This means the representation of the carbon-nitrogen cycle reflects a synthesis of diverse regional and sub-annual dynamics. Consequently, the results presented here are at a global scale and may differ from regional or sub-regional studies, necessitating cautious interpretation.

## 5.1 The simulation of mineral nitrogen pool dynamics

Our emulation is less effective in capturing the mineral nitrogen pool sizes for some ESMs (e.g., NorESM2-LM, Fig. 3 and Fig. A6). However, the large mineral nitrogen pool in NorESM2-LM (Fig. A8D) and the significant discrepancies in mineral nitrogen pool size changes among ESMs (Fig. 4C) and land surface models (Fig. 2) highlight the need for a deeper understanding of these dynamics in complex models. This should include enhanced theoretical underpinning and improved observational constraints (Zaehle and Dalmonech, 2011; Thomas et al., 2015). The heterogeneous terrestrial nitrogen cycle results in challenges and discrepancies even compared to the regional observations we do have (Schulte-Uebbing and De Vries, 2018; Ramm et al., 2022; Menge et al., 2012). The uncertain atmospheric deposition and nitrogen fertilizer application further complicate the evaluation of the mineral nitrogen pool size and its dynamics (Mulvaney et al., 2009; Reay et al., 2008; Gruber and Galloway, 2008).

The two main controls of the mineral nitrogen pool size, the nitrogen mineralization (from organic decomposition) and mineral nitrogen loss, are still poorly understood at the process level (Manzoni et al., 2008; Hedin et al., 2005). For instance, the microbial decomposition of organic matter (heterotrophic respiration) can be limited, stimulated, or even unaffected by the nitrogen addition as a result of differences in soil microbial biomass or activity changes (Bardgett et al., 1999). The nitrogen effect on decomposition has been found to be sensitive to the types of substrates, but generally the impact on decomposition rate is negative or neutral (Hobbie, 2008). The root allocation, plant growth, litter production, biodiversity, etc., are all influenced by nitrogen (Phoenix et al., 2006; Wright et al., 2011). However, a 13-year-long nitrogen addition study has found that lower nitrogen addition rates had no effect on litter production or soil respiration in a Pinus sylvestris forest (Forsmark et al., 2020). This raises questions about the overall nitrogen effect at the ecosystem level, particularly considering the uneven geographical distribution of atmospheric nitrogen deposition (Phoenix et al., 2006). Alongside climate factors such as warming and precipitation, as well as other ecological or physical constraints, the situation becomes even more complicated (Plett et al., 2020; Lim et al., 2015; Cusack et al., 2010; Reich et al., 2014; Li et al., 2019). These findings underscore the highly complex dynamics of mineral nitrogen, suggesting that the current formulation of mineral nitrogen loss in CNit, characterized by simple first-order decay and a single temperature response (Eq. 29), is very likely an oversimplification. Introducing constraints to the global stoichiometry of nitrogen mineralization could potentially enhance the modeling of mineral nitrogen pool dynamics (Manzoni et al., 2008; Meyerholt and Zaehle, 2015). Nevertheless, these new developments in MAGICC are a step towards better representation of carbon-nitrogen dynamics. The calibration results also demonstrate that the current CNit formulation mostly captures the trend of mineral nitrogen pool sizes in the majority of the studied CMIP6 ESMs and experiments at the global-, annual-mean level (Fig. 3 and Fig. A6).

## 5.2 The biological nitrogen fixation as an input instead of being simulated

'Biological nitrogen fixation' (BNF) serves as the primary non-anthropogenic nitrogen input in the global nitrogen cycle (Vitousek et al., 2002; Gruber and Galloway, 2008; Fowler et al., 2013). The trend of BNF generally mirrors that of NPP

(Fig. 7A and Fig. 5A), but the relative differences between ESMs are more pronounced (e.g., the largest initial biological nitrogen fixation found in MPI-ESM1-2-LR is approximately four times that of the smallest one found in NorESM2-LM). This similarity in trend arises because the model representations of BNF in all the studied ESMs were based, at least partly, on its empirical relationship with either NPP or evapotranspiration derived from the widely recognized global BNF estimate (Cleveland et al., 1999). Fig. 7B illustrates a variety of global patterns of BNF:NPP in the CMIP6 ESM outputs, showcasing diverse trends such as decreases (CMCC-CM2-SR5, CMCC-ESM2, and MIROC-ES2L in the 1pctCO2 experiment), increases (UKESM1-0-LL in the 1pctCO2 experiment and MIROC-ES2L in SSP experiments), stability (UKESM1-0-LL in SSP scenarios), or even a peak followed by a decrease (NorESM2-LM in low SSP scenarios and MIROC-ES2L in high SSP scenarios). It is noteworthy that the mean BNF:NPP remains relatively constant at approximately 0.0018 GtN/GtC across all experiments.

Recent findings from a meta-analysis of field measurements challenge the notion of a statistically significant relationship between biological nitrogen fixation and NPP/evapotranspiration (Davies-Barnard and Friedlingstein, 2020). An analysis of model uncertainty in recent studies highlighted that variations in the BNF representation could significantly impact future climate projections (Wieder et al., 2015a; Kou-Giesbrecht and Arora, 2022). For instance, employing different BNF representations within a shared framework has resulted in modeled BNF responses to elevated (200 ppm higher) $CO_2$ ranging from -4 to 56 $\times 10^{-3}$ GtN/yr (Meyerholt et al., 2016). This variation has led to a global land carbon storage range of 281 to 353 GtC (over ~150 years of simulation), with $N_2O$ emissions fluctuating from -1.6 to 0.5 $\times 10^{-3}$ GtN/yr (Meyerholt et al., 2016). A recent study assessing BNF structural uncertainty in CMIP6 ESMs has revealed that the response of BNF and other nitrogen cycle variables could differ, even among models with similar structures (Davies-Barnard et al., 2022). Conflicting empirical relationships and updated observations underscore the considerable uncertainty and potential need for revisions to BNF formulations in ESMs. While we could develop parameterizations to emulate the BNF formulations used in ESMs, the BNF flux represents a relatively minor flux with questionable data quality and highly uncertain formulations and/or mechanisms (e.g., CABLE prescribes constant BNF). Therefore, we opt not to pursue it further here. Instead, CNit directly prescribes BNF from CMIP6 ESM outputs to circumvent further structural uncertainty stemming from simplified parameterization.

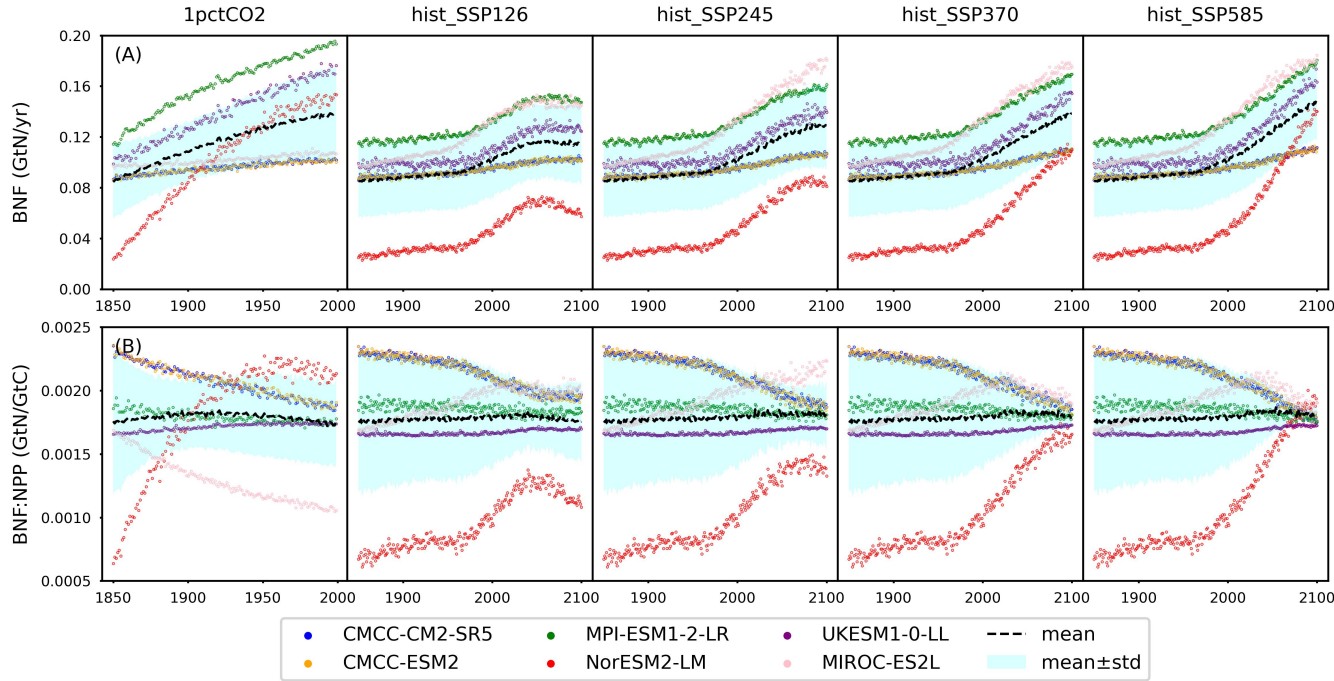

**Figure 7. Biological nitrogen fixation (BNF) and its ratio to net primary production (BNF:NPP) from CMIP6 ESMs across different scenarios.**

### 5.3 The disentangled climate feedback and nitrogen effect from emulation

One advantage of RCMs is that their simplified formulations attempt to capture the overall effects of complex processes in ESMs, aiding the identification and quantification of key effects in the system. Based on the assumptions and definitions in this updated carbon-nitrogen cycle, we can separate the temperature feedback and carbon-nitrogen coupling feedback for different pool turnovers (Eq. 29). However, a significant challenge in this separation arises from the exponential formulation of both feedbacks and the increasing trends of all the feedback proxies ['temperature change' (dT), 'nitrogen plant uptake' (PU), and 'nitrogen atmospheric deposition' (AD), Fig. A4 and Fig. A6]. This setup suggests that these feedbacks (or the related sensitivity parameters) may change in opposite directions to compensate each other while still producing a similar overall feedback. The formulations (Eq. 29) suggest that the turnover time and the overall feedback can also offset each other to reach a similar turnover flux.

To examine the correlation of parameter values and feedback separation, we applied Markov chain Monte Carlo (MCMC) sampling for the sensitivity parameters and turnover times for each of the individual ESMs (60 walkers × 1,000 iterations = 60,000 runs, starting from the "best-estimate" parameter values). The results show that there is a weak-to-moderate negative correlation between temperature sensitivity and plant uptake sensitivity for most of the ESMs (the absolute value of Spearman's r = 0.04-0.72 for most cases, highlighted in yellow in Fig. A9). The strongest correlations are found in

NorESM2-LM (Spearman's r = -0.53, -0.72, and -0.46 for the litter production, litter decomposition and soil respiration, respectively). The weakest correlations are found in MIROC-ES2L (Spearman's r = -0.04, -0.34, and 0.04 for the litter production, litter decomposition and soil respiration, respectively). The temperature sensitivity and atmospheric deposition sensitivity show relatively weak correlations (the absolute value of Spearman's r < 0.3 for most cases, Fig. A9).

The weak-to-moderate correlations between temperature sensitivity and plant uptake sensitivity are mainly due to the feedback proxies, dT and PU, which, though both exhibiting the same increasing trend, do not strictly change with the same gradients (e.g., the temperature change fluctuates near zero while plant uptake shows a clear increasing trend from 1850-1975 in the historical simulations, Fig. A4 and Fig. A6). Increasing one sensitivity while decreasing another, though it could lead to similar overall feedback at the early stage when temperature change and plant uptake are less perturbed, cannot guarantee similar overall feedback throughout the entire time series (e.g., the difference of overall feedback from different sensitivities amplifies as the temperature change gets higher and plant uptake becomes larger). In other words, the parameter values cannot simply vary all in the opposite direction to compensate for the feedback. Instead, to reach the desirable turnover flux to satisfy the pool size dynamics, the parameter values need to be adjusted (not necessarily offset each other) to obtain the respective "correct" or "best-estimate" feedback. Such results suggest that separating the carbon-nitrogen coupling feedback from the temperature feedback (the assumption for Eq. 29) is a reasonable assumption, although this should be investigated further in future work and is achieved most readily in simulations that specifically target this separation, for example the nitrogen-on and nitrogen-off experiments performed by the land surface models we used here.

The turnover times and plant uptake sensitivities exhibit strong positive correlations in all the ESMs (displayed in bold in Fig. A9), indicating that plant uptake sensitivity is the predominant factor influencing the overall feedback to compensate for turnover time changes. This is supported by the dominance of carbon-nitrogen coupling feedback in both the magnitudes and trends of the overall feedback (Fig. 6 and Fig. A10), further emphasizing the substantial disparity between temperature feedback and carbon-nitrogen coupling feedback, and the imperative of distinguishing the two.

One limitation of disentangling the carbon-nitrogen coupling feedback from the climate feedback is that the feedback strength is primarily derived from emulation. Although several factors support this distinction: 1) the evidence presented in Fig. A9 and Fig. A10 underscores the clear differentiation between climate feedback and carbon-nitrogen coupling feedback; 2) the dynamics of pool size offer indirect yet compelling constraints (referred to as "emergent" constraints) for the feedback; and 3) the selection of feedback formulations (exponential rather than linear relationships) and proxies (with varying magnitudes) further restricts the parameters from offsetting each other. However, the absence of nitrogen-off simulations from the CMIP6 ESMs presents challenges for direct verification. Given the computational expense of running all scenarios in nitrogen-off mode, it is recommended that ESMs perform nitrogen-off simulations for select idealized scenarios [e.g., 1pctCO2 or flat10 - constant emissions of $CO_2$ of 10 GtC per year (Sanderson et al., 2024)] for diagnostic purposes.

# 6 Conclusion and future work

In this work we have detailed a new coupled carbon-nitrogen model - CNit - within the reduced complexity model, MAGICC. Based on the offline calibration results from land surface models and multiple CMIP6 ESMs, we have demonstrated that the new carbon-nitrogen cycle model is able to effectively emulate the behavior of the carbon-nitrogen cycles from various, more complex models, encompassing a broad spectrum of carbon-nitrogen states and dynamics.

The temperature change and carbon-nitrogen state/dynamics (especially those related to the nitrogen cycle and mineral 840 nitrogen) exhibit significant variability among CMIP6 ESMs, particularly in their 1pctCO2 and high SSP scenario runs, which highlights the model structure uncertainty. The contrasting trends in mineral nitrogen dynamics and the magnitude of order differences in pool sizes underscore the limited agreement in mineral nitrogen modeling. A thorough analysis, focusing on the new uncertainties introduced by the nitrogen cycle is imperative for the CMIP6 and future ESMs. The upcoming (in future research) sensitivity analysis, perturbation parameter analysis, and feedback analysis of the updated MAGICC model 845 are expected to provide insights for uncertainty attribution.

The CNit emulation indicates a general nitrogen limitation on NPP, which follows a similar trend across the studied CMIP6 ESMs. Combining the results from NPP and turnovers suggests that, at the multi-model mean level, the carbon-nitrogen coupling limits both NPP and plant and litter carbon pool turnovers, though the weaker NPP nitrogen limitation could also lead to significantly enhanced litter production. Soil respiration is instead enhanced in most of the ESMs. The combination 850 indicates that terrestrial ecosystems may become net carbon sources sooner than we would expect based on models that do not consider the impact of the nitrogen cycle.

The presented carbon-nitrogen coupling in MAGICC demonstrates the ability to emulate many complex models, while nonetheless having limitations, particularly in simulating mineral nitrogen pool dynamics and biological nitrogen fixation. There are currently significant inconsistencies between ESM outputs and observations of the mineral nitrogen pool size and 855 biological nitrogen fixation, both in terms of magnitudes and trends, suggesting that substantial revisions are possible in the near future. Therefore, the current formulation and treatment of these aspects in MAGICC may have to be updated too, while aiming to continue to strike a balance between model simplicity, process representation, and emulation performance, reflecting a fundamental design principle for RCMs and MAGICC in particular. Future work on MAGICC's carbon-nitrogen cycle will focus on the calibration of the full MAGICC structure to CMIP6 ESMs (and/or observational data), evaluation of 860 model performance with respect to computational efficiency and mechanistic insight, incorporation of additional constraints, uncertainty quantification, sensitivity analysis, application of probabilistic projections, and continued model development (e.g., land use emission implementation and nitrogen process representation) to align with advances in complex models and emerging theoretical frameworks.

 **Appendix**

### Table A1. Full list of calibrated CNit parameters: Long name and range

| Parameter | Long name | Range for the calibration |
|---|---|---|
| $NPP_0$ | Initial 'net primary production' (NPP), base NPP without any effect | 40-60 GtC/yr |
| $CO_{2ref}$ | reference CO2 concentration, typically the CO2 at pre-industrial level | Fixed as initial year CO2 |
| $CO_{2b}$ | CO2 concentration when NPP = 0 in the rectangular hyperbolic formulation of CO2 fertilization | Fixed as 31 ppm |
| $s_{CO_2}^{log}$ | CO2 sensitivity of NPP in the logarithmic formulation | 0-3 (dimensionless) |
| $s_{CO_2}^{sig}$ | CO2 sensitivity of NPP in the sigmoidal formulation | 250-350 ppm |
| $m_{CO_2}$ | Method factor for NPP CO2 fertilization calculation | 0-2 (dimensionless) |
| $s_{dT(NPP)}^{exp}$ | Temperature sensitivity of NPP in the exponential formulation | -0.3-0.3 K$^{-1}$ |
| $s_{dT(NPP)}^{sig}$ | Temperature sensitivity of NPP in the sigmoidal formulation | -1.5-1.5 K$^{-1}$ |
| $m_{dT}$ | Method factor for NPP temperature response calculation | 0-1 (dimensionless) |
| $\varphi$ | Fraction of regrowth from deforestation | 0-1 (dimensionless) |
| $\tau_{rgr}$ | Time for deforestation regrowth | 50-150 yr |
| $PU_{max}$ | Maximum 'nitrogen plant uptake' (PU) | 0-3 GtN/yr |
| $NPP_{ref}$ | Reference NPP for PU | 0-120 GtC/yr |
| $s_{dT(PU)}$ | Temperature sensitivity of PU | -0.3-0.3 K$^{-1}$ |
| $\epsilon_{CN(NPP)0}$ | Base carbon-nitrogen coupling effect on NPP | 0-3 (dimensionless) |
| $f_1$ | Fitting parameter representing the nitrogen deficiency from net mineralization alone | -3-0 (dimensionless) |
| $f_2$ | Fitting parameter representing the nitrogen supply from atmospheric deposition | 0-3 (dimensionless) |
| $LPR_0$ | Initial 'litter production respiration' (LPR), base LPR without any effect | 0-10 GtC/yr |
| $s_{dT(LPR)}$ | Temperature sensitivity of LPR | -0.3-0.3 K$^{-1}$ |
| $f_{NPP2P}$ | Fraction of NPP allocated to plant carbon pool within one year | 0-1 (dimensionless) |
| $f_{NPP2L}$ | Fraction of NPP allocated to litter carbon pool within one year | 0-1 (dimensionless) |
| $f_{LP2L_c}$ | Fraction of litter carbon production allocated to litter carbon pool within one year | 0-1 (dimensionless) |
| $f_{LD2S_c}$ | Fraction of litter carbon decomposition allocated to soil carbon pool within one year | 0-1 (dimensionless) |
| $f_{LU2P_c}$ | Fraction of land use carbon loss from plant carbon pool within one year | 0-1 (dimensionless) |
| $f_{LU2L_c}$ | Fraction of land use carbon loss from litter carbon pool within one year | 0-1 (dimensionless) |
| $\tau_{C_P}$ | Turnover time of plant carbon pool | 0-800 yr |
| $\tau_{C_L}$ | Turnover time of litter carbon pool | 0-800 yr |
| $\tau_{C_S}$ | Turnover time of soil carbon pool | 0-800 yr |
| $s_{dT(LP_c)}$ | Temperature sensitivity of litter carbon production | -0.3-0.3 K$^{-1}$ |
| $s_{dT(LD_c)}$ | Temperature sensitivity of litter carbon decomposition | -0.3-0.3 K$^{-1}$ |
| $s_{dT(SR_c)}$ | Temperature sensitivity of soil carbon decomposition | -0.3-0.3 K$^{-1}$ |
| $s_{PU(LP_c)}$ | Nitrogen plant uptake sensitivity of litter carbon production | -10-10 yr/GtN |
| $s_{PU(LD_c)}$ | Nitrogen plant uptake sensitivity of litter carbon decomposition | -10-10 yr/GtN |
| $s_{PU(SR_c)}$ | Nitrogen plant uptake sensitivity of soil carbon respiration | -10-10 yr/GtN |

| | | |
|---|---|---|
| $s_{AD(LP_c)}$ | Nitrogen atmospheric deposition sensitivity of litter carbon production | -10-10 yr/GtN |
| $s_{AD(LD_c)}$ | Nitrogen atmospheric deposition sensitivity of litter carbon decomposition | -10-10 yr/GtN |
| $s_{AD(SR_c)}$ | Nitrogen atmospheric deposition sensitivity of soil carbon respiration | -10-10 yr/GtN |
| $f_{BNF2P}$ | Fraction of 'biological nitrogen fixation' (BNF) allocated to plant nitrogen pool within one year | 0-1 (dimensionless) |
| $f_{BNF2L}$ | Fraction of BNF allocated to litter nitrogen pool within one year | 0-1 (dimensionless) |
| $f_{PU2P}$ | Fraction of PU allocated to plant nitrogen pool within one year | 0-1 (dimensionless) |
| $f_{PU2L}$ | Fraction of PU allocated to litter nitrogen pool within one year | 0-1 (dimensionless) |
| $f_{LP2L_n}$ | Fraction of litter nitrogen production allocated to litter carbon pool within one year | 0-1 (dimensionless) |
| $f_{LD2S_n}$ | Fraction of litter nitrogen decomposition allocated to soil carbon pool within one year | 0-1 (dimensionless) |
| $f_{LU2P_n}$ | Fraction of land use carbon loss from plant nitrogen pool within one year | 0-1 (dimensionless) |
| $f_{LU2L_n}$ | Fraction of land use carbon loss from litter nitrogen pool within one year | 0-1 (dimensionless) |
| $\tau_{N_P}$ | Turnover time of plant nitrogen pool | 0-800 yr |
| $\tau_{N_L}$ | Turnover time of litter nitrogen pool | 0-800 yr |
| $\tau_{N_S}$ | Turnover time of soil nitrogen pool | 0-800 yr |
| $\tau_{N_M}$ | Turnover time of mineral nitrogen pool | 0-100 yr |
| $s_{dT(LP_n)}$ | Temperature sensitivity of litter nitrogen production | -0.3-0.3 K$^{-1}$ |
| $s_{dT(LD_n)}$ | Temperature sensitivity of litter nitrogen decomposition | -0.3-0.3 K$^{-1}$ |
| $s_{dT(SR_n)}$ | Temperature sensitivity of soil nitrogen decomposition | -0.3-0.3 K$^{-1}$ |
| $s_{dT(LS_n)}$ | Temperature sensitivity of mineral nitrogen loss | -0.3-0.3 K$^{-1}$ |
| $s_{PU(LP_n)}$ | Nitrogen plant uptake sensitivity of litter nitrogen production | -10-10 yr/GtN |
| $s_{PU(LD_n)}$ | Nitrogen plant uptake sensitivity of litter nitrogen decomposition | -10-10 yr/GtN |
| $s_{PU(SR_n)}$ | Nitrogen plant uptake sensitivity of soil nitrogen respiration | -10-10 yr/GtN |
| $s_{AD(LP_n)}$ | Nitrogen atmospheric deposition sensitivity of litter nitrogen production | -10-10 yr/GtN |
| $s_{AD(LD_n)}$ | Nitrogen atmospheric deposition sensitivity of litter nitrogen decomposition | -10-10 yr/GtN |
| $s_{AD(SR_n)}$ | Nitrogen atmospheric deposition sensitivity of soil nitrogen respiration | -10-10 yr/GtN |

**Table A2. Full list of calibrated CNit parameters: Value**

| Parameter | CMCC-CM2-SR5 | CMCC-ESM2 | MPI-ESM1-2-LR | NorESM2-LM | UKESM1-0-LL | MIROC-ES2L | CABLE | OCN |
|---|---|---|---|---|---|---|---|---|
| $NPP_0$ | 41.92 | 41.88 | 68.93 | 36.23 | 69.95 | 62.06 | 57.38 | 53.98 |
| $CO_{2ref}$ | 284.317 | 284.317 | 284.317 | 284.317 | 284.317 | 284.317 | 296.474 | 285.24 |
| $CO_{2b}$ | 31 | 31 | 31 | 31 | 31 | 31 | 31 | 31 |
| $s_{CO_2}^{log}$ | 0.000 | 0.788 | 1.451 | 0.948 | 0.113 | 0.004 | 2.582 | 0.594 |
| $s_{CO_2}^{sig}$ | 335.90 | 297.10 | 337.71 | 263.82 | 250.77 | 269.25 | 315.82 | 289.88 |
| $m_{CO_2}$ | 1.82 | 2.00 | 0.53 | 0.01 | 0.99 | 2.00 | 0.00 | 1.00 |
| $s_{dT(NPP)}^{exp}$ | -0.293 | 0.108 | -0.300 | -0.121 | -0.223 | -0.016 | -0.143 | -0.156 |
| $s_{dT(NPP)}^{sig}$ | 0.245 | 0.143 | 1.192 | 0.314 | 0.249 | 0.278 | -0.147 | 0.512 |
| $m_{dT}$ | 0.82 | 0.99 | 0.30 | 0.56 | 0.84 | 0.93 | 0.83 | 0.38 |
| $\varphi$ | 0.99 | 1.00 | 0.97 | 1.00 | 1.00 | 1.00 | 1.00 | 0.94 |
| $\tau_{rgr}$ | 96.00 | 73.02 | 149.00 | 149.18 | 50.05 | 93.51 | 52.72 | 107.39 |
| $PU_{max}$ | 2.57 | 3.00 | 2.06 | 2.30 | 2.67 | 2.42 | 1.89 | 2.17 |
| $NPP_{ref}$ | 49.45 | 55.91 | 48.21 | 41.78 | 118.97 | 107.90 | 54.17 | 40.79 |
| $s_{dT(PU)}$ | -0.003 | -0.013 | 0.011 | 0.015 | -0.048 | -0.019 | 0.014 | 0.008 |
| $\epsilon_{CN(NPP)0}$ | 0.96 | 1.03 | 1.23 | 1.41 | 1.23 | 1.07 | 1.58 | 1.19 |
| $f_1$ | 2.31 | 2.26 | 0.00 | 1.25 | 1.69 | 2.54 | 0.64 | 0.26 |
| $f_2$ | -0.09 | -0.19 | -0.33 | -0.46 | -0.84 | -0.47 | -0.80 | -0.17 |
| $LPR_0$ | 8.58 | 6.20 | 9.90 | 6.56 | 4.16 | 0.96 | 7.81 | 9.00 |
| $s_{dT(LPR)}$ | -0.25 | -0.22 | -0.12 | 0.16 | -0.19 | 0.30 | -0.10 | 0.06 |
| $f_{NPP2P}$ | 0.62 | 0.38 | 0.36 | 0.63 | 0.49 | 0.66 | 0.95 | 0.54 |
| $f_{NPP2L}$ | 0.20 | 0.36 | 0.59 | 0.35 | / | 0.20 | 0.03 | 0.41 |
| $f_{LP2L_c}$ | 0.94 | 0.92 | 0.96 | 0.72 | / | 0.49 | 0.89 | 0.99 |
| $f_{LD2S_c}$ | 0.01 | 0.11 | 0.07 | 0.07 | / | 0.97 | 0.02 | 0.00 |
| $f_{LU2P_c}$ | 0.88 | 0.88 | 0.88 | 0.92 | 0.94 | 0.56 | 0.53 | 0.11 |
| $f_{LU2L_c}$ | 0.00 | 0.04 | 0.10 | 0.05 | / | 0.30 | 0.09 | 0.84 |
| $\tau_{C_P}$ | 31.26 | 66.11 | 24.42 | 22.56 | 14.96 | 16.14 | 15.46 | 22.89 |
| $\tau_{C_L}$ | 1.30 | 1.35 | 7.40 | 1.15 | 99.99 | 7.72 | 4.09 | 6.98 |
| $\tau_{C_S}$ | 452.96 | 283.42 | 117.35 | 476.71 | 20.99 | 22.01 | 125.82 | 290.81 |
| $s_{dT(LP_c)}$ | 0.040 | 0.056 | 0.054 | -0.124 | 0.040 | -0.051 | 0.001 | -0.061 |
| $s_{dT(LD_c)}$ | 0.073 | 0.063 | 0.024 | -0.028 | / | 0.032 | 0.045 | -0.007 |
| $s_{dT(SR_c)}$ | 0.043 | 0.045 | 0.046 | -0.042 | 0.064 | 0.027 | 0.066 | 0.046 |
| $s_{PU(LP_c)}$ | -0.032 | -0.078 | -0.517 | -0.562 | -1.778 | 0.142 | 0.079 | 0.060 |
| $s_{PU(LD_c)}$ | -0.740 | -0.759 | 0.222 | -0.142 | / | -1.009 | -0.008 | 0.104 |
| $s_{PU(SR_c)}$ | 0.309 | 0.288 | 0.098 | 0.299 | -1.363 | -0.113 | -0.058 | 0.009 |
| $s_{AD(LP_c)}$ | 5.716 | 6.354 | 3.095 | 3.929 | 6.884 | 3.015 | 0.304 | -0.826 |

| | | | | | | | | |
|---|---|---|---|---|---|---|---|---|
| $s_{AD(LD_c)}$ | 3.925 | 3.738 | 1.453 | 1.235 | / | 4.438 | 0.417 | -0.420 |
| $s_{AD(SR_c)}$ | 0.833 | 1.158 | -0.190 | -1.236 | 3.092 | 2.355 | 0.043 | 0.026 |
| $f_{BNF2P}$ | 0.00 | 0.05 | 0.32 | 0.13 | 0.13 | 0.15 | 0.73 | 0.23 |
| $f_{BNF2L}$ | 0.01 | 0.17 | 0.48 | 0.02 | / | 0.21 | 0.04 | 0.25 |
| $f_{PU2P}$ | 0.23 | 0.14 | 0.04 | 0.41 | 0.04 | 0.98 | 0.17 | 0.13 |
| $f_{PU2L}$ | 0.74 | 0.34 | 0.82 | 0.06 | / | 0.01 | 0.72 | 0.00 |
| $f_{LP2L_n}$ | 0.66 | 0.16 | 0.40 | 0.03 | / | 0.17 | 0.04 | 0.19 |
| $f_{LD2S_n}$ | 0.77 | 0.46 | 0.01 | 0.78 | / | 0.88 | 0.37 | 0.89 |
| $f_{LU2P_n}$ | 0.61 | 0.12 | 0.31 | 0.24 | 0.13 | 0.10 | 0.51 | 0.16 |
| $f_{LU2L_n}$ | 0.33 | 0.13 | 0.30 | 0.41 | / | 0.28 | 0.39 | 0.25 |
| $\tau_{N_P}$ | 14.12 | 30.66 | 28.03 | 36.84 | 46.98 | 15.74 | 12.81 | 33.79 |
| $\tau_{N_L}$ | 0.61 | 2.20 | 6.81 | 10.85 | 1.02 | 670.15 | 3.03 | 14.23 |
| $\tau_{N_S}$ | 690.09 | 728.42 | 318.18 | 601.92 | 222.76 | 247.10 | 108.00 | 180.87 |
| $\tau_{N_M}$ | 6.49 | 6.73 | 11.46 | 58.17 | 0.92 | 1.37 | 1.99 | 0.44 |
| $s_{dT(LP_n)}$ | 0.011 | -0.015 | -0.051 | -0.062 | -0.010 | -0.018 | 0.027 | -0.031 |
| $s_{dT(LD_n)}$ | 0.065 | 0.038 | 0.000 | 0.014 | 0.002 | 0.005 | 0.021 | 0.037 |
| $s_{dT(SR_n)}$ | 0.014 | 0.011 | 0.010 | 0.005 | 0.049 | 0.028 | 0.056 | 0.007 |
| $s_{dT(LS_n)}$ | 0.056 | 0.051 | -0.005 | 0.299 | 0.012 | 0.042 | -0.007 | 0.088 |
| $s_{PU(LP_n)}$ | -0.306 | 0.134 | -0.724 | 0.600 | -2.519 | 1.159 | -0.896 | 0.583 |
| $s_{PU(LD_n)}$ | -1.029 | -0.449 | 0.498 | 0.148 | 0.000 | 1.909 | 0.473 | -0.661 |
| $s_{PU(SR_n)}$ | 0.810 | 0.892 | 0.161 | 0.977 | -1.047 | 1.041 | -0.073 | 0.685 |
| $s_{AD(LP_n)}$ | 2.282 | 0.791 | 4.425 | 2.845 | 6.632 | 1.790 | 0.188 | -2.124 |
| $s_{AD(LD_n)}$ | 2.504 | 1.205 | 0.955 | 0.224 | 0.001 | -0.064 | -0.052 | -1.172 |
| $s_{AD(SR_n)}$ | -0.510 | -0.777 | -0.745 | -0.844 | 2.294 | 1.851 | -0.975 | -0.401 |

Note: The missing values (indicated by "/") for UKESM1-0-LL are due to the model's lack of a litter pool. As a result, there are no turnover times, feedback-related parameters, or fractionation values associated with the litter pool.

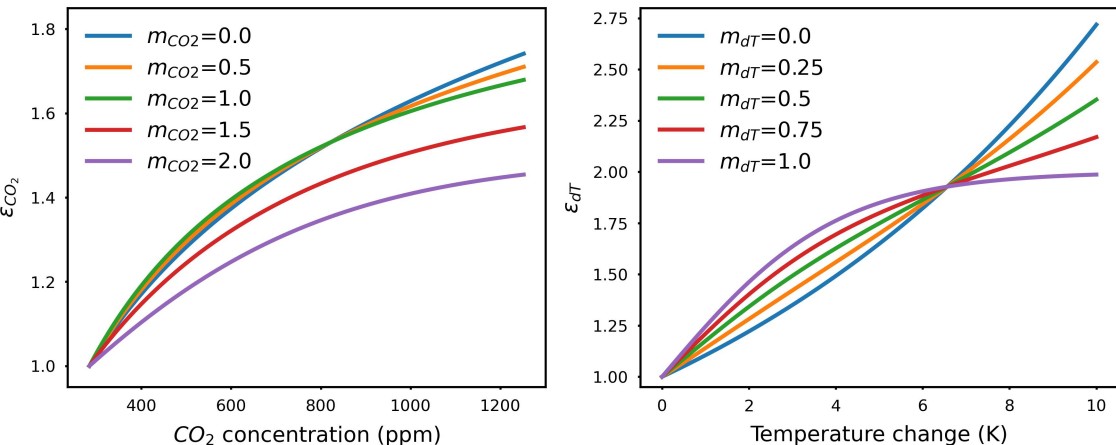

Figure A1. Illustration of the functionality of the method factor for $CO_2$ fertilization ($m_{CO2}$) and temperature feedback ($m_{dT}$). An $m_{CO2}$ of 0, 1, and 2 represents the logarithmic, rectangular, and sigmoidal $CO_2$ fertilization formulations, respectively (Eq. 17). Similarly, an $m_{dT}$ of 0 and 1 corresponds to the exponential and sigmoidal temperature response formulations, respectively (Eq. 20). Intermediate values represent a linear combination of the two formulations.

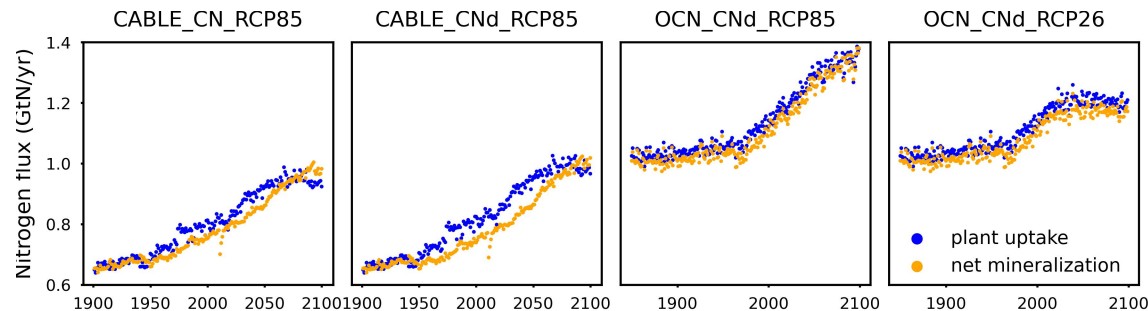

Figure A2. Relationship between nitrogen plant uptake and net mineralization as simulated by CABLE and OCN.

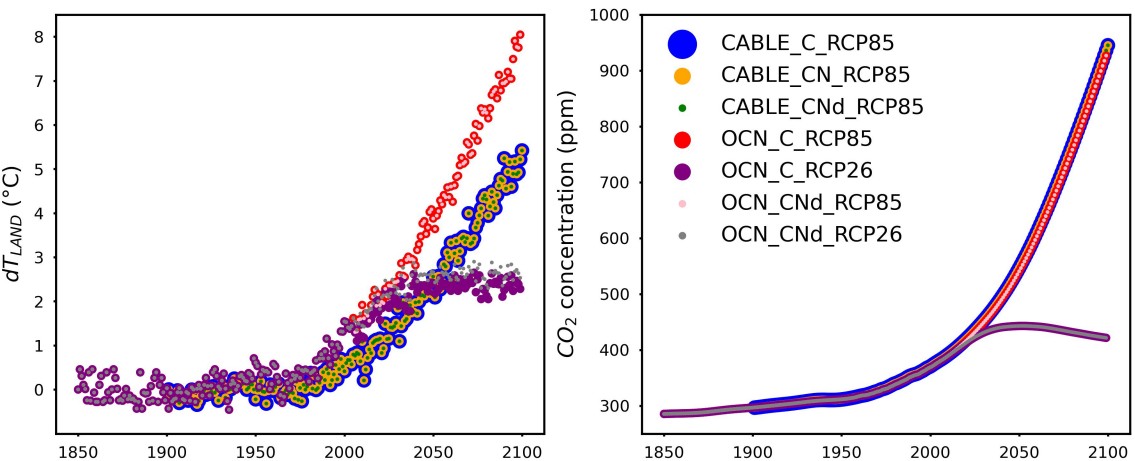

**Figure A3. Global average surface temperature change over land ($dT_{LAND}$, delta annual mean tas over land) and CO₂ concentrations from land surface models across different scenarios.**

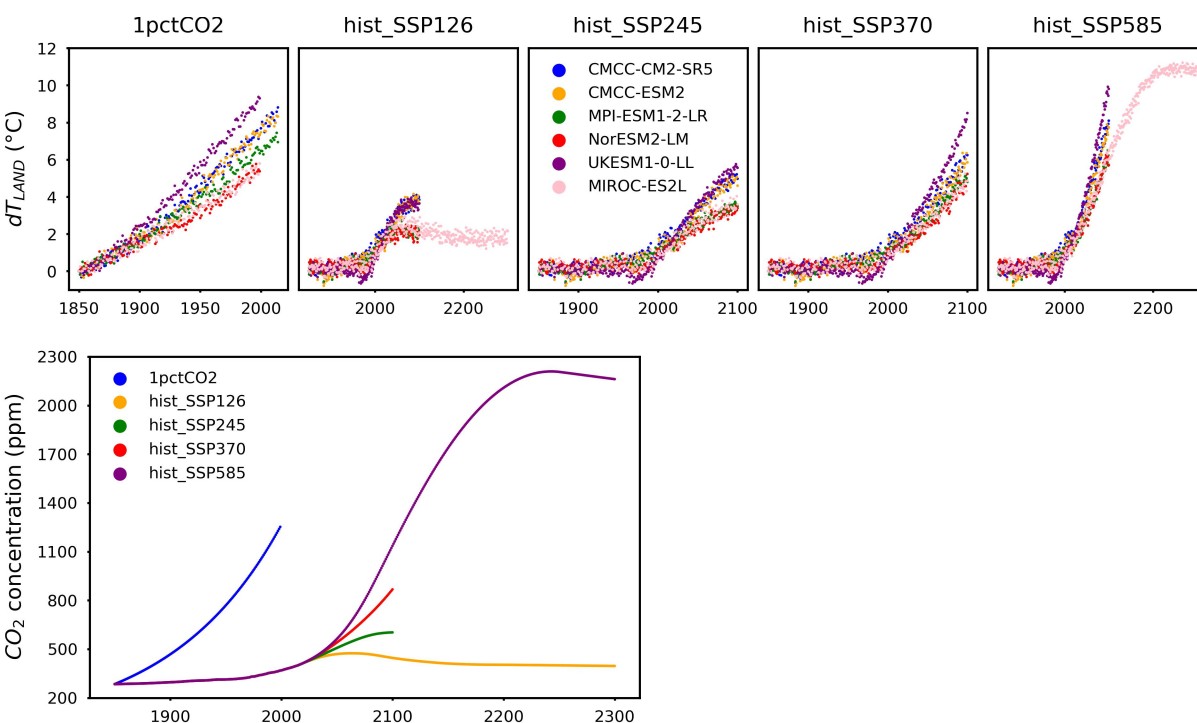

**Figure A4. Global average 'surface temperature change over land' ($dT_{LAND}$, delta annual mean tas over land) and CO₂ concentrations from CMIP6 ESMs across different scenarios.**

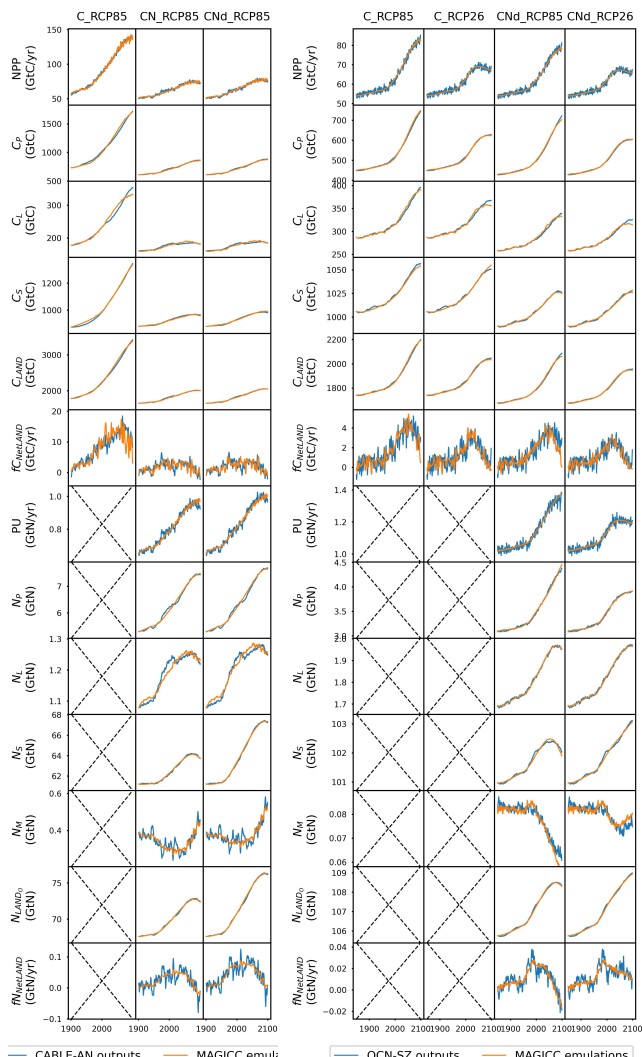

**Figure A5.** Comparison of 'net primary production' (NPP), 'plant carbon pool size' ($C_P$), 'litter carbon pool size' ($C_L$), 'soil carbon pool size' ($C_S$), 'land carbon pool size' ($C_{LAND}$), 'net land carbon flux' ($fC_{NetLAND}$), 'nitrogen plant uptake' (PU), 'plant nitrogen pool size' ($N_P$), 'litter nitrogen pool size' ($N_L$), 'soil nitrogen pool size' ($N_S$), 'mineral nitrogen pool size' ($N_M$), 'land organic nitrogen pool size' ($N_{LANDo}$, sum of nitrogen in plant, litter, and soil pools), and 'net land nitrogen flux' ($fN_{NetLAND}$) between CABLE or OCN outputs (blue lines) and CNit emulations (orange lines). The experiments labeled as C, CN, and CNd denote the carbon-only, carbon-nitrogen coupled with constant nitrogen atmospheric deposition, and carbon-nitrogen coupled with dynamic nitrogen atmospheric deposition configurations in the land surface models, respectively.

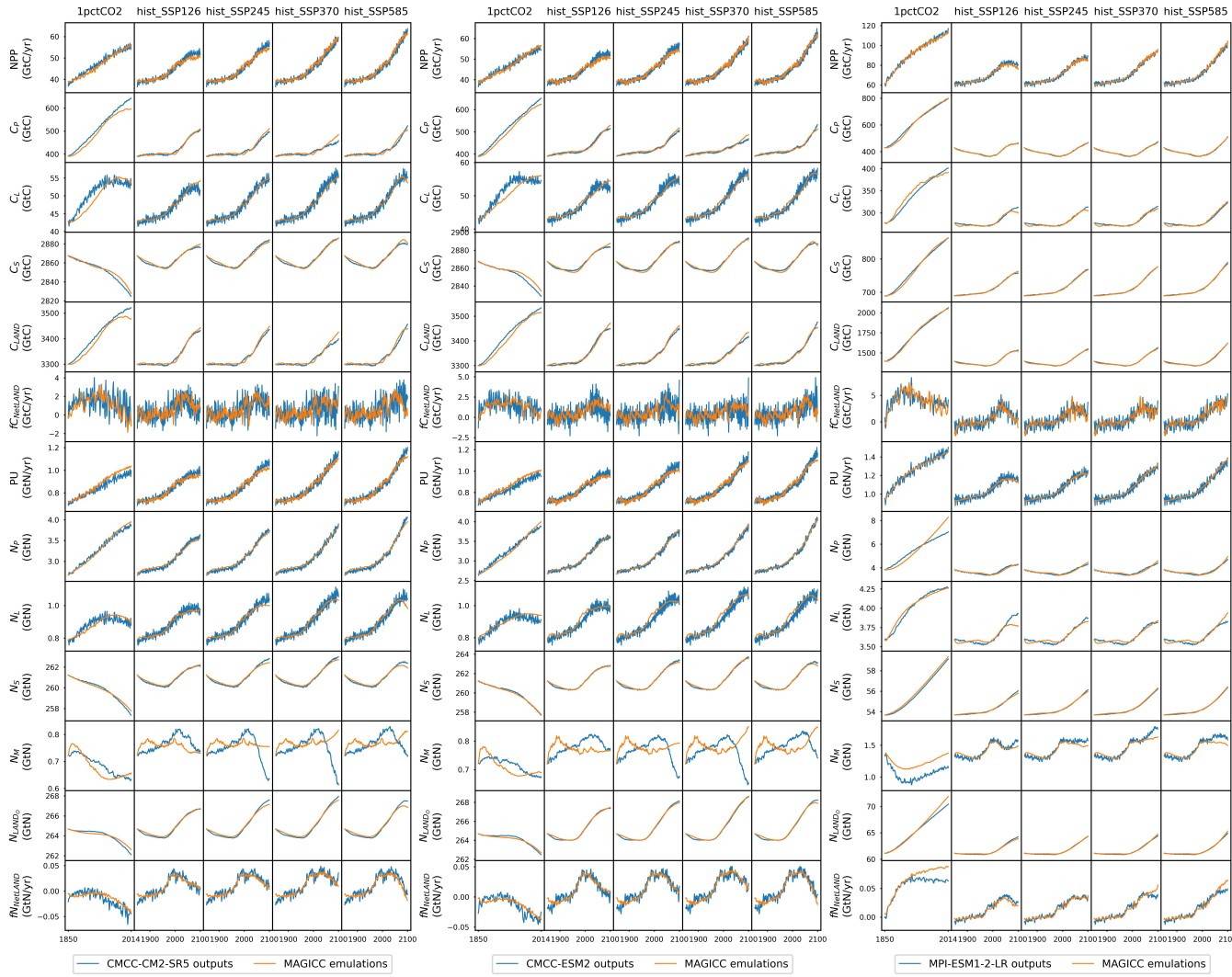

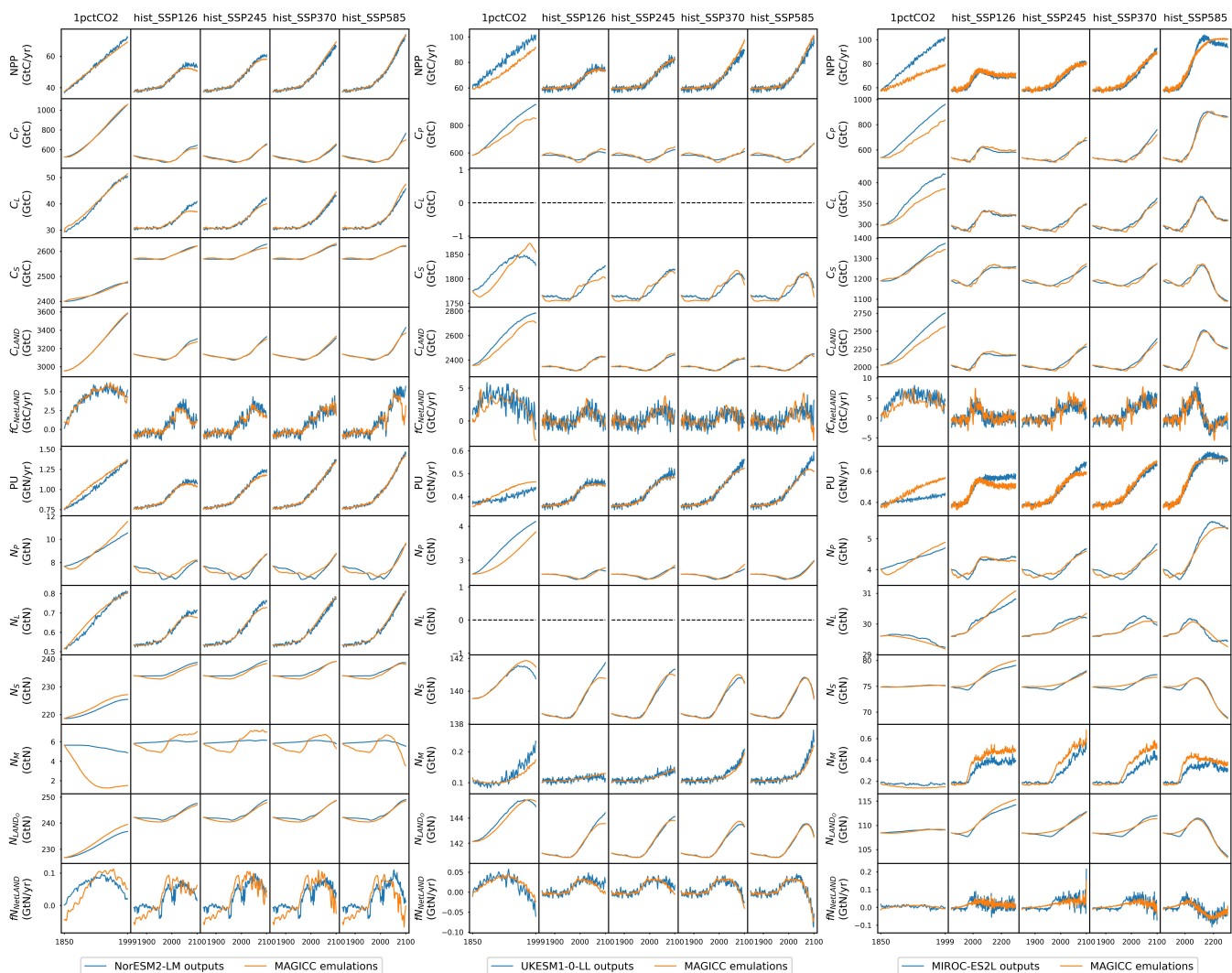

**Figure A6.** Comparison of 'net primary production' (NPP), 'plant carbon pool size' (C_P), 'litter carbon pool size' (C_L), 'soil carbon pool size' (C_S), 'land carbon pool size' (C_LAND), 'net land carbon flux' (fC_NetLAND), 'nitrogen plant uptake' (PU), 'plant nitrogen pool size' (N_P), 'litter nitrogen pool size' (N_L), 'soil nitrogen pool size' (N_S), 'mineral nitrogen pool size' (N_M), 'land organic nitrogen pool size' (N_LANDo, sum of nitrogen in plant, litter, and soil pools), and 'net land nitrogen flux' (fN_NetLAND) between CMIP6 ESM outputs (blue lines) and CNit emulations (orange lines).

**Text A1. The diversity of temperature output from CMIP6 ESMs**

Temperature change, a pivotal driving force for the carbon-nitrogen cycle, exhibits significant variation between the two land surface models (Fig. A3) and among the simulations of CMIP6 ESMs (Fig. A4), even when they undergo the same experiment. For the sake of comparison, Fig. A7 and the subsequent discussions focus on the common experimental periods for scenarios (e.g., 1850-2100 for the hist_SSPs).

UKESM1-0-LL shows the highest temperature change among all models and experiments, whereas NorESM2-LM exhibits the lowest temperature change. Both the idealized 1pctCO2 - one of the base experiments in the Diagnostic, Evaluation and

915 Characterization of Klima (DECK) experiments - and the historical simulation is in the core set of experiments performed under CMIP5, CMIP6, and previous CMIPs (Eyring et al., 2016; Taylor et al., 2012). As CMIP6 and C4MIP necessitate consistent forcings and experimental protocols for simulations conducted by participating ESMs (Eyring et al., 2016; Jones et al., 2016), the wide spread of the land surface temperature change - especially from the 1pctCO2 experiment where land use change is not included (with a standard deviation of 1.3°C and absolute difference of 3.9°C at the end of the simulation) -

920 highlights the various parameterizations of physical processes in ESMs resulting in large differences of the ESMs climate sensitivities (Rugenstein et al., 2020; Meehl et al., 2020), for example, the structural uncertainty (Deser et al., 2020; Duan et al., 2021). The previous MAGICC simulation with constrains from historical $CO_2$ measurements and temperature observations is found reducing uncertainty in the temperature projections (Bodman et al., 2013).

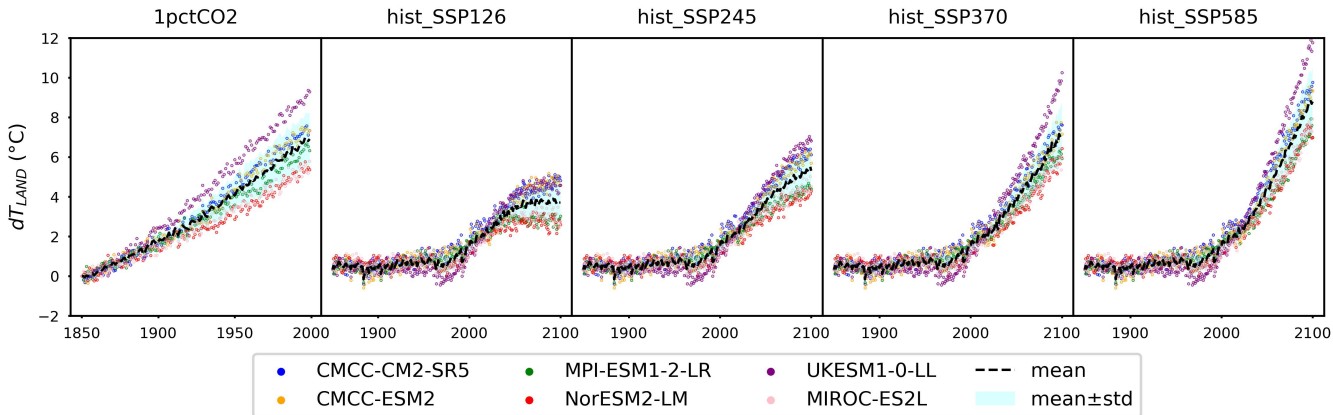

**Figure A7. Global average 'surface temperature change over land' ($dT_{LAND}$, delta annual mean tas over land) from CMIP6 ESMs across different scenarios (over the common experimental period).**

Recent studies interpreting surface air temperature outputs from multiple CMIP6 ESMs indicate that the multi-model mean effectively captures the historical temperature trend in observations (Fan et al., 2020; Papalexiou et al., 2020). Results from a study using outputs from 29 CMIP6 ESMs show that the post-1988 warming is overestimated in 90% of the simulations and

930 the observed long-term persistence of global mean temperature (for the period of 1880-2014) is not accurately captured in most of ESMs (Papalexiou et al., 2020), suggesting further model selections based on the case-specific intended uses. However, previous evaluation of the long-term persistence of temperature on continental areas (60°S-60°N) during 1930-2004 in CMIP5 ESMs demonstrated that most models captured the long-term persistence reasonably well (Kumar et al., 2013). Moreover, grouping CMIP6 ESMs and re-analysing the global mean temperature based on the grouped models can

also lead to different conclusions on the warming trend (Scafetta, 2023). These results indicate that more careful interpretation of the simulated temperature was needed. They also justify using each ESM's global mean land temperature as

input in this study instead of the global mean temperature (to avoid differences in calibration based on inconsistency with the target model's temperature rather than any issue with the reduced complexity model).

**Text A2. The diversity of carbon-nitrogen cycle in CMIP6 ESMs**

Based on the varied temperature results, it is not surprising that the carbon-nitrogen cycle fluxes and pools from the CMIP6 ESMs are diverse (Fig. A8). The initial land carbon pool ranges from 1396 GtC (MPI-ESM1-2-LR) to notably higher values of 3300 GtC (CMCC-CM2-SR5 and CMCC-ESM2). The initial nitrogen pool sizes are even more inconsistent among ESMs. The largest initial organic nitrogen pool is 265 GtN from the two CMCC models, which is more than four times of

945 the smallest one from MPI-ESM1-2-LR (61 GtN). These results have led to a wide range of initial organic carbon:nitrogen ratio from 12 (the two CMCC models) to 23 (MPI-ESM1-2-LR) (Fig. A8C). The trends for the carbon pool size and carbon:nitrogen ratio exhibit a similar pattern. They display a consistent increase in the 1pctCO2 scenario, while in the hist_SSP simulations, they initially decrease (1850-1970) and then rise again (1970-2100). The mineral nitrogen pool, on the contrary, shows significant variations in both pool sizes and trends across CMIP6 ESMs (Fig. A8D). The initial pool sizes

range from <0.2 GtN (UKESM1-0-LL and MIROC-ES2L) to 5.6 GtN (NorESM2-LM). The trends are found either opposite or unrelated.

The substantial variation of simulated carbon pools is a long-standing issue for both CMIP5 and CMIP6 ESMs (Anav et al., 2013; Varney et al., 2022). The initial condition differences are responsible for the models' internal variability (Deser et al., 2020; Kumar and Ganguly, 2018), which accounts for more than half of the inter-model spread in near-term climate

projections (Deser et al., 2012). Such differences also contribute to their respective carbon cycle projections. The different initial carbon-nitrogen cycle state among different ESMs further complicates their comparison (Spafford and Macdougall, 2021). The standard deviation of initial land carbon pool sizes is different for the 1pctCO2 (699 GtC) and historical (720 GtC) scenarios. Four of the studied ESMs have used nearly the same starting land carbon pool size for both scenarios (difference <1.5 GtC) while UKESM1-0-LL and NorESM2-LM have a large difference of 9 GtC and 184 GtC (out of a total

of roughly 2300 GtC in UKESM1-0-LL and 3000 GtC in NorESM2-LM), respectively. The varying initial pool sizes can pose significant challenges for emulators employing first-order decay for pool turnovers, as the turnover time predominantly influences the magnitude of the "base" turnover flux (e.g., without any feedback scalers). Since MAGICC emulation has used the same set of parameters (including the turnover times) to emulate all the experiments, it explains the jump of the emulated soil carbon pool sizes in the 1pctCO2 experiment for these two models (Fig. A6).

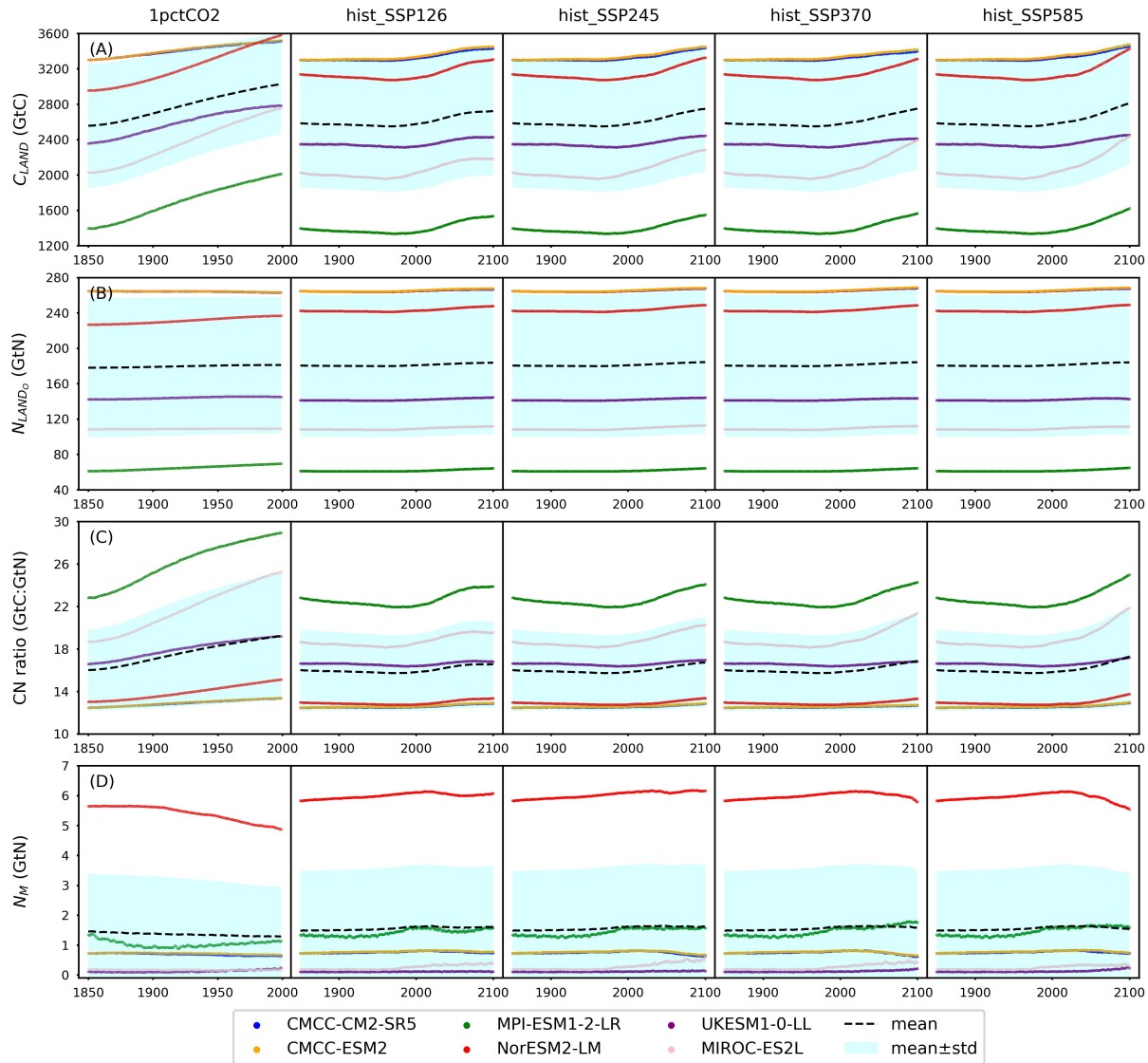

**Figure A8.** Diversity of 'land carbon pool size' ($C_{LAND}$), 'land organic nitrogen pool size' ($N_{LAND_0}$, sum of nitrogen in plant, litter, and soil pools), 'carbon:nitrogen ratio' (CN ratio), and 'mineral nitrogen pool size' ($N_M$) from CMIP6 ESM outputs.

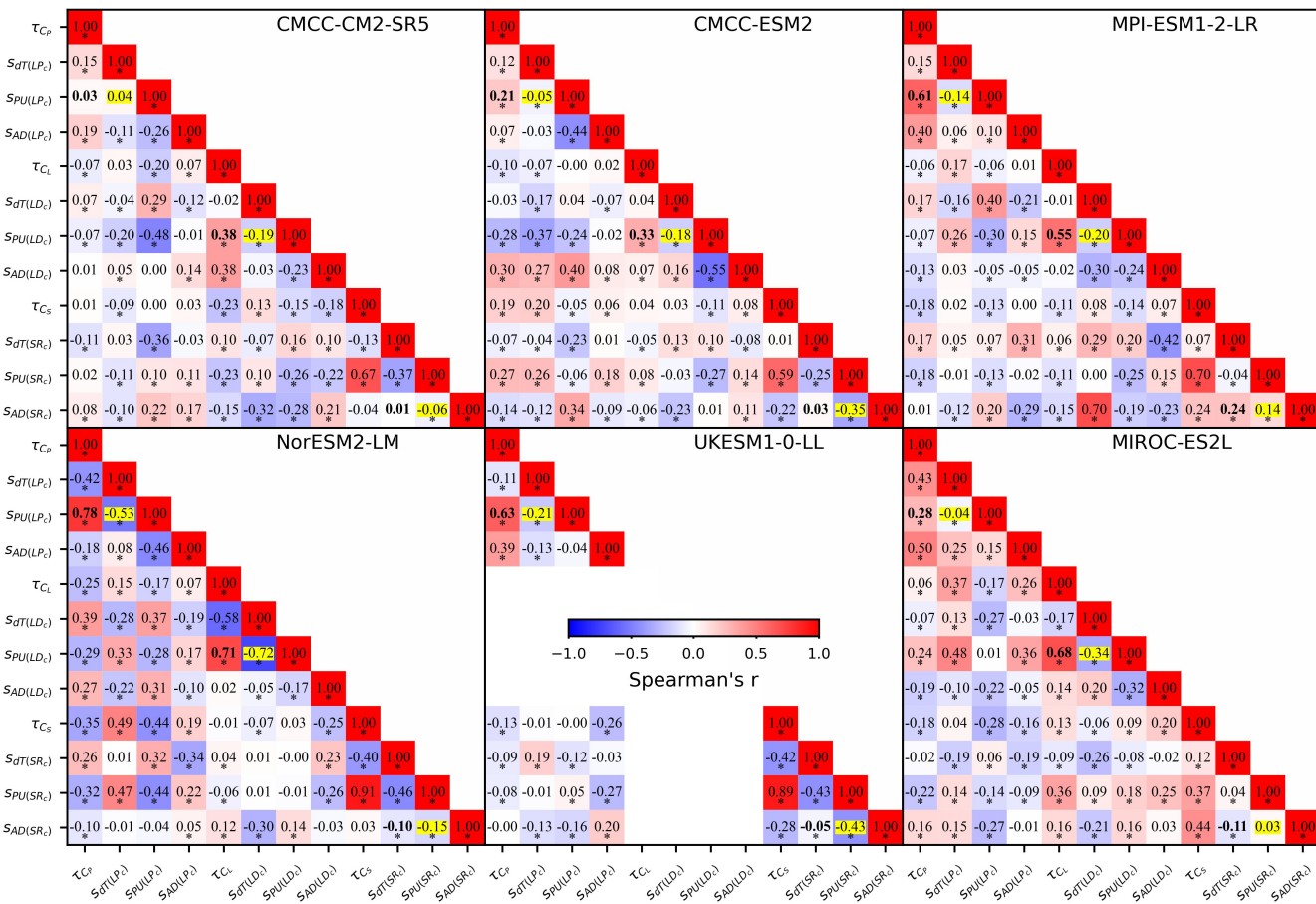

**Figure A9. Correlation of turnover times and feedback-related parameters from the CMIP6 ESMs. The numbers indicate Spearman's correlation coefficients (r) between pairs of parameters, with * denoting p-values < 0.001. Correlations between temperature sensitivities and plant nitrogen uptake sensitivities are highlighted in yellow, while correlations between turnover times and plant nitrogen uptake sensitivities are shown in bold. Missing values for UKESM1-0-LL are due to the absence of a litter pool in this model, resulting in no turnover time or feedback-related parameters for the litter pool.**

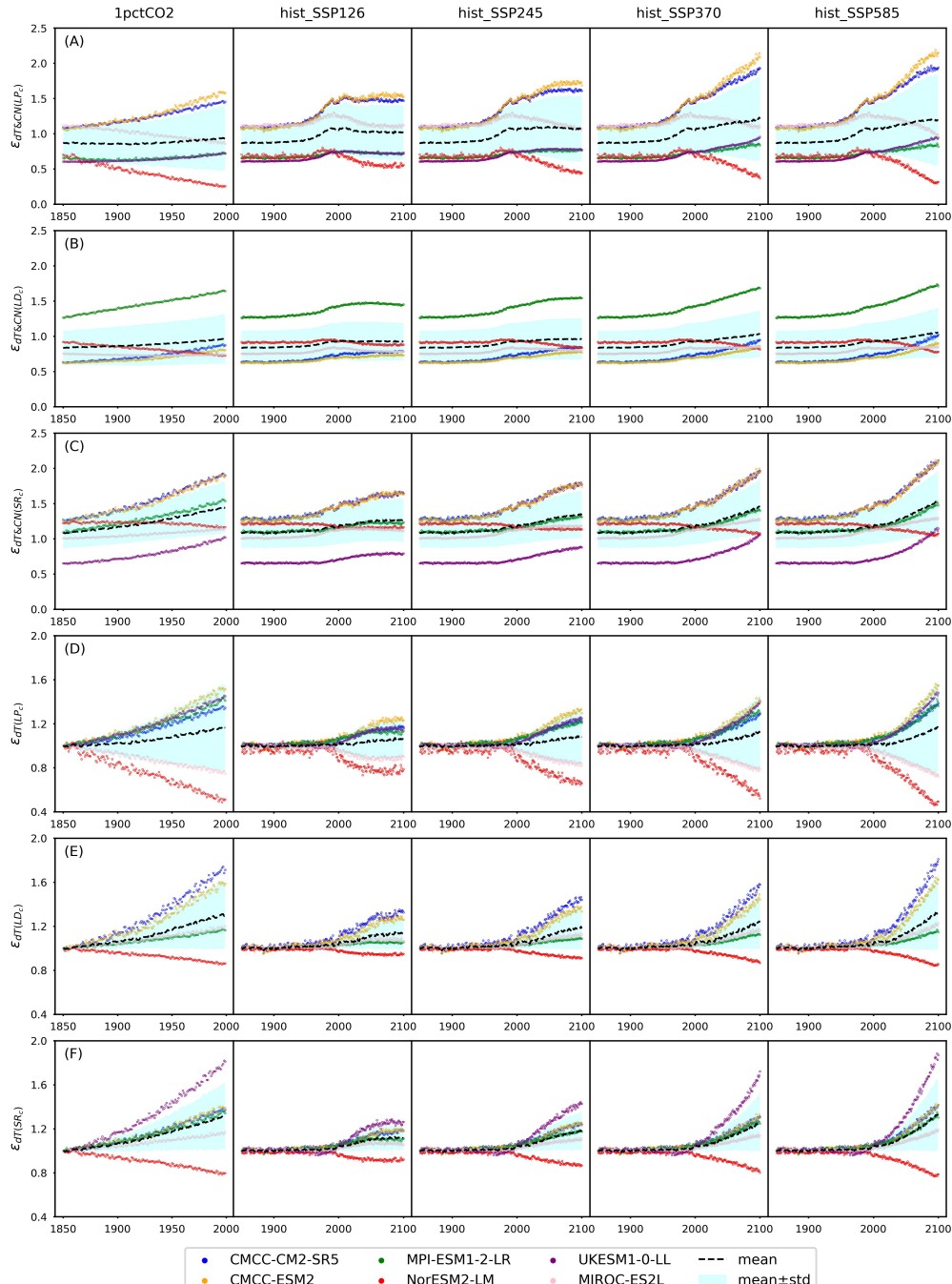

**Figure A10. The emulated overall and temperature effect on carbon pool turnovers, including litter production ($\epsilon_{dT\&CN(LP_c)}$ and $\epsilon_{dT(LP_c)}$), litter decomposition ($\epsilon_{dT\&CN(LD_c)}$ and $\epsilon_{dT(LD_c)}$), and soil respiration ($\epsilon_{dT\&CN(SR_c)}$ and $\epsilon_{dT(SR_c)}$) from CMIP6 ESMs across different scenarios.**

**Code and data availability**

The CNit model code is available at https://doi.org/10.5281/zenodo.12204421 (Tang et al., 2024b). The Python code provided is intended primarily to facilitate the review of its functionality. Comprehensive documentation for the code will be made available in the future, either as part of a standalone Python package or integrated with the MAGICC Fortran code (available at gitlab.com/magicc/magicc). The calibration data is accessible either from the original publications [for CABLE (Fleischer et al., 2019) and OCN (Meyerholt et al., 2020)] or through the Earth System Grid Federation (ESGF, for CMIP6 ESMs), with details provided in Section 3.1 Data acquisition and processing.

**Author contribution**

GT and MM conceptualized the idea of carbon-nitrogen coupling in MAGICC. GT designed, coded, and calibrated the coupled carbon-nitrogen model. ZN, AN, SZ, and MM contributed to the model improvement. ZN assisted with code implementation and optimization. SZ provided the OCN dataset used for model calibration. ZN, SZ, and MM were responsible for funding acquisition. GT analyzed and interpreted the results and wrote the first draft of the manuscript. All authors contributed to writing and revising the manuscript.

**Competing interests**

The authors declare that they have no conflict of interest.

**Acknowledgements**

This work is supported by the Australian National Environmental Science Program (Climate Systems Hub, Malte Meinshausen) and the European Union's Horizon 2020 Research and Innovation Funding Programme (No. 101003536, Earth System Models for the Future, ESM2025, Zebedee Nicholls and Sönke Zaehle).

We express our sincere gratitude to Dr. Peter Rayner from the Superpower Institute and The University of Melbourne, Dr. Chris Jones and Dr. Andrew Wiltshire from the Met Office Hadley Centre, Dr. Ying-Ping Wang and Dr. Tilo Ziehn from the Commonwealth Scientific and Industrial Research Organisation, Dr. Cheng Gong and Dr. Katrin Fleischer from the Max Planck Institute for Biogeochemistry, Dr. Trevor Sloughter and Dr. Bonnie Waring from Imperial College London, and others for their valuable contributions through insightful discussions, collaborations, data sharing, and continued support. Additionally, we extend our thanks to the World Climate Research Programme (WCRP), the Coupled Model Intercomparison Project Phase 6 (CMIP6), the Coupled Climate-Carbon Cycle Model Intercomparison Project (C4MIP), the

Earth system model (ESM) groups, and the Earth System Grid Federation (ESGF) for their collective efforts in making the

ESM output available and accessible.

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
