# Peer review of "Synthesizing Global Carbon-Nitrogen Coupling Effects – the MAGICC Coupled Carbon-Nitrogen Cycle Model v1.0"

_EGUsphere, 2024_

## Referee Comment (RC2)

Referee's report on:

**Coupled Carbon-Nitrogen Cycle in MAGICC v1.0.0: Model Description and Calibration**

This manuscript proposes the description of a new module for a coupled carbon-nitrogen cycle in MAGICC, with its calibration based on land surface models and a set of CMIP6 ESMs. Overall, the model is correctly described and well presented. Its design is common to climate emulators, with pools and fluxes representing the essential elements and processes. This well-established approach does indeed improve simplifications, but with appropriate reasons, and leaving room for improvements or sophistication in future works. The comparisons to the training data show relatively good performances, albeit lower for CMIP6. Besides, adding this CN module to MAGICC would be an important improvement on this key climate emulator, enhancing the robustness of future constraints on the land carbon cycle. **To summarize, this work is then important, timely and relatively well presented, and I consider that it fits perfectly the scope of GMD.**

**Nevertheless, after careful consideration, I would recommend that this manuscript should go through major corrections before publication. The major reasons are the oversimplification on the plant uptake of N (comment 1) and the lack of clarity in the data handling (comment 2).** The first one would require either a better justification or to account more explicitly for the inorganic N pool either in the plant uptake of N or in the limitation on the NPP. The second one is necessary for better understanding by the readers.

There are some additional minor points (comments 3-9), that matter for the overall quality of the manuscript, but are not sufficient to justify a major correction. Finally, some details are simply brought to the attention of the authors (comments 10-13), without requiring any action.

**Major comments:**

**1. Modelling of the plant uptake of N**

The plant uptake is currently a relationship based on the NPP, with a temperature dependency (equations 22, 24 & 27 in Section 2.4). First, a required plant uptake given the NPP is estimated. Then, the limited NPP is estimated. Thus the plant uptake given this limited NPP is obtained. The limitation on the NPP depends only on atmospheric deposition and the required plant uptake.

I was expecting to see either the plant uptake or the limitation to NPP to depend on Nmin, the pool that provides the N. If the limitation on NPP would have depended on Nmin here, it would have affected the actual plant uptake. Yet, Nmin affects neither the limitation of the NPP nor the plant uptake. Having this disconnection between Nmin and the plant uptake or the limitation of NPP could cause inconsistencies, for instance in the following examples:

- For an excessive fertilization, we should expect no limitation from the N cycle on the NPP. Yet, in the current modelling, Nmin would be saturated but it would have no effect either on the

limitation $\epsilon_{CN(NPP)}$ or on the P uptake. Thus excessive Nmin isn't accounted for. It may be the reason for the discrepancy described Lines 457-458.

-   In the current modelling, without fertilization and atmospheric deposition, the only flux going in Nmin is $LD_N 2M$, ie the fraction of the decomposition flux from litter to the inorganic N pool. According to equations (34) and (37), this flux could become very small. Thus, the Nmin pool could be depleted by a continuous plant uptake and an insufficient decomposition flux, and then would become negative. Thus insufficient Nmin isn't accounted for. There may be a link with the issue mentioned in Lines 666-668.

So far, the only explanations are Lines 183-191, mentioning that plant uptake minus net mineralization is linked to the required plant uptake. However, this is insufficient to justify removing the dependency on the size of the inorganic N pool, either in the (required) plant uptake or in the N limitation on the NPP. I checked the papers proposed as sources, but I could not find a justification for this modelling of the plant uptake:

-   Zaehle and Dalmonech, 2011, 10.1016/j.cosust.2011.08.008: The approaches outlined in sections "Nitrogen limitation on plant C uptake" and "Plant nitrogen uptake and competition with soil microbes" insist on the need to represent N availability/limitation, without introducing the approach shown in the reviewed manuscript.
-   Zaehle et al., 2014, 10.1111/nph.12697: none of the equations introduce the equations of the reviewed manuscript. Eqn 4a-c would actually confirm a dependency of the plant uptake flux with Nmin.

This point is crucial for the modelling of the N uptake and NPP limitation, thus I strongly recommend the authors to properly justify this modelling, or adapt it. Adequate sources making use of this modelling or observed relationships would be needed. Additionally, if these equations were still used, biophysical justifications would be needed in my opinion. Finally, discussions on the limits would be needed as well, e.g. in the two cases outlined earlier.

This point is the main reason for switching the manuscript from Minor to Major Correction. If I have missed something, I am sorry. I tripled-checked, but could not find anything to prove me wrong. Therefore even if I were indeed wrong, other readers may also miss the point, and it would be necessary to clarify.

**2. Lack of clarity on data handling**

The Section 3.1 "Data acquisition and processing" isn't clear enough at the moment. CABLE & OCN provided CMIP5 runs on RCPs. Besides, some ESMs provided CMIP6 runs on SSPs. Both sources are used for calibration, hence the analysis in Section 3.4 and further. However, the authors write that:

"Unfortunately, a robust and feedback-specific emulation is not feasible for CMIP6 ESMs, as the results from experiments without the nitrogen effect are unavailable." (lines 363-364)

Apparently, the authors still managed to train and emulate the CN module of MAGICC? Is it about the robust and feedback-specific part on the training & emulation, and the ensuing workaround outlined Section 3.2?

Besides, it remains unclear how data from CABLE & OCN is used for training in comparison to CMIP6. Is it a two-step training, first on CABLE & OCN, then on CMIP6, i.e. using the parameters obtained from CABLE & OCN as a first guess for the optimization on CMIP6 data? Or is it a one-step training, pulling

all samples together? Although the results of the calibration are shown for CABLE & OCN in Section 3.3 and for CMIP6 models in Section 3.4, the Section 3.2 explaining the calibration setup does not mention CABLE or OCN.

I strongly suggest clarifying how both datasets are used precisely, and the questions that I present here. Ideally, a reader should not have to re-read this section to understand the data flow. It may require the authors to adapt the structure of Section 3.1 and 3.2, but it would be worthwhile for the readers.

**Minor comments:**

**3. Land-use & deforestation**

I do appreciate the effort in modelling land use in the C & N cycles, but I have to flag two important limits.

The first one is in the parametrization of the regrowth flux (Lines 313-318). At the moment, the regrowth depends only on the gross deforestation, with two constant parameters $\varphi$ and $\tau_{rgr}$ (equation 40). Yet, I would rather expect the regrowth of a deforested parcel to depend on its NPP rather than how much was deforested. In other words, a primary forest would have a high C stock; deforesting would be a strong C flux because of its past unperturbed growth under a favourable climate; but its regrowth under a less favourable climate would be towards a lower C stock. A potential correction would be to approximate the deforested area using the ratio of $LU_{grsd}$ with $C_P$ (thus neglecting $C_L$ and $C_S$). The regrowth of this area would depend on the NPP, with some parameters to account that the regrowth on a deforested area is not exactly the same than the one aggregated on all biomes as modelled by MAGICC-CN.

Additionally, these fluxes are not only due to deforestation (lines 308-312). Biomass extraction from croplands will also matter a lot, especially for the N cycle. This is an important limit of this current modelling, that must be mentioned.

To be clear, I'm not asking the authors to modify the C-N modelling to account for both effects. I am aware that it would require an extensive work (example with OSCAR as illustrated in Gasser et al, 2017: 10.5194/gmd-10-271-2017). This manuscript already provides a significant modification. However, I suggest to explicitly mention both limits in the manuscript, and keep them for future developments of the model.

**4. Calibration**
**a. Ensemble members**

It is not detailed which ensemble members are used for the calibration of the CN module in Sections 3.1-3.2. Is that only the first one, e.g. *r1i1p1f1*, or all available ones? If all, is there an averaging? I recommend the authors to give some information on these questions.

Additionally, is there some form of weighting on the sample from the samples from the SSPs and the historical, to account for a varying density of points in the space of predictors. To be clear, I'm not asking the authors to apply such a weighting, but I'm asking whether they apply it, and simply suggest to mention it. There are imperfect solutions, like accounting for the length of the runs, and more

sophisticated ones, like the inverse of the density in the predictor space, but such solutions may not be feasible for simple climate models.

**b. **Base period for calibration**

At the moment, the base period is the first year, at least for temperatures (Lines 399-400). Due to internal variability in ESMs runs, I would recommend taking an average over a longer base period, e.g. 1851-1900.

**5. Model design**
**a. **CO2 fertilization**

I appreciate the approach on the CO2 fertilization (Section 2.3.1), especially how to deal with an overreliance on the rectangular hyperbolic function. Yet, I would appreciate having a Figure in appendix showing the response of CO2 fertilization with CO2, for different values of the method factor, e.g. 0, 0.25, etc to 2. It would help the readers get a better idea on the impact of this parametrization.

**b. **Overfitting?**

The model proposed for the CN module is very well designed, I appreciate the representation of the crucial fluxes and pools in a synthetic approach. Yet, the high flexibility in the parametrizations of the fluxes make me wonder about overfitting. For instance, to what extent should the BNF flux be split between the plant, litter and soil pools? Given all fluxes being split, isn't there a risk to have a spurious & non-physical parametrization of the cycles?

To answer these questions, I would have two recommendations. First, the Table A.1 should include the significance of the coefficients, with a discussion in the manuscript. Then, the differences in the N cycles in ESMs could be further discussed, be it for the partitioning or the behaviours. Of course, an exhaustive analysis would make a full paper, but I would suggest to keep it to one paragraph.

**6. Performances for CMIP6?**

In Section 3.2, the authors write that for MIROC-ES2L and UKESM1-0-LL, the NPP over 1pctCO2 is higher than in SSP126, while the opposite is seen for the plant uptake. They conclude in an inconsistency in their modelling. I would argue that it is not necessarily inconsistent for two potential reasons. First, the 1pctCO2 does not assume any change in land management, thus no increase in fertilization, while SSP126 does.

Figure 2 shows good performances for the CN module on CABLE & OCN. For CMIP6 models, Figure 3 shows a more contrasted image. The authors explain issues for instance related to the Nmin pool of the ESMs, but there are still important fluxes that seem not adequately modelled. For instance, MIROC-ES2L exhibit differences on the NPP.

I would be interested in seeing the comparison up to 2300, which is provided for MIROC-ES2L.

**7. Showing the C:N ratio of plants**

I would be curious to see the C:N ratio of the plant pool in SSPs. There is one mention Line 552, but this is for the land, while I would consider the one for the plant pool to have a stronger interpreraton. Current Figure A4 seems to suggest a varying C:N, in particular in 1pctCO2. I would appreciate such a figure in the appendix, if the authors agree that it would provide worthwhile inputs for the manuscript.

**8. Mentioning before the limit on resolutions**

The aggregation to global & annual resolutions is an usual limit of the simple climate models. This is typical from these models, because their model design is not meant to analyse spatial heterogeneity, but rather the Earth system modelling through the interaction of many processes. It should be the first limit reminded in the Section 5, yet it is for now the last point in Section 5.3 (Lines 797-800). These lines do apply to the content of Section 5.3, but it applies as well to Sections 5.1 and 5.2. Thus, it would make sense to mention the issue of resolutions from the beginning of Section 5.

**9. Limits on modelling to mention as potential future works**

In my opinion, the Section 6 "Conclusion and future works" should remind the limits mentioned in Section 5 as potential future works. For instance, the comments in Lines 712-720 clearly suggest that this modelling is just a first step. It is common for simple climate models to be designed that way, to start with a first simple version, and then to sophisticate where necessary. The authors mention oversimplifications, I mention others in comments 1 and 3, such limits can be future works.

**Details:**

**10. Position of Figure 1**

The Figure 1 is crucial to visualize the design of the CN module. It should appear early for the readers to structure its understanding of the model. At the moment, it is only at the very end, in Section 2.7, which is too late.

I strongly suggest shifting Figure 1 to the Section 2.2 for improved clarity.

**11. Difficulties in calibration due to data reporting by ESMs**

I congratulate the authors for acknowledging that, and explaining how. This is a recurring issue in CMIP exercises. Although technical, it does matter a lot for calibration, and it may be useful to raise awareness on this issue.

**12. Code of MAGICC-CN**

The code is well structured, relatively well commented. However, the code of MAGICC v7 itself remains openly but not anonymously available. pymagicc is available for the v6, but not the v7. The requirement for this manuscript is met, with the -CN module provided. However, I would simply

suggest that future versions of MAGICC itself should be openly AND anonymously available. Additionally, development on GitHub would provide an open perspective on the developments on MAGICC and foster collaborations.

**13. RCM vs SCM**

As a simple reminder, the acronym RCM may not necessarily be great for models like MAGICC, FaIR, OSCAR, HECTOR, etc. I acknowledge that we used this acronym for the RCMIP phase 1 & 2 papers, but this choice was criticized by researchers using Regional Climate Models, thus RCMs as well... At some point, the community of climate emulators should decide what to do, RCMs, or SCMs (Simple Climate Models), or else.

---

## Referee Comment (RC3)

**General comments**

This paper fully describes the newly developed coupled carbon-nitrogen cycle to be incorporated into MAGICC, a leading methodology in the reduced-complexity climate model (RCM) category. MAGICC is one of the standard tool for climate assessment of emissions scenarios, and the new component is expected to enhance the tool's functionality and improve the quality of climate assessment. RCMs deal with the global aggregate effects of Earth system responses to given forcing changes based on complex Earth system models. Among them, the nitrogen cycle has not been adequately addressed in RCMs, and this study is the first attempt of its full-scale modeling and coupling with the carbon cycle. Despite limited base data from model experiments and relevant observations, this study conducted calibrations to adjust a number of model parameters to each of target models and validated the performance of emulations.

This study also compares and discusses the behaviors of the target models, considering underlying literature, through calibrated parameters in terms of their evolutions and inter-parameter relationships. This is an interesting analytical examination enabled by the emulator method. The findings are worth feeding back to studies on Earth system modeling, supporting observations, and process understanding.

Thus, the paper is well suitable for publication in GMD. Having said that, the manuscript may need minor revisions for further clarity and usefulness. The followings are my concerns and suggestions to be considered as appropriate.

**Specific comments**

**Main text**

L55–56. Wording of 'smaller feedback' is ambiguous to me. Does it adequately represent the effect of the nitrogen cycle mentioned in the preceding sentence?

L58. This is the first appearance of JSBACH. A brief description should be given to readers unfamiliar with this abbreviation.

L89. Balancing simplicity and performance is one important factor to consider in design. It would also be useful to indicate the extent to which the coupled carbon-nitrogen cycle would involve an increase in computational load and whether the increased parameters would cause any calibration difficulties.

L141. CO2ref definition is redundant because already defined on L137.

L342. Grassi et al. (2023, https://doi.org/10.5194/essd-15-1093-2023) may also be cited on this issue.

Figure 1. Is 'Plant P' correct? I think it is 'Plant C'. Flux partition labeling related to LU is a bit confusing because LU flux directions to the atmosphere are not consistent with those inferred from labeling, which reads 'to plant', 'to litter', or 'to soil' although the text describes the meaning in the end of 2.3. Are there any differences between '2S' between '2S_N'?

L373–391. This paragraph describes the model selection and data processes very well. Is there anything to be added about normalization to eliminate model drift or some biases in the preindustrial control?

L400–402. It seems that the extended period to 2300 applies only to SSP126 and SSP585 of MIROC-ES2L. Do the calibration results depend on the period selection? This concern arises from large differences between the model outputs and emulations in 1pctCO2.

L403. Is 'imputed' a typo?

L407. A paper in preparation is cited.

L411–416. Are all scenario data simultaneously used without weighting in the calibration for each model? This kind of information would be useful.

L438. Probably 'leads to'.

L574. Citing AR6 Chapter 1, specifically Section 1.5.3, is suitable here.

L586–589. It needs a reference of the online calibration. Are Hajima et al. (2020) and Lawrence et al. (2019) suitable references in this context?

L732. It may need 'in low SSP scenarios' after 'NorESM2-LM'.

L741. Citing Meyerholt et al. (2016) is more suitable at the previous sentence.

L763–764. I don't understand how this sampling is enabled from the set of single parameter value for each model. The MCMC sampling may need supporting information.

L796. 'flat10' needs definition.

**Appendix**

Table A1. Missing values in UKESM1-0-LL need explanation.

L897–899. For the land organic nitrogen pool size, the differences between the models are too large to identify the trend from the figure.

Figure A2. It seems that the left panel shows four cases although the legend contains seven cases.

Text A1. This text does not necessarily support the discussion on Figure A5 and may be omitted. I understand that the magnitude of inter-model spread is consistent with the magnitude of forcing changes, and I don't think that 1pctCO2 is special.

L904. Is the description about the initial condition appropriate? I think that it is an issue of ESM spin-up rather than internal variability.

Figure A7. Trivial one values may be omitted for simplicity. Missing values in UKESM1-0-LL need explanation.

**Code and data availability**

To ensure reproducibility, it is recommended that the processed CMIP6 outputs described in 3.1 be included in the data, and that the calibration procedures described in 3.2 be included in the code.

The calibrated data is provided in a Python pickle file, but reading the pickled object seems to require associated modules not provided.

---

## Author Comment (AC1)

**Author Comments (ACs)**

In this Author Comments:

- The original referee comments are in black (directly copied from the comments).

- Our responses are in blue.

- *The text we quoted from the manuscript is in gray italics.*

We sincerely thank all referees for their constructive comments and feedback on our manuscript.

Best regards,

Gang Tang (GT, as referenced below)

on behalf of all co-authors

Top-Level Updates Before Addressing Individual Comments:

- Title Revision:

  The manuscript title has been updated to "Synthesizing Global Carbon-Nitrogen Coupling Effects – the MAGICC Coupled Carbon-Nitrogen Cycle Model v1.0" This new title more accurately reflects the content and scope of the manuscript and is in line with the title used for other modules of MAGICC (e.g., Synthesizing long-term sea level rise projections – the MAGICC sea level model v2.0, https://doi.org/10.5194/gmd-10-2495-2017).

- Terminology Clarification:

  To avoid confusion, we now exclusively use "MAGICC" to refer to the full model (the online model including all components) and "CNit" solely for the coupled carbon-nitrogen cycle model. This eliminates the previous ambiguity caused by the frequent use of "MAGICC" in varying contexts.

**RC1: 'Comment on egusphere-2024-1941', Anonymous Referee #1, 01 Oct 2024**

In general this is an interesting topic and a very valuable effort which I think warrant publication. However, the organization and writing of the paper is quite confusing and could really benefit from a good overhaul, making the paper and the code more useful and reproducible. There is also a lot of discussion on the results, but since this is a model description paper, I would have liked a lot more discussion on the modelling choices. Though some emphasis is put on the limitation of process parametrization and a global approach, the model is still quite complex and little to no time is spent discussing the additional value of this approach against something of lower complexity and with faster computational time. I would very much like to see some discussion, both of how much compute time this module adds to a typical MAGICC run, if some of the complexity could have been shaved off (maybe this is negligible and totally worth it, but from the paper I have no idea). Here is a list of additional questions that I feel should be addressed in a discussion section:

GT: Thank you for reviewing our manuscript and providing positive feedback on the topic and its value. We appreciate your constructive comments and suggestions, which will help improve the paper.

This manuscript, although originally titled as a "model description and calibration," aims to address two key aspects: (1) the model itself and its implementation and (2) the insights into nitrogen effects in ESMs as revealed by the emulation. Initially, we considered splitting these into two separate papers—one focusing on the model description and another on the nitrogen-related findings. However, we decided to combine them into a single, comprehensive paper to provide readers with a cohesive understanding of the subject. We have now revised the title to better align with the manuscript.

Regarding your specific concerns:

Modeling Choices and Complexity: The current model represents a balance between complexity, process representation, and emulation performance, as discussed in the model limitations and conclusion sections. We acknowledge that further discussion on comparisons with lower-complexity parameterizations could be beneficial. However, to balance the manuscript's length and focus, we prioritized results and discussions most relevant to the nitrogen cycle emulation.

Computational Cost: We have not yet tested the additional runtime added by the coupled carbon-nitrogen cycle model (CNit) to MAGICC runs. This is primarily because the CNit has not been implemented into the MAGICC Fortran code yet. However, the additional calculations introduced by CNit are relatively simple, and we expect the impact on runtime to be minimal. Future work will include quantitative assessments of this aspect.

1. The models being emulated show a big spread in their predictions and there is uncertain observational data. Is this the right setting and time to make an emulator of this kind? (The answer might be that this is the perfect time to have such an emulator to explore and push for better data, and be ready.)

GT: Thanks for the comment. Yes, the spread of the ESM carbon-nitrogen cycle and the lack of global observations (especially for nitrogen) are exactly the reasons for the emulator development. The emulator allows us to explore these variations systematically and provides insights that may help inform and potentially constrain the more complex models.

2. The model though simple in a way, is really quite complex with a really large number of parameters. The fact that it can emulate model behaviour relatively well using so many parameters is not very surprising. I think a more important question is whether there is enough input data to constrain such a large parameter space.

GT: Thank you for the comment. It is important to note that this paper also serves as a detailed documentation of the MAGICC carbon cycle. Apart from the nitrogen components and the updated land-use emission treatment, the remainder of the model description is derived from the previous MAGICC carbon cycle model (details available at https://doi.org/10.5194/acp-11-1417-2011). The added nitrogen cycle (consequently the parameters) is trying to find a balance between the reality (i.e., the process representation) and the computational efficiency. The successful emulation of CABLE and OCN, across diverse experiments (e.g., nitrogen on/off, varying temperature/$CO_2$ forcings), demonstrates that the nitrogen-related parameters are well-constrained despite the model's complexity.

3. Context. I imagine this will be used as part of MAGICCs default setup, or at least a fairly accessible version of it. How does this number of parameters add to the current number of free MAGICC parameters? Can they be fit in an online fit? Do they affect other MAGICC parameters? How easy is it to fit MAGICC parameter ensembles with this? What does it add to computational time for the full MAGICC? Etc. I see that you plan to discuss some of these issues in a different paper, but I expect at least some discussion of this here. Also can this module easily be coupled to other simple models? Perhaps the offline use demonstrated in this paper is just as interesting as online use?

GT: Thank you for the insightful questions. At this stage, we have not conducted any online calibration. Theoretically, this could be feasible given the simplicity of MAGICC. However, we generally prefer to calibrate individual modules separately rather than all at once. The primary reason is the lack of sufficient constraints to disentangle the counteracting effects of different modules within the larger model. Moreover, if such constraints were available, the difference between performing online calibration of the entire model and offline calibration of individual modules would be negligible.

The primary interaction between CNit and other components of MAGICC is through the net land-to-atmosphere carbon flux, which influences atmospheric $CO_2$ levels and consequently affects temperature. This temperature change then feeds back into the land carbon-nitrogen cycle, altering the land-to-atmosphere carbon flux. Beyond this interaction, CNit does not directly modify parameters in MAGICC's climate, atmospheric, or ocean modules. We have now included this information in the Section 2.1 Overview of MAGICC and CNit.

Regarding computational time, as noted in my general response, we have not yet explicitly tested the additional time required. However, we have performed coupled experiments and the increase in computational time is not noticeable (MAGICC's current computational time is around a tenth of a second and we don't notice big changes from this). Moreover, incorporating a nitrogen cycle addresses significant limitations in the model, making the added complexity worthwhile.

CNit can also function as a standalone model, requiring only inputs such as $CO_2$ concentration, temperature, land-use emissions, and nitrogen forcings. From this perspective, it should be straightforward to couple with other simple climate models.

To clarify these points further, we have updated the conclusion to explicitly address them. The relevant section, quoted below, discusses future work and the integration of CNit:

*Therefore, the current formulation and treatment of these aspects in MAGICC may have to be updated too, while aiming to continue to strike a balance between model simplicity, process representation, and emulation performance, reflecting a fundamental design principle for RCMs and MAGICC in particular. Future work on MAGICC's carbon-nitrogen cycle will focus on the calibration of the full MAGICC structure*

*to CMIP6 ESMs (and/or observational data), evaluation of model performance with respect to computational efficiency and mechanistic insight, incorporation of additional constraints, uncertainty quantification, sensitivity analysis, application of probabilistic projections, and continued model development (e.g., land use emission implementation and nitrogen process representation) to align with advances in complex models and emerging theoretical frameworks.*

One more overall point before going over the paper from top to bottom: The structure and "story" of the paper is confusing to me. I often don't know where I am and feel like I'm scrambling for an overview when reading it. I think this can be solved without too much work using the following few principles: 1. Spoilers are great. Tell me what is going to happen and give me more of an overview on top, and on top of every section and subsection. Forcing yourself to write such miniature summaries might also help you understand how you've structured your text and see whether it makes sense. Sometimes it might not... 2. Tables are great, long lists in sentences are less so. Tables of models used, input data used to calibrate, parameters to be calibrated, experiments used etc. Also, there are so many variable abbreviations that a dictionary in the supplement would actually be useful. 3. What does MAGICC mean? It is highly unclear when you mean the full model, just this coupled carbon-nitrogen module, the new version of MAGICC the old version of MAGICC? This confusion is present in nearly every reference to the model. Please find a way to distinguish between these three things (MAGICC old version, MAGICC new version and the Coupled Carbon-Nitrogen Cycle model (MAGICC-CCNC?))

GT: Thank you for the comments and suggestions regarding the structure and clarity of the manuscript. In response:

We have added section overviews to improve the logical flow and help guide the reader through the paper. For instance, for the model description section, we now add:

*The following sections outline the key components of CNit: Section 2.2 introduces the mass conservation framework and key fluxes; Section 2.3 details the NPP simulation; Section 2.4 explains carbon-nitrogen coupling, where we link 'nitrogen plant uptake' (PU) and NPP; Section 2.5 describes the litter production respiration flux; Section 2.6 focuses on carbon and nitrogen turnover calculations; and Section 2.7 addresses the implementation of land-use emissions.*

For the model description section, we now add:

*This section presents the offline calibration results for CNit, using prescribed land surface temperature and atmospheric $CO_2$ concentration from the original model outputs. We first describe the data acquisition and post-processing of land surface model outputs and CMIP6 ESM outputs (Section 3.2). Next, we define the calibration targets (major fluxes and pool sizes) and weight them to create a cost function. Finally, we apply optimization algorithms to identify the "best-estimate" parameter set (Section 3.3). For a single model, all experiments are calibrated simultaneously, resulting in one "best-estimate" parameter set that captures the model's behavior across experiments. Using these "best-estimate" parameter values, we evaluate CNit emulation against model outputs and calculate the 'root mean squared error' (RMSE) and normalized RMSE to assess model performance. The discussion of the calibration results for CABLE, OCN, CMIP6 ESMs are provided in Section 3.4 and Section 3.5.*

To enhance clarity, we have incorporated tables where appropriate, such as for the models used and parameters to be calibrated.

We have standardized the terminology to avoid confusion. Specifically:

"MAGICC" now refers exclusively to the online model, representing the full structure with all components.

"CNit" is used solely to refer to the coupled carbon-nitrogen cycle module described in this paper.

Specific comments from text:

Title: Is the version number correct? Is this version number for an upcoming python version of MAGICC? Isn't MAGICC on a much higher version number. This is already confusing, and it shouldn't be.

GT: Thank you for the comment. The version number v1.0.0 specifically refers to the coupled carbon-nitrogen cycle model (CNit) described in this paper, rather than the overall MAGICC model. We have revised the title to: "Synthesizing global carbon-nitrogen coupling effects – the MAGICC coupled carbon-nitrogen cycle model v1.0."

Line 68: "Section 2 presents a detailed descritpion ... in MAGICC". The new version of MAGICC? Also, as far as I can understand it is only the land carbon-nitrogen cycle and carbon-coupling which is described and not how they actually feed in to the wider MAGICC code, though I'd prefer the text in the section to change to reflect that rather than this sentence.

GT: It is only the land carbon-nitrogen cycle model (CNit) itself rather than how it fits into the full MAGICC structure. We have revised it to "*Section 2 presents a detailed description of the CNit model.*" for clarity. We have also added descriptions on how CNit is linked with MAGICC, as quoted below. Technically CNit can be coupled with any model.

*CNit is a globally integrated, annually averaged box model (Fig. 1) designed to simulate terrestrial carbon and nitrogen dynamics. It includes carbon and nitrogen pools for 'plant' (P), 'litter' (L), and 'soil' (S), along with an inorganic 'mineral' (M) nitrogen pool. The 'atmosphere' (A) exchanges carbon with the land carbon pools via 'net primary production' (NPP), 'heterotrophic respiration' (RH), and 'land-use or other anthropogenic fluxes' (LUC). Similarly, the atmosphere exchanges nitrogen with the land nitrogen pools via 'nitrogen atmospheric deposition' (AD), 'biological nitrogen fixation' (BNF), 'gaseous nitrogen loss' (LS2A), and land-use or anthropogenic fluxes LUN. CNit takes the land use emissions of carbon and nitrogen, 'nitrogen fertilizer application' (FT), AD, and BNF, as the inputs. Then, it models key fluxes and solves a system of mass conservation equations to determine the fluxes and states for carbon and nitrogen. The resulting net land-to-atmosphere carbon and nitrogen fluxes are then used to estimate atmospheric concentrations, which subsequently inform radiative forcing and climate responses. These climate responses, in turn, interact with the carbon-nitrogen cycle, creating a feedback loop (see details in Meinshausen et al., 2011a).*

Line 72: "In future work...". Is this specific planned work or just an aspirational statement? Both are fine, but clarity is preferable.

GT: It is specially planned work. Thanks.

Line 75: Add a sentence or two on what the model description will involve. This is a great opportunity to prepare the reader for the overall model structure. Mass balance equations including pools for plants, soil litter and mineral nitrogen with feedbacks from temperature and CO2 etc…

GT: Thank you for the suggestion. We have now added additional description, as quoted below:

*The following sections outline the key components of CNit: Section 2.2 introduces the mass conservation framework and key fluxes; Section 2.3 details the NPP simulation; Section 2.4 explains carbon-nitrogen coupling, where we link nitrogen plant uptake and NPP; Section 2.5 describes the litter production*

*respiration flux; Section 2.6 focuses on carbon and nitrogen turnover calculations; and Section 2.7 addresses the implementation of land-use emissions.*

Line 75/ section 2 overall: The Figure 1 Flowchart shoud appear much earlier, it should be explained, and it should be annotated with equations and sections. It should also be much clearer how the code flows through the various equations. Does each section refer to a method or function or are they not built that way? This is a model description paper, I expect to get some idea about this from reading it.

GT: Thank you for your comments. We have moved Figure 1 to the beginning of the model description section for better clarity. We have also updated Figure 1 to annotate the specific sections. Additionally, we have included descriptions that provide an overview of the model, explain how the code flows through the equations, and describe its integration with the full MAGICC structure. The updated text is quoted below.

The implementation of the code flow may differ slightly from the model description presented here. For example, while the first section of the model description addresses overall mass conservation, the individual calculations prioritize solving for fluxes, which do not always align precisely with the described sections.

*CNit is a globally integrated, annually averaged box model (Fig. 1) designed to simulate terrestrial carbon and nitrogen dynamics. It includes carbon and nitrogen pools for 'plant' (P), 'litter' (L), and 'soil' (S), along with an inorganic 'mineral' (M) nitrogen pool. The 'atmosphere' (A) exchanges carbon with the land carbon pools via 'net primary production' (NPP), 'heterotrophic respiration' (RH), and 'land-use or other anthropogenic fluxes' (LUC). Similarly, the atmosphere exchanges nitrogen with the land nitrogen pools via 'nitrogen atmospheric deposition' (AD), 'biological nitrogen fixation' (BNF), 'gaseous nitrogen loss' (LS2A), and land-use or anthropogenic fluxes LUN. CNit takes the land use emissions of carbon and nitrogen, 'nitrogen fertilizer application' (FT), AD, and BNF, as the inputs. Then, it models key fluxes and solves a system of mass conservation equations to determine the fluxes and states for carbon and nitrogen. The resulting net land-to-atmosphere carbon and nitrogen fluxes are then used to estimate atmospheric concentrations, which subsequently inform radiative forcing and climate responses. These climate responses, in turn, interact with the carbon-nitrogen cycle, creating a feedback loop (see details in Meinshausen et al., 2011a).*

Line 76: I would like to see a flowchart of the workings of MAGICC which shows me where the new coupled carbon-nitrogen model fits in. Also this section does very little in the way of giving me an overview of the workings of MAGICC, a sentence or two to explain a flowchart would really improve this.

GT: Thank you for the comments. While CNit is designed to integrate into MAGICC, this paper primarily focuses on the carbon-nitrogen cycle itself. As explained above, the net land-to-atmosphere carbon flux from the CNit and the temperature response of the carbon-nitrogen cycle are the key interactions between CNit and MAGICC's other modules. The description added (please see the above quotes) should make this clear.

Line 85: "intial design..." give some reference to equations and sections coming up that the describe the design that you landed on, if I want to flip over and have a look from here. Also since this is an update to MAGICC, what did MAGICC have to treat this before? A sentence on that would be helpful before the initial design of this model.

GT: Thank you for the feedback. This line serves as a disclaimer about the model design principles of MAGICC (and CNit). We have added examples to clarify these principles, as quoted below. Please note there is no nitrogen cycle in MAGICC before, which is the

*However, during model parameterization and refinement, some processes were simplified or integrated with others to improve efficiency. For instance, biological nitrogen fixation is directly allocated to organic nitrogen pools, bypassing the intermediate step of mineral nitrogen enrichment and subsequent plant uptake (Fig. 1). Additionally, certain representations, such as land-use emissions, were updated to achieve a balance between model simplicity and mechanistic insight (Section 2.7). These refinements align with MAGICC's design philosophy of being as simple as possible but as mechanistic as necessary.*

Lines 95-110: Please be much more explicit about which equation or equations you are explaining when. Also referring to the different parts of the flowchart of figure 1 might be helpful. Again I'd very much like the supplement to include a vocabulary for reference as the amount of shorthand used is daunting. Also giving more overview would be helpful. For instance is the sentence from line 99 to 100 describing equation 4? Is equation 4 a sum of equations 1 to 3? Is equation 9 the sum of equations 5-8? Is the sentence from line 104 to 105 explaining equation 9?

GT: Thank you for pointing this out, and we apologize for any confusion caused by the writing. We have now reorganized the equations and revised the descriptions to improve clarity - now equations are always accompanied by their descriptions. Regarding the vocabulary table, we have rechecked the manuscript to ensure that all abbreviations are introduced in full the first time they appear in a new section or figure (including figure captions). This work introduces four new abbreviations—LP (litter production), LD (litter decomposition), SR (soil respiration), and LS (mineral nitrogen loss)—which are essential for formulating the equations. All other abbreviations refer to commonly used variables. The updated Table A1 provides the full names of all parameters, linking abbreviations to their complete forms, effectively serving as a vocabulary table.

Lines 110-123: Can the equations have an explanatory text to the side which would make them easier to come back to and review, such as Plant, Litter, Soil, Atmosphere to land?

GT: Thanks for the suggestion. The whole section is now revised to make it clear. Please see the Section 2.2 Carbon and nitrogen mass conservation in CNit.

Lines 162-165: In my opinion this should be a single equation with two domains or something like that \epsilon_CO2 = { and then two lines with different conditions. I find that easier to read.

GT: Thanks for the suggestion. I have revised the equations.

Line 172: That is a mouthful... A table of free parameters with a longer descriptive name/ explanation in the supplement would be a really good thing. Perhaps with possible ranges and ranges in the calibration set? Mean + std for ESMs and actual values in the OCN and CABLE calibrations?

GT: Thanks for the comment. The Table A1 (now Table A2) lists all the parameters and their values for the calibration, which is necessary because we referenced the values in the discussions. The descriptive long name and range are now added as a new table (new Table A1, as it comes before the model calibration results).

Line 226-228: I would like a reference either external or internal (i.e. to a specific upcoming section/figure) for this statement.

GT: We have now added a reference figure to the appendix. Thanks.

[Figure]

*Figure A2. Relationship between nitrogen plant uptake and net mineralization as simulated by CABLE and OCN.*

Lines 264-265: I assume recurring parameters are the same as presented before, but if there was a supplemental table to peruse, that would be much easier to understand.

GT: Thank you for your comment. The reason we used the full descriptive names here is exactly as you mentioned—to enhance readability. While we have now added a reference table for parameter names and ranges in the supplement (the new Table A1), we do not expect all readers to rely on the table while reading. Therefore, we retain the practice of repeating the full name and description at the start of a new section.

Lines 277-282: To me these look like they are more or less the same equation. That could be used to explain it in a schematic way as in equation 37, and these individual equations could be moved to the supplement. Even equation 36 is the same equation, only it doesn't include a carbon-nitrogen coupling because there is no corresponding carbon mineral pool. This also becomes such a long list of independent parameters, where a table would be useful, and I would like a discussion on whether so many separate parameters can be meaningfully calibrated from the available data (maybe they can and it's no big deal, then tell me).

GT: Thank you for the feedback. We have chosen not to include equations in the appendix because the model is not overly complex. While these equations are similar, they describe turnover processes for different pools, each with distinct turnover times. We will revise it soon for simplification.

As mentioned earlier, the model design principle is to keep it as simple as possible while remaining as mechanistic as necessary. From a mechanistic modeling perspective, I do not find the formulations overly complex; in fact, they might even be too simple. The temperature response is clearly important. I assume we all agree on the necessity of the carbon-nitrogen coupling effect for the turnovers, otherwise complex models would not model them and and we would not develop the emulator for them. If your concern is about the nitrogen effect in the emulator, I believe it is also critical. Without it, we risk misattributing such effects to other factors, such as the temperature response. The emulator should retain the ability to disentangle different effects, just as complex models do. Specifically regarding the nitrogen effect, it is unrealistic to require all the complex models to perform nitrogen-off experiments and explore the nitrogen effect due to the high computational cost. Therefore, using an emulator to at least quantify the effect—despite the uncertainty arising from data limitations (as discussed in our limitations section)—becomes critical.

Regarding the necessity of these free parameters and whether the data can constrain them, I do not expect all turnover processes to have the same sensitivity to plant nitrogen states or atmospheric nitrogen deposition. Since this is a global box model, nitrogen effects may behave more diversely. For instance,

using plant uptake as a single proxy for nitrogen status may not be sufficient—other factors, such as plant carbon:nitrogen stoichiometry, may also play significant roles. However, as an emulator, CNit aims to provide strong emulation performance, as shown by the results, while maintaining relative simplicity. Thus, the emulator's performance justifies these formulations to some extent. The data we have, which includes experiments with nitrogen on/off from various models and scenarios, covers a range of carbon-nitrogen cycle behaviors. This allows us to disentangle the nitrogen effect and calibrate the nitrogen parameters effectively. The data provides constraints for these parameters.

Line 289: "PU and atmospheric deposition" this statement mixes a shorthand and a full writing out of something that has a similar shorthand. Please do one or the other, at least in a single term like this. With a supplementary vocabulary, I think using the shorthands systematically would make a lot of sense. When you mix terms like this I also get confused on how to interpret "atmospheric deposition", does it now mean something else than the "atmospheric deposition = AD" shorthand? What would that be?

GT: Apologies for the confusion. We have revised the text accordingly (quoted below). As mentioned earlier, we prefer to introduce abbreviations at the beginning of each section rather than using them throughout the paper, as this approach is generally clearer for readers.

*The carbon-nitrogen coupling feedback takes current 'nitrogen plant uptake' (PU) and 'nitrogen atmospheric deposition' (AD) as proxies to represent the plant nitrogen status and the nitrogen forcing, respectively.*

Figure 1: I'd like to reiterate that this figure should come sooner, and that the caption, and possibly the arrows themselves could do with references to sections and equations that they depict.

GT: Thanks a lot for the suggestion. We have now moved the figure to the overview of CNit.

Line 352: Before you start the subsections, give an overview of the calibration process and datasets involved so I know what I am getting into in this section.

GT: Thanks for the comments. We have now added an overview of the calibration section, as quoted below:

*3.1 Overview of the calibration process and results*

*This section presents the offline calibration results for CNit, using prescribed land surface temperature and atmospheric $CO_2$ concentration from the original model outputs. We first describe the data acquisition and post-processing of land surface model outputs and CMIP6 ESM outputs (Section 3.2). Next, we define the calibration targets (major fluxes and pool sizes) and weight them to create a cost function. Finally, we apply optimization algorithms to identify the "best-fitting" parameter set (Section 3.3). For a single model, all experiments are calibrated simultaneously, resulting in one "best-fitting" parameter set that captures the model's behavior across experiments. Using these "best-estimate" parameter values, we evaluate CNit emulation against model outputs and calculate the 'root mean squared error' (RMSE) and normalized RMSE to assess model performance. The discussion of the calibration results for CABLE, OCN, CMIP6 ESMs are provided in Section 3.4 and Section 3.5.*

Lines 355-360: Are these datasets publicly available? Where can I get them? If I wanted to reproduce your paper how would I do that? This should be very clear either here or in the data availability statement. I find it in neither.

GT: Thanks. We have cited the related papers that can direct the readers to the original data source. We have now clarified this in the data availability section.

Line 364-363: "as results from experiments without the nitrogen effect are unavailable". Is this a wish for future experiments? How useful would these be? How could they aid in work similar to this? I think this might be worth revisiting in the outlook section.

GT: Thanks for the comments. This is a wish from the emulator's perspective. The only way to get a robust emulated nitrogen effect is to calibrate the CMIP6 EMSs with the nitrogen off experimental data - which is, unfortunately, not available for the current ESMs. We hope the modeling groups will run such experiments in the future but we also understand that this might be too computational demanding. We have discussed this in the limitation part Section 5.3 The disentangled climate feedback and nitrogen effect from emulation, as quoted below:

However, the absence of nitrogen-off simulations from the CMIP6 ESMs presents challenges for direct verification. Given the computational expense of running all scenarios in nitrogen-off mode, it is recommended that ESMs perform nitrogen-off simulations for select idealized scenarios (e.g., 1pctCO2 or flat10) for diagnostic purposes.

Line 369-370: I suspect the reference is not in the right place here "calibration in this paper" refers to the current paper I am reviewing while "the original paper" is Meyerholt et al 2020. Please move it if my assumption is correct.

GT: Thanks for the reminder. Yes it should be moved to "the original paper" and now it is moved.

Line 381-384: This long listing of models, land models and references are perfect for a table. They are considerable less perfect for this sentence format.

GT: Thanks for the suggestion. We agree with it and now a table (Table 1) is added to the manuscript.

Line 392-417: This section left me with a lot of questions: 1. What experiments are used for the calibration? All? Or do you reserve some for testing? Which parameters are you fitting and which variables are you fitting to (I'd like a table or two for that in the supplement)? How do you weight the different variables when you fit? Do you fit entire timeseries? Again how do you measure the fitness of a run? Do you scale them? You say offline calibration, please define? In fact, is anything in this paper online? I could not reproduce your procedure from this text.

GT: We are sorry about the confusion of the calibration section. We have rewritten the whole section for clarifying the details. Short answers for the questions here: All experiments are used for calibration; the variables (targets) include NPP, heterotrophic respiration, nitrogen plant uptake, and carbon and nitrogen pool sizes - the entire time series; All the parameters are fitted; we defined cost function (with normalization of errors for different variables) for the optimization; The offline calibration means the CNit is not connected to the climate module or atmosphere module (i.e., temperature and $CO_2$ concentration are prescribed). There is nothing online in this paper. Thanks.

Line 403: I assume "imputed" should be "inputed"

GT: Thanks. We have checked and revised the typo.

Line 412-416: These three sentences read to me like a three step process. Numbering it as such might aid the understanding.

GT: Thanks. We have added "first, next, and finally" to indicate it is a step-by-step process.

Line 419- 463: I am very confused by this section. It reads mainly as a discussion of differences between CABLE and OCN with occasional references to MAGICC (which MAGICC, BTW? the offline CCNC I presume, but I would bet on it...) being able to capture them. If this is calibration results maybe this makes sense, but then the headline should reflect that. Also some summary table of how well the fits are doing would make sense to have here. It is also not clear to me whether you've used the same calibration for all experiments (I think so, but I shouldn't have to figure that out from gathering scattered clues in the text...)

Lines 472- 528: This section too seems to be about calibration results rather than the detailing the calibration. I also expect some sort of error or performance summaries. I am also still confused about what MAGICC we are talking about when and whether the results are all "offline" (whatever that means) and if so why no "online" results are included. Maybe they wouldn't change anything? Maybe it's not ready for that? I'm fine with either, but make it clear.

GT: Thanks for the comments. Both sections are indeed the calibration results - calibrating CNit to CABLE/OCN/CMIP6 ESMs. We are sorry for the confusion. We have now clarified this in the subtitles as well as in the section overview. Now the "MAGICC" is removed following our rules that MAGICC refers only to the full model structure coupled with climate, land, ocean, etc (i.e., the online model).

The new subtitles:

*3.4 Calibrating CNit to CABLE and OCN: Results and comparison*

*3.5 Calibrating CNit to CMIP6 ESMs: Results and comparison*

All the results are from the offline calibration, which is clarified in the revised calibration section. The calibration performance is presented at the beginning of each section - we quantified the error of each target variable by RMSE and normalized RMSE.

There are no online results provided. From our perspective those results would be part of a different study, which is not ready yet.

Lines 485-500: This discussion on mismatching is interesting, but I'd wish you'd take the discussion a bit further, to raise questions like: Does this mean you are perhaps fitting to the "wrong" variables? Are the data/observations/models to uncertain for the type of exercise you've done here (that doesn't mean what has been done isn't very useful, in fact in my opinion it might make it all the more interesting in fact)? Is your underlaying parametrization what has an issue? Also, some of this could have been discussed before the calibration and when you did your data selection.

GT: Thanks for the comments. When having noticeable mismatch between emulation and model output, we primarily focus on the potential causes from our formulation part (the discussion following Line 480). We also discussed the strange relationship of NPP and PU in UKESM and MIROC - higher NPP but lower PU

in 1pctCO2 (the discussion before the Line 480). However, this cannot lead to a conclusion that these models or model outputs are too uncertain because it is only found for the 1pctCO2 run.

Lines 530-536: There is so much going on in the axis here that I had trouble initially understanding the x- and y-axis setup here. Please make that clearer also in the caption.

Lines 534-535: "Diagonal dashed lines represented points where the emulation equals the target". The use of the past tense here is confusing, maybe also rewrite overall "Diagonal dashed lines represent the line where model and emulation are the same" or something like that.

GT: Thanks for the comments. Now we have added the specific explanation for the axis and the tense is changed. The changes are quoted below.

*Results are normalized to a range of 0-1 using the following transformation: x-axis = (target – target$_{min}$) / (target$_{max}$ – target$_{min}$), y-axis = (emulation – target$_{min}$) / (target$_{max}$ – target$_{min}$). The diagonal dashed line represents points where the emulation matches the target exactly, with positions below and above the line indicating underestimation and overestimation by the emulator, respectively.*

Line 437: Is this discussion or results? What are we discussing? Please give me a few sentences of summary of what discussion are upcoming, and maybe think about whether they are really discussions or just results. (Results are fine too, but they shouldn't be called discussions)

GT: Thanks for the comments. We started by pointing out the nitrogen limitation on NPP is marginal in OCN and we concluded that this minor NPP limitation still leads to considerable change in the land carbon storage. It is a result from the OCN data. Now the revised subtitle should be clear about this.

Line 539: "remain considerably different" from what? Each other? Reality? The emulation here?

GT: Now clarified with "from each other". Thanks.

Lines 549-551: I smell a table here... Also are these model means and spread? Again, this could be more easily communicated in a table.

GT: Thanks. We have claimed at the beginning of the discussion section "If not specified, the value and spread in the discussion are expressed as mean ± one standard deviation across ESMs." Table might be unnecessary here as these values are only shown once.

Line 559: I think you should probably drop "is" here.

GT: Thanks for the comments. We did not find any grammar issues with the "is" here, but we revised the sentence structure a bit for readability.

*The continuous and rapid depletion of mineral nitrogen is observed in NorESM2-LM under the 1pctCO2 scenario, coinciding with the highest accumulation of organic nitrogen (Fig. 4C).*

Line 589: This would be a great please to discuss whether the model presented here is perhaps too complex. Maybe it isn't, but you've done nothing to convince me.

GT: Thanks for the comments. From my perspective the discussion here emphasizes the uncertainty and complexity of the carbon cycle and nitrogen cycle. This may indicate CNit might be too simple for emulating the system rather than too complex. In emulator development, we are trying to use the least complexity model to emulate the complex system - with the consideration of necessary biophysic processes. The latter is critical because we are not only craving model simplicity but also seeking the explanation of the processes. For instance, when designing the carbon-nitrogen coupling effect for carbon turnovers [turnover = pool size / turnover time * effect (temperature) * effect (carbon-nitrogen coupling)], we inevitably introduce new parameters. Without it, the model [turnover = pool size / turnover time * effect (temperature)] may still be calibrated, but then it effectively means we misattribute the nitrogen effect on turnover to temperature effect and/or turnover time. We would like to quantify how this kind of lack of process representation can lead to the misattribution (planning work), but that would be another study. We do agree that, overall, the question of whether the model is too complex or too simple is a valid one. We also think that, given the lack of available data, it is challenging to provide a highly confident answer. For simple models, the key point is always to strike a balance between computational simplicity and mechanistic insight.

Line 620-621: Is this emulation also consistent with this for CLM?

GT: Yes, it is (as evidenced that the NPP in CMCC models is well emulated).

Line 590-636: This whole discussion is interesting, but it is not entirely clear whether the results discussed includes information gleamed specifically from the emulation or whether it is just model comparison which you could do without it. If it is some mix of those it should be clearer what insights have actually arisen from the emulation.

GT: Sorry for the confusion. The discussion is specifically based on the emulation because no ESMs run the nitrogen-off experiments (i.e., direct nitrogen effect from the ESMs is not available). We added "based on our calibration" at the beginning of all the discussion when we started talking about the findings from the emulation. The discussion here is first to present what the nitrogen effect on NPP is and the difference between different ESMs; followed by the explanation with either our emulation results or the intrinsic differences between complex models to support the emulated nitrogen effect differences. Finally we discuss the potential limitations from our emulation. We have thought about renaming the subtitle to "4.3 The emulated nitrogen effect on NPP". However, as this section mixes the emulation and model comparison, we think the original might be more suitable.

The insight from the emulation results is as follows: Given that ESMs cannot afford comprehensive nitrogen-on/off experiments to isolate the nitrogen effect, we currently have no clear understanding of the nitrogen effect in the model. However, the emulation, which demonstrates good performance, can at least provide a quantitative sense of "what the nitrogen effect is and how much difference it can cause."

Line 691: I want a summary of what's to come... Somewhere here I also want a real and open discussion on whether this level of complexity is the right one. How important is it versus how expensive is it? How well do we believe in the tuned parameters? Are there too many tuned parameters? Are some of the parameters very constrained overall (i.e. maybe they don't need to be free)? Are some all over the place (completely unconstrained)?  Also I'm not even sure if this model needs specific inputs or if it can be run online with MAGICC for something like an AR6 scenario database member? Has it been tested with online MAGICC? How fast is it? How compute intensive was the calibration? For a future online run with this for impacts, what would you recommend calibrating to? All of these are questions that you don't have to have an answer to, but I expect you to acknowledge them, and say something about how they may be addressed in the future.

GT: Thanks for the comments. As explained in many of the previous replies, one key point I want to reiterate to answer your questions here is: The current model is using relatively the lowest complexity to model the key processes and effects, given that we already know there should be a "nitrogen effect" to nearly all the processes and we want to represent that in our model. The calibrations with nitrogen-off experiments provide direct constraints to the nitrogen effect. The analysis of independence of the climate effect and nitrogen effect (Section 5.3 The disentangled climate feedback and nitrogen effect from emulation) further evidenced that our parameterization is reasonable.

CNit does require further inputs like nitrogen deposition, fertilizer use and biological nitrogen fixation. We have now explicitly explained this in the model overview and calibration section (full texts quoted before so below we only quoted the key sentence).

*CNit takes the land use emissions of carbon and nitrogen, 'nitrogen fertilizer application' (FT), AD, and BNF, as the inputs.*

Not all the scenarios in the AR6 scenario database provided all of the inputs, which means some further assumptions or data sources will be used for the scenario exploitation - it would be another study. We have tested the CNit with MAGICC full structure for the 1pctCO2 and SSP scenarios (simply for testing the functionality so far), there is no noticeable change in speed as a result of using CNit.

We have now covered the above in the conclusion and outlook of the work.

Line 748: Will this  prescribing of biological nitrogen fixation make the model less easy to run "in the wild" so to speak, with just scenario information online or offline?

GT: Thanks for the comments. Short answer to this question: yes it adds up to difficulties for the scenario use. However, our justification of not modeling it is that it already shows large differences within the same SSP scenario family, so modeling it might be challenging. In complex models (like CABLE), it is common to prescribe a constant biological nitrogen fixation. The magnitude of this flux is not that significant so even a constant assumption is an ok starting point.

We assume this is a major limitation of the current model. But as mentioned in the outlook, the modeling of biological fixation is a planned next step, which we are currently working on.

Figure A1: Maybe state explicitly that this figure is like figure 2 and what the difference between them are. Also the plots here are so close that ylabels go into each other making them hard to read.

Figure A4: Again maybe like figure 2 but with differences blahblah..

GT: Figure A1 and A4 provides the full variables calibrated to supplement Figure 2 (with only the key fluxes and pool sizes plotted). We have now revised the figure to make the y label clearer.

Lines 859-887: Text A1 - It is entirely unclear to me what this section has to do with this model and its calibration.

GT: For offline calibration the temperature is prescribed (now explicitly explained in the model calibration section). Thus, the different temperature is part of the reasons for the difference in carbon-nitrogen cycle response. We present the temperature output from CMIP6 ESMs and discuss the uncertainties in Text A1. We have now changed the title to "Text A1. The diversity of temperature output from CMIP6 ESMs", which should make it clear. Thanks.

Line 879: indented -> intended

GT: We have revised the typo. Thanks a lot for checking.

Line 904: model's or models'? (It says the former, I think maybe you mean the latter?)

GT: We have revised the typo. Thanks a lot for checking.

Line 925-928: Be more clear and explicit here, especially regarding the CABLE and OCN datasets.

GT: We have now cited the original publication for the data source. Thanks.

*The model code is available at https://doi.org/10.5281/zenodo.12204422 (Tang et al., 2024). The calibration data is accessible either from the original publications [for CABLE (Fleischer et al., 2019) and OCN (Meyerholt et al., 2020)] or through the Earth System Grid Federation (ESGF, for CMIP6 ESMs), with details provided in Section 3.1 Data acquisition and processing.*

I have had a look at the code, though not in great detail. In an ideal world, I'd like to see it as an importable library function, but I guess the reason why it isn't is that it will be part of a not yet publicly available MAGICC codebase. Maybe say that explicitly if that is the case. In an ideal world I'd also like more function doc-strings.

GT: Thanks for the comments, they are good questions to ask. The way we think about this is this: in this study, we are checking that the model's form and behaviour is sensible. We are testing this with a purely Python codebase for ease of exploration. Once we are happy with the setup, we will invest the time to port this to MAGICC's Fortran code base, and will move the code up to reusable, library-compatible standards at that point. We understand why having it as a library now would be helpful, but we have chosen a different order of effort for the reasons described above. Please keep an eye on the MAGICC repository to keep track of this porting activity.

The Python code provided for CNit is intended primarily to facilitate the review of its functionality. Comprehensive documentation for the code will be made available in the future, either as part of a standalone Python package or integrated with the Fortran code. We have now made this explicitly in the paper, which is quoted below:

*The CNit model code is available at https://doi.org/10.5281/zenodo.12204422 (Tang et al., 2024). The Python code provided is intended primarily to facilitate the review of its functionality. Comprehensive documentation for the code will be made available in the future, either as part of a standalone Python package or integrated with the MAGICC Fortran code.*

---

## Author Comment (AC2)

**Author Comments (ACs)**

In this Author Comments:

- The original referee comments are in black (directly copied from the comments).

- Our responses are in blue.

- *The text we quoted from the manuscript is in gray italics.*

We sincerely thank all referees for their constructive comments and feedback on our manuscript.

Best regards,

Gang Tang (GT, as referenced below)

on behalf of all co-authors

Top-Level Updates Before Addressing Individual Comments:

- Title Revision:

  The manuscript title has been updated to "Synthesizing Global Carbon-Nitrogen Coupling Effects – the MAGICC Coupled Carbon-Nitrogen Cycle Model v1.0" This new title more accurately reflects the content and scope of the manuscript and is in line with the title used for other modules of MAGICC (e.g., Synthesizing long-term sea level rise projections – the MAGICC sea level model v2.0, https://doi.org/10.5194/gmd-10-2495-2017).

- Terminology Clarification:

  To avoid confusion, we now exclusively use "MAGICC" to refer to the full model (the online model including all components) and "CNit" solely for the coupled carbon-nitrogen cycle model. This eliminates the previous ambiguity caused by the frequent use of "MAGICC" in varying contexts.

**RC3: 'Comment on egusphere-2024-1941',    Junichi Tsutsui, 24 Oct 2024**

General comments

This paper fully describes the newly developed coupled carbon-nitrogen cycle to be incorporated into MAGICC, a leading methodology in the reduced-complexity climate model (RCM) category. MAGICC is one of the standard tool for climate assessment of emissions scenarios, and the new component is expected to enhance the tool's functionality and improve the quality of climate assessment. RCMs deal with the global aggregate effects of Earth system responses to given forcing changes based on complex Earth system models. Among them, the nitrogen cycle has not been adequately addressed in RCMs, and this study is the first attempt of its full-scale modeling and coupling with the carbon cycle. Despite limited base data from model experiments and relevant observations, this study conducted calibrations to adjust a number of model parameters to each of target models and validated the performance of emulations.

This study also compares and discusses the behaviors of the target models, considering underlying literature, through calibrated parameters in terms of their evolutions and inter-parameter relationships. This is an interesting analytical examination enabled by the emulator method. The findings are worth feeding back to studies on Earth system modeling, supporting observations, and process understanding.

Thus, the paper is well suitable for publication in GMD. Having said that, the manuscript may need minor revisions for further clarity and usefulness. The followings are my concerns and suggestions to be considered as appropriate.

GT: Thanks a lot for your reviewing and the feedback. We have now revised the paper based on the comments provided. Please see our response below.

Specific comments

Main text

L55–56. Wording of 'smaller feedback' is ambiguous to me. Does it adequately represent the effect of the nitrogen cycle mentioned in the preceding sentence?

GT: Thanks. We have now revised the sentence to make it clear.

*On average, the carbon-nitrogen coupled ESMs have smaller carbon-concentration feedback and smaller carbon-climate feedback (weaker absolute strength of the feedback parameters) compared to their carbon-only counterparts (Arora et al. 2020).*

L58. This is the first appearance of JSBACH. A brief description should be given to readers unfamiliar with this abbreviation.

GT: Thank you. The full name is now added, as quoted below

*"Jena Scheme for Biosphere-Atmosphere Coupling in Hamburg" (JSBACH)*

L89. Balancing simplicity and performance is one important factor to consider in design. It would also be useful to indicate the extent to which the coupled carbon-nitrogen cycle would involve an increase in computational load and whether the increased parameters would cause any calibration difficulties.

GT: Thanks for the comment. In early ports of the model to Fortran, we do not detect notable changes in code performance. The exact extra computational need is hard to compare with the previous MAGICC carbon cycle as now we are writing python rather than directly putting it within the MAGICC Fortran. There are also many code updates in the MAGICC Fortran now. Since this paper we are primarily focusing on the model introduction, we did not mingle it with the code updates in the MAGICC Fortran (thus not the speed comparison). Also, the added parameters are necessary to disentangle the temperature response from the carbon-nitrogen coupling effect. The current model basically uses the least number of parameters to realize this. In addition, the model is still simple from our perspective. It should not burden the computation.

As for the difficulties for the calibration. The root problem is more like whether we have enough constraints to constrain all these parameters. For the land surface models with nitrogen-off and -on runs, the nitrogen effect should be well constrained as evidenced by the calibration results. For the CMIP6 ESMs without nitrogen-off runs, it would be challenging as there are no direct constraints. Thus we have discussed the limitation of the emulated nitrogen effect in Section 5.3 The disentangled climate feedback and nitrogen effect from emulation. The results support that the parameters are independent to each other - in another word, they are constrained in the calibration.

L141. CO2ref definition is redundant because already defined on L137.

GT: Thanks for checking. We have now removed the duplicates.

L342. Grassi et al. (2023, https://doi.org/10.5194/essd-15-1093-2023) may also be cited on this issue.

GT: Thanks for the suggestion. It is well related and we have added the citation now.

Figure 1. Is 'Plant P' correct? I think it is 'Plant C'. Flux partition labeling related to LU is a bit confusing because LU flux directions to the atmosphere are not consistent with those inferred from labeling, which reads 'to plant', 'to litter', or 'to soil' although the text describes the meaning in the end of 2.3. Are there any differences between '2S' between '2S_N'?

GT: Thanks a lot for checking. The "Plant P" should be "Plant C". The "_N" in the land use nitrogen is redundant. We have now cleared the typos.

Regarding the LU, the 2P/2L/2S does not necessarily mean "into" plant/litter/soil. Instead, it is just a partition of flux x to plant/litter/soil, either entering (e.g., NPP) or leaving (e.g., LU) the pools. We have thought about $LU_cP$, $LU_cL$, $LU_cS$, but without a number in the middle, it looks a bit strange. We have now made it clear in the caption.

L373–391. This paragraph describes the model selection and data processes very well. Is there anything to be added about normalization to eliminate model drift or some biases in the preindustrial control?

GT: Thanks for the comments. We have now rewritten the whole model calibration section to provide more details. We did not apply normalization for the starting year. Instead, initial states are directly taken from the data.

L400–402. It seems that the extended period to 2300 applies only to SSP126 and SSP585 of MIROC-ES2L. Do the calibration results depend on the period selection? This concern arises from large differences between the model outputs and emulations in 1pctCO2.

GT: Thanks for the comments. Only the SSP126 and SSP585 experiments in MIROC-ES2L provided the -2300 data, as now detailed in the calibration section.

For a single model, all experiments are calibrated simultaneously, resulting in one "best-estimate" parameter set that captures the model's behavior across experiments.

That means the calibration is not dependent on the experiment/period selection. The difference between the model outputs and emulations in 1pctCO2 is primarily because our formulation assumption - higher NPP needs higher nitrogen plant uptake - is conflicting with the 1pctCO2 outputs in MIROC (also in UKESM). We have discussed this in Section 3.5.

L403. Is 'imputed' a typo?

GT: Thanks for checking. Now the typos are cleared.

L407. A paper in preparation is cited.

GT: Thanks for checking. At the time of the draft writing, that work is still in preparation. Now we have added the preprint citation.

L411–416. Are all scenario data simultaneously used without weighting in the calibration for each model? This kind of information would be useful.

GT: Thanks for the comments. Such details are added now, which are quoted below:

*3.3.3 Calibration target and optimization*

*Calibration targets for both land surface models and CMIP6 ESMs included NPP, heterotrophic respiration, nitrogen plant uptake, and all carbon and nitrogen pool sizes. The cost function was calculated as the sum of normalized errors for each target flux or pool size timeseries [i.e., square (emulation – target) / (targetmax – targetmin)]. This normalization accounted for the differing magnitudes among target variables. All available experiments were calibrated simultaneously without additional weighting, meaning the final cost was calculated as the sum of the costs from all experiments.*

L438. Probably 'leads to'.

GT: Revised, thanks.

L574. Citing AR6 Chapter 1, specifically Section 1.5.3, is suitable here.

GT: Thanks for the specification. It has been added now.

L586–589. It needs a reference of the online calibration. Are Hajima et al. (2020) and Lawrence et al. (2019) suitable references in this context?

GT: Thanks for the comment. The Hajima et al. (2020) and Lawrence et al. (2019) were cited as both of them, though as model description papers, discussed the model uncertainties. After consideration, we think

more direct discussion on the model uncertainty would be more appropriate for this citation here. Thus, we have now replaced it with the followings:

Model Structure and Climate Data Uncertainty in Historical Simulations of the Terrestrial Carbon Cycle (1850–2014) https://doi.org/10.1029/2019GB006175

Insights from Earth system model initial-condition large ensembles and future prospects. https://www.nature.com/articles/s41558-020-0731-2

L732. It may need 'in low SSP scenarios' after 'NorESM2-LM'.

GT: Thanks a lot for checking. It is added for clarification.

L741. Citing Meyerholt et al. (2016) is more suitable at the previous sentence.

GT: Thanks for the comment. I have added the citation to both sentences as the values mentioned are taken from the original literature.

L763–764. I don't understand how this sampling is enabled from the set of single parameter value for each model. The MCMC sampling may need supporting information.

GT: Thanks for the comment. We have now specified the sampling method.

*To examine the correlation of parameter values and feedback separation, we applied Markov chain Monte Carlo (MCMC) sampling for the sensitivity parameters and turnover times for each of the individual ESMs (60 walkers × 1,000 iterations = 60,000 runs, starting from the "best-estimate" parameter values).*

L796. 'flat10' needs definition.

GT: Thanks for checking. It has been added now.

*[e.g., 1pctCO2 or flat10 (constant emissions of CO2 of 10 GtC per year)]*

Appendix

Table A1. Missing values in UKESM1-0-LL need explanation.

GT: Thanks. The missing values are because UKESM does not have a litter pool. It is explained in the notation now.

*Missing values for UKESM1-0-LL are due to the absence of a litter pool in this model, resulting in no turnover time or feedback-related parameters for the litter pool.*

L897–899. For the land organic nitrogen pool size, the differences between the models are too large to identify the trend from the figure.

GT: Thanks for the comments. I have now removed the "land organic nitrogen pool size" as the trend is not that obvious. The revised is as follows:

*The trends for the carbon pool size and carbon:nitrogen ratio exhibit a similar pattern.*

Figure A2. It seems that the left panel shows four cases although the legend contains seven cases.

GT: Thanks for the comments. We have now updated Fig. A2 as follows to make it clear:

[Figure]

Text A1. This text does not necessarily support the discussion on Figure A5 and may be omitted. I understand that the magnitude of inter-model spread is consistent with the magnitude of forcing changes, and I don't think that 1pctCO2 is special.

GT: Thanks for the comments. The supporting discussion here highlights the temperature profile is different among ESMs. Since we were doing an offline calibration (i.e., prescribed temperature and CO2 concentration), this could partially contribute to the emulation differences. We agree that "magnitude of inter-model spread is consistent with the magnitude of forcing changes". However, with the same forcings in 1pctCO2, the large spread of temperature projection in ESMs suggested model structure uncertainty.

L904. Is the description about the initial condition appropriate? I think that it is an issue of ESM spin-up rather than internal variability.

GT: Thanks for the comments. From my understanding, the spin-up (of different models) leads to differences in the initial condition (of different models), while running one single model with various initial conditions explains the uncertainty from the model internal variability. Here we intended to say that model internal variability is important for the uncertainty of modelled carbon-nitrogen cycle while the different initial states (of different models) from spin-up further complicates the model comparison.

Figure A7. Trivial one values may be omitted for simplicity. Missing values in UKESM1-0-LL need explanation.

GT: Thanks for the suggestion. We aim to provide the full data for Fig. A7 and most of them are needed for the discussion. We have now added the explanation for missing values. The new figure caption is as below

*Figure A7. Correlation of turnover times and feedback-related parameters from the CMIP6 ESMs. The numbers indicate Spearman's correlation coefficients (r) between pairs of parameters, with * denoting p-values < 0.001. Correlations between temperature sensitivities and plant nitrogen uptake sensitivities are highlighted in yellow, while correlations between turnover times and plant nitrogen uptake sensitivities are*

*shown in bold. Missing values for UKESM1-0-LL are due to the absence of a litter pool in this model, resulting in no turnover time or feedback-related parameters for the litter pool.*

Code and data availability

To ensure reproducibility, it is recommended that the processed CMIP6 outputs described in 3.1 be included in the data, and that the calibration procedures described in 3.2 be included in the code.

GT: Thanks for the suggestion. We have now uploaded the processed CMIP6 outputs. We have updated the description for calibration details and also the data availability.

The calibrated data is provided in a Python pickle file, but reading the pickled object seems to require associated modules not provided.

GT: Sorry for the inconvenience. Now the updated data included an .csv file for the calibrated parameter values.

---

## Author Comment (AC3)

**Author Comments (ACs)**

In this Author Comments:

- The original referee comments are in black (directly copied from the comments).

- Our responses are in blue.

- *The text we quoted from the manuscript is in gray italics.*

We sincerely thank all referees for their constructive comments and feedback on our manuscript.

Best regards,

Gang Tang (GT, as referenced below)

on behalf of all co-authors

Top-Level Updates Before Addressing Individual Comments:

- Title Revision:

  The manuscript title has been updated to "Synthesizing Global Carbon-Nitrogen Coupling Effects – the MAGICC Coupled Carbon-Nitrogen Cycle Model v1.0" This new title more accurately reflects the content and scope of the manuscript and is in line with the title used for other modules of MAGICC (e.g., Synthesizing long-term sea level rise projections – the MAGICC sea level model v2.0, https://doi.org/10.5194/gmd-10-2495-2017).

- Terminology Clarification:

  To avoid confusion, we now exclusively use "MAGICC" to refer to the full model (the online model including all components) and "CNit" solely for the coupled carbon-nitrogen cycle model. This eliminates the previous ambiguity caused by the frequent use of "MAGICC" in varying contexts.

**RC2: 'Comment on egusphere-2024-1941',        Yann Quilcaille, 21 Oct 2024**

This manuscript proposes the description of a new module for a coupled carbon-nitrogen cycle in MAGICC, with its calibration based on land surface models and a set of CMIP6 ESMs. Overall, the model is correctly described and well presented. Its design is common to climate emulators, with pools and fluxes representing the essential elements and processes. This well-established approach does indeed improve simplifications, but with appropriate reasons, and leaving room for improvements or sophistication in future works. The comparisons to the training data show relatively good performances, albeit lower for CMIP6. Besides, adding this CN module to MAGICC would be an important improvement on this key climate emulator, enhancing the robustness of future constraints on the land carbon cycle. To summarize, this work is then important, timely and relatively well presented, and I consider that it fits perfectly the scope of GMD.

Nevertheless, after careful consideration, I would recommend that this manuscript should go through major corrections before publication. The major reasons are the oversimplification on the plant uptake of N (comment 1) and the lack of clarity in the data handling (comment 2). The first one would require either a better justification or to account more explicitly for the inorganic N pool either in the plant uptake of N or in the limitation on the NPP. The second one is necessary for better understanding by the readers.

There are some additional minor points (comments 3-9), that matter for the overall quality of the manuscript, but are not sufficient to justify a major correction. Finally, some details are simply brought to the attention of the authors (comments 10-13), without requiring any action.

GT: Thanks a lot for reviewing our paper and providing us with constructive and positive comments, especially those regarding the model formulations. For the major concerns, actually we (the author team) had a lot of discussions on the first one (nitrogen plant uptake simulation and nitrogen limitation on NPP) at the model development stage. The data handling details are added now for clarification. Please see our justifications of both in the respective responses.

Major comments:

1. Modelling of the plant uptake of N

The plant uptake is currently a relationship based on the NPP, with a temperature dependency (equations 22, 24 & 27 in Section 2.4). First, a required plant uptake given the NPP is estimated. Then, the limited NPP is estimated. Thus the plant uptake given this limited NPP is obtained. The limitation on the NPP depends only on atmospheric deposition and the required plant uptake.

I was expecting to see either the plant uptake or the limitation to NPP to depend on Nmin, the pool that provides the N. If the limitation on NPP would have depended on Nmin here, it would have affected the actual plant uptake. Yet, Nmin affects neither the limitation of the NPP nor the plant uptake. Having this disconnection between Nmin and the plant uptake or the limitation of NPP could cause inconsistencies, for instance in the following examples:

- For an excessive fertilization, we should expect no limitation from the N cycle on the NPP. Yet, in the current modelling, Nmin would be saturated but it would have no effect either on the limitation $\epsilon CN(NPP)$ or on the P uptake. Thus excessive Nmin isn't accounted for. It may be the reason for the discrepancy described Lines 457-458.

GT: Thank you for your comments. The key question here is whether the mineral nitrogen supplied by fertilizer application (the FT flux in our model) can enhance ecosystem NPP. We discussed whether to combine atmospheric deposition and fertilizer application as the external nitrogen forcing for the ecosystem.

Ultimately, we decided against this, as fertilizer use primarily boosts productivity in agricultural systems, which is harvested within the annual timescale. The current version of MAGICC focuses on the terrestrial ecosystem and does not simulate agricultural dynamics. The nitrogen that is not utilized by crops either releases to the atmosphere or leaches into the ocean, making it effectively unavailable to the ecosystem. Including fertilizer application in our model could introduce bias, as it might unrealistically fertilize the ecosystem. To avoid this, we excluded this flux from the nitrogen limitation.

- In the current modelling, without fertilization and atmospheric deposition, the only flux going in Nmin is $LDN2M$, ie the fraction of the decomposition flux from litter to the inorganic N pool. According to equations (34) and (37), this flux could become very small. Thus, the Nmin pool could be depleted by a continuous plant uptake and an insufficient decomposition flux, and then would become negative. Thus insufficient Nmin isn't accounted for. There may be a link with the issue mentioned in Lines 666-668.

GT: Please note the nitrogen that predominantly goes into the mineral nitrogen pool is the $SR_N$ (Fig. 1), which is the soil organic nitrogen decomposition (mineralization of soil organic matter). The sum of $SR_N$ and $LD_N2M$ is effectively the net mineralization nitrogen. In most of the cases, either from the complex models or our emulations, the net mineralization is comparable with (usually a bit higher than) the nitrogen plant uptake at the annual time step. This will prevent the mineral nitrogen pool from depletion. We have now made this clear in the equation explanation, as quoted below:

*The sum of the fraction of litter decomposition nitrogen entering the mineral pool (LD2Mn, i.e., litter mineralization) and the nitrogen released during soil respiration (SRn, i.e., soil organic matter mineralization) constitutes the ecosystem's net mineralization is effectively the ecosystem's net mineralization.*

So far, the only explanations are Lines 183-191, mentioning that plant uptake minus net mineralization is linked to the required plant uptake. However, this is insufficient to justify removing the dependency on the size of the inorganic N pool, either in the (required) plant uptake or in the N limitation on the NPP. I checked the papers proposed as sources, but I could not find a justification for this modelling of the plant uptake:

- Zaehle and Dalmonech, 2011, 10.1016/j.cosust.2011.08.008: The approaches outlined in sections "Nitrogen limitation on plant C uptake" and "Plant nitrogen uptake and competition with soil microbes" insist on the need to represent N availability/limitation, without introducing the approach shown in the reviewed manuscript.

- Zaehle et al., 2014, 10.1111/nph.12697: none of the equations introduce the equations of the reviewed manuscript. Eqn 4a-c would actually confirm a dependency of the plant uptake flux with Nmin.

This point is crucial for the modelling of the N uptake and NPP limitation, thus I strongly recommend the authors to properly justify this modelling, or adapt it. Adequate sources making use of this modelling or observed relationships would be needed. Additionally, if these equations were still used, biophysical justifications would be needed in my opinion. Finally, discussions on the limits would be needed as well, e.g. in the two cases outlined earlier.

GT: Thanks for careful checking. We really appreciate the posted question. First, the cited works are simply to support "*linking plant nitrogen status with net primary production (or photosynthesis) is common in complex models*". Back to the question: Why not use mineral nitrogen pool size to explicitly represent the nitrogen availability and use it for the nitrogen limitation on NPP. The reason is simple: The mineral nitrogen pool size is magnitude smaller than the nitrogen plant uptake requirement at the annual-mean scale. This is explained in our manuscript:

*This formulation is transformed from complex models with the key idea of comparing mineral nitrogen availability and plant nitrogen requirement. In complex carbon-nitrogen models, the nitrogen availability is typically based on the current mineral nitrogen pool size (with mass unit) and the nitrogen requirement is computed from the integrated fluxes in a given time step (with mass unit) (Thornton et al., 2007; Wiltshire et al., 2021; Zaehle et al., 2014). The competition from microbial immobilization is also considered in some complex models. However, in a model with a much longer time step (e.g., annually) like ours, such a system would be inherently unstable since the mineral nitrogen pool size would be orders of magnitude smaller than the annual nitrogen demand (i.e., the system would be unstable because the turnover of the mineral nitrogen pool would be substantially smaller than the time-step).*

Further explanation and background:

In complex models, nitrogen limitation is commonly represented by comparing the current mineral nitrogen availability to nitrogen requirements. In such models, the availability may be based on the mineral nitrogen pool size, as the required nitrogen is typically comparable to the pool size over short time steps. However, in a global-mean, annual-averaged box model, this direct comparison is not feasible, as converting nitrogen requirements (measured in N per year) to finer resolutions would either introduce unnecessary assumptions (e.g., monthly or daily estimates) or require additional parameters specific to each land surface model, which is undesirable.

The current formulation, using the required plant uptake (PUreq), is more reasonable when considering the sources of mineral nitrogen: net mineralization (predominant), biological nitrogen fixation (which is already fixed by plants within one year and thus not included in our nitrogen mass balance), atmospheric deposition, and fertilizer application (primarily relevant to agriculture). As explained in the manuscript, net mineralization directly supports plant uptake, which is why the mineral nitrogen pool size does not fluctuate significantly—plant uptake and net mineralization tend to balance each other.

Using net mineralization, which is on the same order of magnitude as plant uptake at the annual-mean scale, is therefore a more reasonable approach than using the annual mean size of the mineral nitrogen pool. The key assumption in our model is that net mineralization is linearly correlated with plant uptake, meaning the unmet plant uptake requirement from net mineralization can be expressed as a linear function of the required plant uptake (i.e., required plant uptake minus net mineralization = $f_2 \times$ PUreq, where $f_2$ is a constant). This assumption is supported by the relationship between plant uptake and net mineralization data (below).

[Figure]

*Figure A2. Relationship between nitrogen plant uptake and net mineralization as simulated by CABLE and OCN.*

This point is the main reason for switching the manuscript from Minor to Major Correction. If I have missed something, I am sorry. I tripled-checked, but could not find anything to prove me wrong. Therefore even if I were indeed wrong, other readers may also miss the point, and it would be necessary to clarify.

GT: We really appreciate your careful reading of our manuscript and posing this question. We agree that there might be something missing in our writing. Thus, I have rearranged the writing and added new subtitles for better clarity.

**2. Lack of clarity on data handling**

The Section 3.1 "Data acquisition and processing" isn't clear enough at the moment. CABLE & OCN provided CMIP5 runs on RCPs. Besides, some ESMs provided CMIP6 runs on SSPs. Both sources are used for calibration, hence the analysis in Section 3.4 and further. However, the authors write that:

"Unfortunately, a robust and feedback-specific emulation is not feasible for CMIP6 ESMs, as the results from experiments without the nitrogen effect are unavailable." (lines 363-364)

Apparently, the authors still managed to train and emulate the CN module of MAGICC? Is it about the robust and feedback-specific part on the training & emulation, and the ensuing workaround outlined Section 3.2?

GT: Thanks for the comment. Here we mean, since CABLE and OCN provided the nitrogen on/off experiments, the calibration would recognize the difference caused solely by nitrogen. But for CMIP6 ESMs with a nitrogen cycle, none of them provide the nitrogen off experiment. Thus the disentangled nitrogen effect in the CMIP6 ESMs is solely based on our emulation, though we applied constraints that were informed by the CABLE and OCN calibration. We cannot say the nitrogen effect in CMIP6 ESMs from our emulation is 100% robust. But when the emulation turned out matching the ESM outputs, it gave us confidence that at least our model is not too wrong. The only way to get a robust emulated nitrogen effect is to calibrate the CMIP6 EMSs with the nitrogen off experimental data - which is, unfortunately, not available for the current ESMs. We hope the modeling groups will run such experiments in the future but we also understand that this might be too computational demanding. That is also why we have a lot of discussion and further analysis about the limitation of the disentangled nitrogen effect in Section 5.3.

Besides, it remains unclear how data from CABLE & OCN is used for training in comparison to CMIP6. Is it a two-step training, first on CABLE & OCN, then on CMIP6, i.e. using the parameters obtained from CABLE & OCN as a first guess for the optimization on CMIP6 data? Or is it a one-step training, pulling all samples together? Although the results of the calibration are shown for CABLE & OCN in Section 3.3 and for CMIP6 models in Section 3.4, the Section 3.2 explaining the calibration setup does not mention CABLE or OCN.

I strongly suggest clarifying how both datasets are used precisely, and the questions that I present here. Ideally, a reader should not have to re-read this section to understand the data flow. It may require the authors to adapt the structure of Section 3.1 and 3.2, but it would be worthwhile for the readers.

GT: Thanks for the comment and suggestion. We realized this calibration section is not clear. We have now rewritten this section with details about the processes we calibrate our model.

Minor comments:

**3. Land-use & deforestation**

I do appreciate the effort in modelling land use in the C & N cycles, but I have to flag two important limits.

The first one is in the parametrization of the regrowth flux (Lines 313-318). At the moment, the regrowth depends only on the gross deforestation, with two constant parameters $\varphi$ and $rrgr$ (equation 40). Yet, I would rather expect the regrowth of a deforested parcel to depend on its NPP rather than how much was

deforested. In other words, a primary forest would have a high C stock; deforesting would be a strong C flux because of its past unperturbed growth under a favourable climate; but its regrowth under a less favourable climate would be towards a lower C stock. A potential correction would be to approximate the deforested area using the ratio of $LUgrsd$ with $CP$ (thus neglecting $CL$ and $CS$). The regrowth of this area would depend on the NPP, with some parameters to account that the regrowth on a deforested area is not exactly the same than the one aggregated on all biomes as modelled by MAGICC-CN.

Additionally, these fluxes are not only due to deforestation (lines 308-312). Biomass extraction from croplands will also matter a lot, especially for the N cycle. This is an important limit of this current modelling, that must be mentioned.

To be clear, I'm not asking the authors to modify the C-N modelling to account for both effects. I am aware that it would require an extensive work (example with OSCAR as illustrated in Gasser et al, 2017: 10.5194/gmd-10-271-2017). This manuscript already provides a significant modification. However, I suggest to explicitly mention both limits in the manuscript, and keep them for future developments of the model.

GT: Thank you for the thoughtful suggestions. Regarding the first limitation, I agree with the mechanisms you mentioned - ecosystems with different NPP will exhibit different regrowth patterns. However, implementing this in a global box model is quite challenging. In my view, it is nearly impossible to model global-averaged NPP as a whole while accounting for such dynamics.

The correction you suggested, if I understand correctly, involves manually separating part of the deforestation flux and assuming it occurs in primary forests, then calculating regrowth based on NPP. This makes sense biophysically and could work, but I have some concerns about its implementation:

a) If regrowth is linked to NPP, it would likely need to reference the NPP from the previous time step (i.e., the unperturbed NPP before deforestation). This requires a reference NPP for each deforestation event, which would change with each time step due to the independent nature of deforestation perturbations. Determining this reference NPP is complex, and it raises the unresolved question in complex models: How do we isolate the effects of different perturbations over time? In an ideal scenario, complex models would need to run an undisturbed experiment first to establish a reference state for each time step, but this is not feasible for our emulator.

b) The fraction of deforestation dependent on NPP is difficult to determine.

c) Ensuring mass conservation with this parameterization would be challenging.

d) We lack sufficient data to constrain the additional parameters needed for regrowth.

In conclusion, while the current formulation is not perfect, it represents a balance between simplicity, functionality, and the availability of calibration data.

For the second limit, I think the current formulation, with the regrowth fraction (/phi), has somewhat captured the biomass extraction from harvest.

To avoid the discussion going too far away, here I also share some justifications for the current formulation. First of all, we have to admit the land use perturbation interacts with nearly every part of the land ecosystem and the interactions are largely heterogeneous. As an emulator focusing on the coupled carbon-nitrogen cycle, we are not aiming to tackle the complexity of land use perturbation itself. The current formulation adapts the idea of regrowth as a linear function of regrowth time (constant regrowth flux ⇒ integration is linear biomass growth), which is not too wrong considering that in the natural world the plant grows as a sigmoidal function. I also had discussions with Thomas Gasser from the OSCAR team, but the preliminary conclusion is, instead of further updating the formulation in MAGICC, using the direct output from OSCAR (or other sources) might be more straightforward. So in summary, we keep the current

formulation for its functionality. The further improvements will be possible only when the data is available (from ESMs, UNFCCC, LUMIP etc).

As suggested, we have now added some discussions on the limitations, as quoted below:

*The formulation presented here has certain limitations. For example, it does not account for variations in regrowth rates among different ecosystems or across ecosystem successional stages - an inherent constraint of the global box model approach. Additionally, it aggregates deforestation and harvest fluxes into a single LU input, although the regrowth fraction parameter may provide some indication of harvest activities that do not result in regrowth.*

**4. Calibration**

**a. Ensemble members**

It is not detailed which ensemble members are used for the calibration of the CN module in Sections 3.1-3.2. Is that only the first one, e.g. r1i1p1f1, or all available ones? If all, is there an averaging? I recommend the authors to give some information on these questions.

Additionally, is there some form of weighting on the sample from the samples from the SSPs and the historical, to account for a varying density of points in the space of predictors. To be clear, I'm not asking the authors to apply such a weighting, but I'm asking whether they apply it, and simply suggest to mention it. There are imperfect solutions, like accounting for the length of the runs, and more sophisticated ones, like the inverse of the density in the predictor space, but such solutions may not be feasible for simple climate models.

GT: Thanks for the suggestion. We have now added the ensemble information ("variant_label" in CMIP6 global attribute). We did not use all the ensemble members as they are of different realizations, initializations, physics, and forcings. The ones we used are now specified in Table 1.

**b. Base period for calibration**

At the moment, the base period is the first year, at least for temperatures (Lines 399-400). Due to internal variability in ESMs runs, I would recommend taking an average over a longer base period, e.g. 1851-1900.

GT: Thanks for the suggestion. For this calibration we used the values in the year 1850 for the base period (i.e., the initial state). We will consider using the average over a longer base period for our future calibrations.

**5. Model design**

**a. CO2 fertilization**

I appreciate the approach on the CO2 fertilization (Section 2.3.1), especially how to deal with an overreliance on the rectangular hyperbolic function. Yet, I would appreciate having a Figure in appendix showing the response of CO2 fertilization with CO2, for different values of the method factor, e.g. 0, 0.25, etc to 2. It would help the readers get a better idea on the impact of this parametrization.

GT: Thanks for the suggestion. We have now added the illustrative figures for the method factor for both CO2 fertilization and temperature effect on NPP.

[Figure]

*Figure A1. Illustration of the functionality of the method factor for CO2 fertilization (mCO2) and temperature feedback (mdT). An mCO2 of 0, 1, and 2 represents the logarithmic, rectangular, and sigmoidal CO2 fertilization formulations, respectively (Eqs. 17 and 18). Similarly, an mdT of 0 and 1 corresponds to the exponential and sigmoidal temperature response formulations, respectively (Eqs. 19 and 20). Intermediate values represent a linear combination of the two formulations.*

b. Overfitting?

The model proposed for the CN module is very well designed, I appreciate the representation of the crucial fluxes and pools in a synthetic approach. Yet, the high flexibility in the parametrizations of the fluxes make me wonder about overfitting. For instance, to what extent should the BNF flux be split between the plant, litter and soil pools? Given all fluxes being split, isn't there a risk to have a spurious & non-physical parametrization of the cycles?

To answer these questions, I would have two recommendations. First, the Table A.1 should include the significance of the coefficients, with a discussion in the manuscript. Then, the differences in the N cycles in ESMs could be further discussed, be it for the partitioning or the behaviours. Of course, an exhaustive analysis would make a full paper, but I would suggest to keep it to one paragraph.

GT: Thanks for the suggestions. This manuscript aimed to present the calibration only from the best-estimate parameters (i.e., we only extract the parameter sets for best-fitting). We were planning to do a more comprehensive analysis on the parameter uncertainties using MCMC, with which the posterior distribution would be more straightforward to see if there is overfitting, correlation, or redundancy of parameters. For nitrogen behaviors, we have done some preliminary analysis for the parameters (Section 5.3), which supported that the nitrogen effect parameters are necessarily constrained in the calibration. For the current calibration, the fractionation factors are set completely free in this calibration (the only constraint is the sum of partition==1), which is why we did not discuss them in the manuscript.

6. Performances for CMIP6?

In Section 3.2, the authors write that for MIROC-ES2L and UKESM1-0-LL, the NPP over 1pctCO2 is higher than in SSP126, while the opposite is seen for the plant uptake. They conclude in an inconsistency in their modelling. I would argue that it is not necessarily inconsistent for two potential reasons. First, the 1pctCO2 does not assume any change in land management, thus no increase in fertilization, while SSP126 does.

Figure 2 shows good performances for the CN module on CABLE & OCN. For CMIP6 models, Figure 3 shows a more contrasted image. The authors explain issues for instance related to the Nmin pool of the

ESMs, but there are still important fluxes that seem not adequately modelled. For instance, MIROC- ES2L exhibit differences on the NPP.

I would be interested in seeing the comparison up to 2300, which is provided for MIROC-ES2L.

GT: Thanks for the comments. "the 1pctCO2 does not assume any change in land management, thus no increase in fertilization, while SSP126 does.", if I understand it correctly, it actually supports our assertion that "for MIROC-ES2L and UKESM1-0-LL, the NPP over 1pctCO2 is higher than in SSP126, while the opposite is seen for the plant uptake. They conclude in an inconsistency in their modelling." There is no fertilization in 1pctCO2, the PU should be lower (agrees with the model output), the NPP should also be lower (not enough N for NPP). I have rechecked our discussion (as quoted below). Our key point here is: NPP and PU should be compatible with each other (which is the assumption of our formulation and it makes biophysical sense).

*The underestimation is mainly from the inconsistent behavior of these two ESMs in the idealized 1pctCO2 experiment and hist_SSP experiments. Both ESMs have simulated higher NPP at the end of their 1pctCO2 runs (~100 GtC/yr in 1999 for both ESMs) than that at the end of their SSP runs (e.g., SSP126, <80 GtC/yr in 2100 for both ESMs), which is contradictory to their plant uptake results (lower in 1pctCO2 and higher in SSPs, Fig. A4). Such behavior is in direct contradiction with our assumption that higher NPP requires a higher PU (section 2.4, Eq. 26).*

We have also explained why "MIROC- ES2L exhibit differences on the NPP", which is quoted below:

*Since NPP and PU are both set as calibration targets, MAGICC has tried to minimize the gap between the emulated fluxes and targets, resulting in the simultaneous underestimation of NPP and overestimation of nitrogen plant uptake for UKESM1-0-LL and MIROC-ES2L (Fig. 3 and Fig. A4). However, such different behavior is only observed in the 1pctCO2 experiments for these two ESMs, indicating there could be either some model response nonlinearities between their 1pctCO2 and SSP runs that our model is not capturing or some regional distinct effects that we are not seeing in the global, annual averages.*

Please also note we have provided the results to 2300 for MIROC-ES2L (where available) in Fig. A4.

7. Showing the C:N ratio of plants

I would be curious to see the C:N ratio of the plant pool in SSPs. There is one mention Line 552, but this is for the land, while I would consider the one for the plant pool to have a stronger interpreraton. Current Figure A4 seems to suggest a varying C:N, in particular in 1pctCO2. I would appreciate such a figure in the appendix, if the authors agree that it would provide worthwhile inputs for the manuscript.

GT: Thanks for the comments. For Line 552 we are talking about the delta land C / delta land N rather than the absolute land C/N to emphasize the stoichiometric relationship between carbon and nitrogen in the carbon or nitrogen accumulation (with exceptions like CMCC model's 1pctCO2, the loss of organic N). The absolute land C/N ratio is mentioned and plotted in Fig. A6 and Text A1, where we described the varying CN ratio in the 1pctCO2 experiment. I have tried to plot the plant C/N ratio (the figure below) but it has the same trend as the land C/N ratio (the multi-model mean, with considerable model difference).

[Figure]

**8. Mentioning before the limit on resolutions**

The aggregation to global & annual resolutions is an usual limit of the simple climate models. This is typical from these models, because their model design is not meant to analyse spatial heterogeneity, but rather the Earth system modelling through the interaction of many processes. It should be the first limit reminded in the Section 5, yet it is for now the last point in Section 5.3 (Lines 797-800). These lines do apply to the content of Section 5.3, but it applies as well to Sections 5.1 and 5.2. Thus, it would make sense to mention the issue of resolutions from the beginning of Section 5.

GT: Thanks for the suggestion. We have now moved this to the beginning of the limitation section.

**9. Limits on modelling to mention as potential future works**

In my opinion, the Section 6 "Conclusion and future works" should remind the limits mentioned in Section 5 as potential future works. For instance, the comments in Lines 712-720 clearly suggest that this modelling is just a first step. It is common for simple climate models to be designed that way, to start with a first simple version, and then to sophisticate where necessary. The authors mention oversimplifications, I mention others in comments 1 and 3, such limits can be future works.

GT: Thanks for the comments. We have now specified these points you mentioned in the future work part. The revision is quoted below:

*Therefore, the current formulation and treatment of these aspects in MAGICC may have to be updated too, while aiming to continue to strike a balance between model simplicity, process representation, and emulation performance, reflecting a fundamental design principle for RCMs and MAGICC in particular.*

*Future work on MAGICC's carbon-nitrogen cycle will focus on the online calibration of the full MAGICC structure to CMIP6 ESMs (and/or observational data), evaluation of model performance with respect to computational efficiency, incorporation of additional constraints, uncertainty quantification, sensitivity analysis, application of probabilistic projections, and continued model development (e.g., land use emission implementation and nitrogen process representation) to align with advances in complex models and emerging theoretical frameworks.*

Details:

**10. Position of Figure 1**

The Figure 1 is crucial to visualize the design of the CN module. It should appear early for the readers to structure its understanding of the model. At the moment, it is only at the very end, in Section 2.7, which is too late. I strongly suggest shifting Figure 1 to the Section 2.2 for improved clarity.

GT: Thanks. We have moved this figure and revised the writing to provide an overview of the model.

**11. Difficulties in calibration due to data reporting by ESMs**

I congratulate the authors for acknowledging that, and explaining how. This is a recurring issue in CMIP exercises. Although technical, it does matter a lot for calibration, and it may be useful to raise awareness on this issue.

GT: We have written another paper to discuss this issue in detail and hopefully it could get attention from the CMIP and ESM community (Investigating Carbon and Nitrogen Conservation in Reported CMIP6 Earth System Model Data, https://doi.org/10.5194/egusphere-2024-3522). I am also planning to contact as many CMIP6 ESMs as I can to specifically elucidate their land carbon-nitrogen data.

**12. Code of MAGICC-CN**

The code is well structured, relatively well commented. However, the code of MAGICC v7 itself remains openly but not anonymously available. pymagicc is available for the v6, but not the v7. The requirement for this manuscript is met, with the -CN module provided. However, I would simply suggest that future versions of MAGICC itself should be openly AND anonymously available. Additionally, development on GitHub would provide an open perspective on the developments on MAGICC and foster collaborations.

GT: Thanks for the suggestions. I will later make the carbon-nitrogen cycle publically available as a python package (though now the code is already publicly available in Zenedo). As for the MAGICC full structure, the team is working on it to make it open-sourced. The full MAGICC code is openly and anonymously available at https://gitlab.com/magicc/magicc.

**13. RCM vs SCM**

As a simple reminder, the acronym RCM may not necessarily be great for models like MAGICC, FaIR, OSCAR, HECTOR, etc. I acknowledge that we used this acronym for the RCMIP phase 1 & 2 papers, but this choice was criticized by researchers using Regional Climate Models, thus RCMs as well… At some point, the community of climate emulators should decide what to do, RCMs, or SCMs (Simple Climate Models), or else.

GT: Yea let's see. I personally do not have a preference for RCM or SCM.

---

## Author Response (AR2)

**Author's response**

Dear Editor,

We have now fixed the technical issue by updating the data and code repository in Zenodo. We have now provided the customer modules and a csv file for the parameter values. This revision only changes the Zenodo link. Thank you for your assistance.

Best regards,
Gang